# Direct observation of the Migdal effect induced by neutron bombardment

Difan Yi[1,2], Qian Liu[1 ✉], Shi Chen[1], Chunlai Dong[3], Huanbo Feng[2], Chaosong Gao[3], Wenqian Huang[1,2], Xinmei Jing[1], Lingquan Kong[1], Jin Li[1], Peirong Li[4], Enwei Liang[2], Ruiting Ma[1], Chenguang Su[1], Liangliang Su[5], Junwei Sun[6], Dong Wang[3], Junrun Wang[4], Zheng Wei[4], Zeen Yao[4], Yunlinchen Yu[1], Yu Zhang[4], Shiqiang Zhou[3], Zhuo Zhou[3], Bin Zhu[6], Jie Zuo[1], Hongbang Liu[2 ✉], Xiangming Sun[3 ✉], Lei Wu[5 ✉] & Yangheng Zheng[1 ✉]

The search for dark matter focuses now on hypothetical light particles with masses ranging from MeV to GeV (refs. 1–12). These particles would leave very faint signals experimentally. A potential avenue for enhancing experimental sensitivity to light matter relies on the Migdal effect[13–15], which involves the detectable ejection of electrons following the instantaneous accelerations of atoms colliding with neutral dark matter. However, although the Migdal effect could be equally generated in controlled experiments with neutral projectiles, a direct experimental observation of this effect is missing, casting doubt on the reliability of detection experiments relying on this effect. Here we report the direct observation of the Migdal effect in neutron–nucleus collisions, achieving a statistical significance of 5 standard deviations, which rests on 6 candidate events selected out of almost $10^6$ recorded events. Our experiments have determined the ratio of the Migdal cross-section to the nuclear recoil cross-section to be $4.9^{+2.6}_{-1.9} \times 10^{-5}$, in which nuclear recoils exceed 35 keVee and electron recoils span 5–10 keV. These findings are consistent with theoretical predictions. This work resolves a long-standing gap in experimental validation, which not only strengthens the theoretical foundation of the Migdal effect but also paves the way for its application in light dark matter detection.

Dark matter, an invisible yet gravitationally interacting component of the Universe[16–18], remains one of the most profound unsolved mysteries in modern physics. Although experiments focusing on weakly interacting massive particles have successfully approached the neutrino fog[1–10], conclusive evidence for dark matter has yet to be found. The dark matter is broadening its focus. A global experimental effort is now probing models in which the dark matter particle has a mass roughly between MeV and GeV. Within this range, theoretical insights strongly support the plausibility of dark matter potentially existing in the mass range from MeV to GeV with a successful thermal production mechanism[11,12]. The faint signals associated with light dark matter necessitate substantial reductions in the detection thresholds of existing detectors. Despite the achievement of detection thresholds around 100 eV in tonne-scale experiments, the abilities remain inadequate now for detecting light dark matter.

A promising new strategy to tackle this challenge is the Migdal effect. It describes a process in which energy transfers from an atomic nucleus to a surrounding electron[13–15]. When a neutral particle, such as dark matter or a neutron, interacts with an atomic nucleus, it causes the nucleus to recoil. Along with this recoil, an additional electronic recoil is produced because of the excitation of atomic electrons. This additional component can generate a signal in the detectors that is above the energy threshold. Theoretical discussions of this effect in the context of dark matter direct detection date back to the mid-2000s (refs. 19–21), and it was also considered in the interpretation of the DAMA experimental data[22]. A reformulation of Migdal's original approach has been done in ref. 23. Further dedicated investigation has since revealed its relevance for direct detection searches[24–31]. To date, measurements of the Migdal effect have been limited to nuclear decay processes involving α-decay[32–35] or β-decay[36–40]. Its role in nuclear scattering, particularly when an electrically neutral projectile interacts with the nucleus, remains unverified. This has motivated recent experimental efforts to observe the Migdal effect in nuclear scattering experiments[41–43]. However, no observational results have been reported so far, to our knowledge. Despite this gap, several direct experiments have leveraged the Migdal effect to search for sub-GeV dark matter[44–51], extending the reach of light dark matter direct detection. The lack of direct observation undermines the conclusions drawn from dark matter experiments that rely on the existence of the Migdal effect.

To address this issue, we develop a specialized gaseous pixel detector designed for high-precision imaging of nuclear recoil (NR) and Migdal electron tracks. This detector features a broad energy detection range, excellent vertex resolution ability, low noise levels and advanced imaging abilities. These design features allow us to identify NRs and

[1]School of Physical Sciences, University of Chinese Academy of Sciences, Beijing, China. [2]School of Physical Science and Technology, Guangxi University, Nanning, China. [3]College of Physical Science and Technology, Central China Normal University, Wuhan, China. [4]School of Nuclear Science and Technology, Lanzhou University, Lanzhou, China. [5]Department of Physics and Institute of Theoretical Physics, Nanjing Normal University, Nanjing, China. [6]School of Physics, Yantai University, Yantai, China. ✉e-mail: liuqian@ucas.ac.cn; liuhb@gxu.edu.cn; xmsun@mail.ccnu.edu.cn; leiwu@njnu.edu.cn; zhengyh@ucas.ac.cn

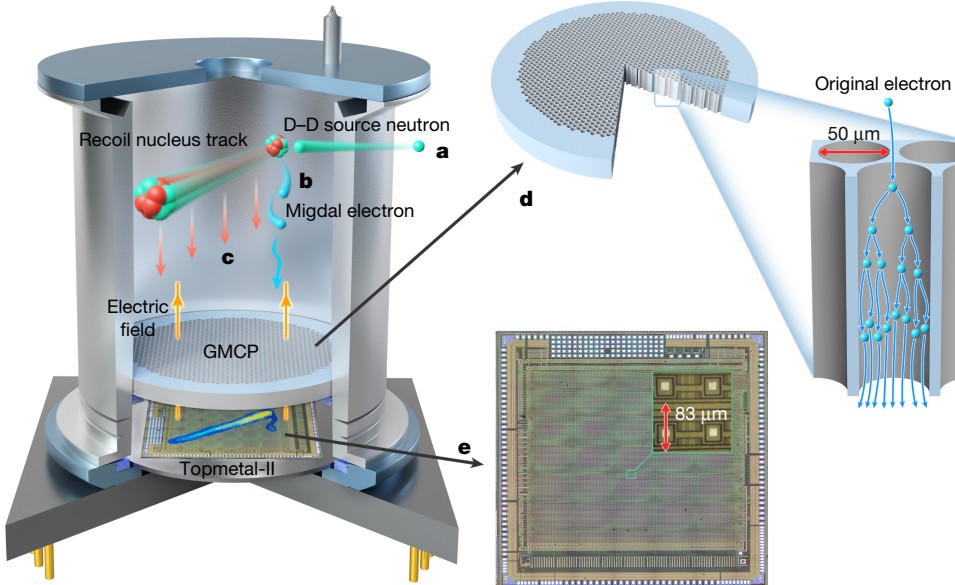

**Fig. 1 | Schematic of gas pixel detector structure and working principle.** The detector consists of a beryllium window at the top, a ceramic tube, a gas microchannel plate (GMCP) in the middle and a Topmetal chip at the bottom, with an electric field applied within the detector to guide electron drift. **a**, The neutron beam passes through the effective area of the detector. **b**, Neutrons scatter with gas molecules in the detector, generating recoiling nucleons. While in the recoil process, atoms emit Migdal electrons. **c**, Charged particles interact with the gas in the drift region, causing ionization of gas atoms. The resulting electrons are drifted by the electric field towards the GMCP. **d**, Electrons enter the pores of the GMCP and undergo ionization multiplication in a strong electric field, amplifying the signal. **e**, The Topmetal-II induces charge signals, in which the distribution and quantity of these ionized electrons reflect the track and energy deposited by the charged particles.

Migdal electrons with a marked reduction in background noise. In this study, we report the direct observation of the Migdal effect. In the experiment, we identify six signal candidates, exceeding the rate of background reactions by 5 standard deviations. The measurements indicate a Migdal effect cross-section relative to the NR cross-section ratio of $(4.9^{+2.6}_{-1.9}) \times 10^{-5}$. Our findings provide robust evidence for enhancing detector sensitivity in direct light dark matter experiments using the Migdal effect.

## Experiment

Confirming the Migdal effect requires the simultaneous observation of both the recoil nucleus and the Migdal electron, with the two tracks forming a topological structure with a common vertex. This places stringent requirements on the imaging ability and position resolution of the detector. In our experiment, we use a mixture of 40% helium and 60% dimethyl ether (DME, $CH_3OCH_3$) as the track imaging medium to ensure the formation of clear and sufficiently long electron tracks. We use a charge-sensitive pixel array chip[52] with a pixel size of 83 μm for track imaging and an equivalent noise charge of 13.9 e⁻. The working principle of the detector is shown in Fig. 1.

To obtain the recoil nucleus signal, we use a compact D–D neutron generator[53] to produce 2.5 MeV neutrons that bombard the mixed gas in the detector. Simultaneously, to shield against background radiation such as gamma rays from the neutron source, the detector is positioned 40 cm away from the D–D neutron source and surrounded by 1 cm lead shield to mitigate environmental gamma radiation. Moreover, liquid scintillation detectors are used during the experiment to monitor neutron flux intensity and the neutron energy spectrum.

During the experiment, the detector triggers and saves the tracks, energy and time information of recoil nucleus events by comparing the number of over-threshold pixels before and after the signal arrival. To effectively distinguish tracks, the cluster segmentation algorithm[54] is applied to separate spatially unrelated tracks in each frame. Tracks with deposition energy exceeding 35 kiloelectron volts (keVee) and

vertices within the effective volume of the detector are chosen. The discrimination between NRs and electron recoils (ERs) is based on their energy deposition density $dE/dX$ (ref. 55) and circularity[56]. Here, $dE/dX$ is defined as the ratio of the energy deposited by the track to the two-dimensional (2D) projected length of the track. Circularity is a geometric feature metric used to quantify the deviation of a planar shape from a perfect circle, with its value range being (0, 1]. A value closer to 1 indicates that the shape more closely approximates an ideal circle. Under the same $dE/dX$ conditions, recoil nucleus tracks with energy deposition exceeding 35 keVee are typically longer and geometrically straighter than electron tracks. Therefore, the 2D distribution

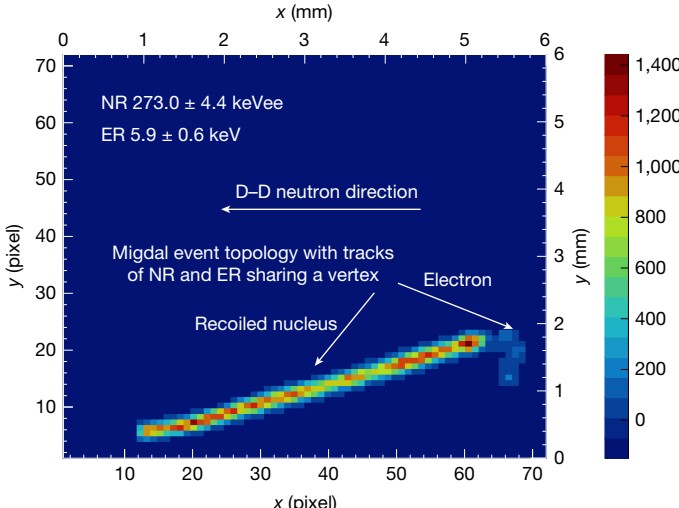

**Fig. 2 | A Migdal event recorded in the experiment.** The bright, long and straight track in the figure represents the NR track, whereas the faint track circled near the vertex of the recoil nucleus track represents the ER track. The horizontal arrow indicates the direction of the neutron beam.

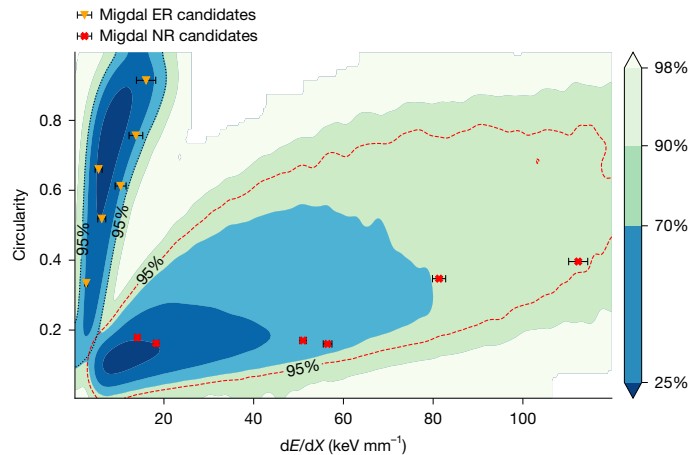

Migdal ER candidates
Migdal NR candidates

**Fig. 3 | Track characteristic distribution.** The cluster on the left represents the characteristic distribution of simulated 4–10 keV electrons, with the electron energy spectrum derived from the theoretically calculated Migdal electron energy spectrum. The differently coloured contour lines indicate the proportion of events enclosed within their respective regions relative to the total number of events. The cluster at the bottom right shows the NR events from experimental data. The black dashed contour outlines the 95% distribution region for electrons, and the red dashed contour outlines the 95% distribution region for recoiled nuclei. The measured ER and NR of six Migdal events are superimposed.

of circularity and $dE/dX$ can be effectively used to distinguish between ERs and NRs. The selection criteria for Migdal events are detailed in the Methods. To exclude low-energy background electrons, the energy deposition of electron tracks must fall within the range of 5–10 keV.

In the experiment, we collect data for approximately 150 h, and a total of $8.17 \times 10^5$ events are recorded. We identify six recoil nucleus–electron common vertex events that meet the event selection criteria, with an example shown in Fig. 2. In Fig. 3, the simulated distributions of electrons and experimental distribution of recoil nuclei are presented. The ERs and NRs are separated into two clearly distinguishable regions. The measured ER and NR of six Migdal events are superimposed.

## Background

The search for Migdal events primarily relies on the topological features of the tracks: NRs share a vertex with ERs and are produced almost simultaneously. By exploiting this characteristic, the rate of background events can be significantly reduced. We have combined experimental data with Monte Carlo simulations to estimate the expected values and associated errors for each background component, normalized to the number of recoil nuclei. Background can be divided into two categories: beam-related and beam-unrelated. In our experiment, the total background value is 0.229 ± 0.032 (stat) ± 0.043 (sys). The largest contribution comes from the background produced by the chance coincidence of NR tracks with photoelectrons and Compton electrons associated with the neutron beam. The second significant contribution is from the δ-rays produced by recoil nuclei. Other beam-related backgrounds are all less than $10^{-3}$. Significant contributions from beam-unrelated backgrounds include cosmic δ-rays and trace contaminants decay background. Details of the backgrounds are discussed in the Methods.

## Ratio of Migdal cross-section to NR cross-section

The observed signals have a statistical significance exceeding 5 standard deviations, strongly suggesting that the observed event topology is due to the Migdal effect. Based on the observed events, the ratio of the Migdal cross-section to the NR cross-section is estimated as follows:

## Table 1 | Systematic uncertainties

| | Efficiency/count | Statistical error | Systematic error |
|---|---|---|---|
| Efficiency $\varepsilon_{ER}$ | 14.4% | ±0.1% | ±1.9% |
| $n_{tot}^{NR}$ (×10⁵) | 8.17 | ±0.01 | ±0.36 |
| $n_{obs}^{ER}$ | 6 | | |
| $n_{obs}^{bg}$ | 0.229 | ±0.032 | ±0.043 |

$$P_{Migdal}(5\,keV < ER < 10\,keV, NR > 35\,keVee) = \frac{\left(\frac{n_{obs}^{ER} - n_{obs}^{bg}}{\varepsilon_{acc}\varepsilon_{NR}\varepsilon_{ER}}\right)}{\left(\frac{n_{tot}^{NR}}{\varepsilon_{acc}\varepsilon_{NR}}\right)} \quad (1)$$

$$= \frac{(n_{obs}^{ER} - n_{obs}^{bg})}{\varepsilon_{ER} n_{tot}^{NR}}$$

where $n_{obs}^{ER}$, $n_{obs}^{bg}$ and $n_{tot}^{NR}$ are the observed numbers of Migdal events, expected background counts and total NR, respectively; $\varepsilon_{acc}$, $\varepsilon_{ER}$ and $\varepsilon_{NR}$ are the acceptance rate of the events and reconstruction selection efficiency for ER and NR, respectively. The parameters and uncertainties used in the calculation are summarized in Table 1. The resulting ratio of the Migdal cross-section to the NR cross-section is $(4.9^{+2.6}_{-1.9}) \times 10^{-5}$.

## Conclusion

In this study, we present the direct evidence of the Migdal effect in neutron–nucleus scattering—a phenomenon predicted more than 80 years ago but confirmed only now with a statistical significance exceeding 5σ. This result establishes a crucial benchmark for nuclear and particle physics, providing an experimental foundation for future theoretical and experimental investigations. By validating the Migdal effect, we address a long-standing gap in the scientific understanding of fundamental interactions and offer a potential approach for the detection of light dark matter. Notably, our measured relative Migdal cross-section of $(4.9^{+2.6}_{-1.9}) \times 10^{-5}$ is in good agreement with the theoretical prediction of $3.9 \times 10^{-5}$ within the experimental uncertainties (see Methods, 'Theory').

The implications of this discovery are profound, particularly in the search for light dark matter. For decades, the Migdal effect has been an unverified assumption in neutral scattering that is regarded as an effective process for enhancing the ability to detect dark matter. Our findings provide a basis for re-evaluating these scenarios, and future work can build on these results to refine detection strategies and potentially enhance the sensitivity of dark matter searches.

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

## Methods

### Theory

For a gas mixture, the differential cross-section of Migdal effect from neutron–nucleus scattering in the soft limit is given by[26,27,57]

$$\frac{d^2\sigma}{dE_R dE_e} = \sum_i \frac{d\sigma_s^i}{dE_R} \sum_{n\kappa} \frac{dp_v^i(n\kappa \to E_e)}{dE_e} \tag{2}$$

where the summation over $i$ accounts for each species in the gas mixture. $\sigma_s^i$ denotes the scattering cross-section, which includes contributions from elastic $(n, n)$, inelastic $(n, n')$, fission $(n, 2n)$ and radiative capture $(n, \gamma)$ processes and can be obtained from the ENDF database ENDF/B-VIII.0 (ref. 58). The term $p_v^i(n\kappa \to E_e)$ is the transition probability for ionization of an electron from an initial state $n\kappa$ to a final state with energy $E_e$. This probability can be obtained from first principles using the Dirac–Hartree–Fock method[27,57]. As we use a 2.5-MeV D–D neutron beam and require an approximately 50 keV NR threshold in our analysis, the chemical bonds of C–H and C–O (about 10 eV) are easily broken at our recoil energies. Accordingly, it is reasonable to approximately treat the C, H and O as the free atoms in our NR energy regime and calculate the likelihood of electrons from Migdal scattering as the sum of the individual Migdal transition probabilities for C, H and O, respectively. Note that we include all final states with at least one electron above the energy threshold of detector, $E_e^{th}$. This is particularly relevant as it allows for a precise description of the Migdal effect in the experiment.

To compare with the elastic scattering cross-section, we calculate the total Migdal cross-section by integrating over the kinematically allowed range of NR energies and the energy spectrum of emitted electrons in equation (2):

$$\sigma_{Migdal} = \int_{\max[E_R^{min}, E_R^{th}]}^{E_R^{max}} \int_{E_e^{th}}^{E_e^{max}} \frac{d\sigma_s^i}{dE_R} \sum_{n\kappa} \frac{dp_v^i(n\kappa \to E_e)}{dE_e} dE_e dE_R \tag{3}$$

where $E_N^{max}$ can be expressed as $E_{R,max} = \frac{4m_n m_T E_n}{(m_n + m_T)^2}$, where $m_n$ is the mass of the incident neutron, $m_T$ is the mass of the target nucleus and $E_n$ is the kinetic energy of incident neutron. $E_e^{max} = 10$ keV. $E_n^{th}$ and $E_e^{th}$ are the thresholds of detector for nucleus and electron recoils, respectively. The ratio $r = \sigma_{Migdal}/\sigma_{Elastic}$ will offer a direct measure to assess the impact of the Migdal effect in dark matter detection experiments. Extended Data Fig. 1e compares the theoretically calculated Migdal differential probabilities for the gas mixture with the experimentally measured results (more detailed information of the theoretical cross-section calculations can be found in Extended Data Fig. 1 and Supplementary Information Note 1). The integrated theoretical probability in the 5–10 keV range is $3.9 \times 10^{-5}$, which is consistent with the experimental result of $(4.9^{+2.6}_{-1.9}) \times 10^{-5}$ within the margin of error.

### Detector assembly

The detector unit is sealed using brazing and laser welding to ensure high gas tightness and good mechanical properties. The main detector components—ceramics, Kovar alloys, beryllium and lead glass—have a low out-gas rate, which greatly reduces the pollution of other impurities in the gas. To fill the detector with working gas, the detector has a gas pipe that is brazed to the cathode. By using brazing technology, the metal ceramic tube shell is constructed of three layers of ceramic rings and four layers of Kovar alloy rings, with a ceramic ring placed between every two layers of Kovar alloy rings. The ceramic layer is used for insulation and positioning between the Kovar alloy layers. The Kovar alloy rings at both ends of the cermet tube shell are used to seal the connection with the cathode and the base by laser welding. The middle two Kovar alloy rings are used to install the gas microchannel plate (GMCP) as the exit electrode of the GMCP. The GMCP is installed in a cermet tube, and the support ring is extruded to fix the GMCP. The two electrodes of the GMCP are electrically connected by the Kovar alloy on the metal cermet tube. The ceramic pedestal is also composed of ceramic and Kovar alloy. A pixel chip is mounted on the ceramic pedestal and electrically connected to the ceramic pedestal using gold wire. The power supply of the pixel chip and information transmission are realized by 24 pins on the ceramic pedestal. The basic performance of the detector is tested[59], the parameters and performance of GMCP are provided[60], detailed parameters of the Topmetal-II chip are described[52,61] and the electronic information of the detector is given[62]. The structure of the detector and its geometric parameters are shown in Extended Data Fig. 2a.

### Electronic system and data acquisition

The electronics system is functionally divided into three layers: (1) the front-end electronics readout board, (2) the back-end electronics board and (3) the high-voltage board. The front-end electronics board primarily includes the gas pixel detector (GPD) unit and the GPD data readout circuit. The back-end electronics board comprises the main controller, data and firmware storage, communication interfaces and devices, and an external clock. The high-voltage board consists of the GMCP bottom-surface feedback circuit, voltage divider circuit, high-voltage chip and monitoring circuit.

**Front-end board.** It is positioned at the top of the electronics system to host the GPD. The performance metrics of the Topmetal-II can be adjusted by external voltages.

**Back-end electronics board.** It is responsible for multi-channel signal processing. The FPGA (field-programmable gate array) is equipped with flash memory, storing multiple files to ensure the system can boot correctly in the event of original file corruption. Backup configuration files are stored at specific addresses to mitigate the risk of system startup failures due to FPGA configuration errors.

**High-voltage board.** It generates and regulates high voltage. The GMCP bottom surface produces pulse signals on electron arrival, which can serve as GPD trigger signals. These signals are processed through a comparator to enhance energy resolution.

**Topmetal-II data processing.** During detection, the output data from the Topmetal-II is quantized, encoded, compressed and stored in real-time in the embedded Multi-Media Card. The GMCP bottom surface signals, coordinated universal time (UTC), system operational status and monitoring data are stored in the embedded Multi-Media Card with different headers. The scanning frequency of the chip determines the performance metrics of the detector, with pixel switching frequency required to be maintained at the MHz level. Consequently, a data compression scheme has been integrated into the data processing. The detector uses the difference compression method for data compression: the apparent diffusion coefficient (ADC) value of each pixel per frame is stored and compared with the ADC value of the previous frame. If the difference exceeds a preset value, the pixel is identified as a signal pixel and transmitted to the erasure module, thereby achieving data volume compression. The frame refresh time of the Topmetal is 2.59 ms, and the coincidence timing resolution between the GMCP signal and the Topmetal is 262 ns (ref. 63).

### Detector calibration

The energy resolution of the detector is calibrated with a 5.9-keV [55]Fe source. Photons emitted by [55]Fe pass through a collimator and then through a beryllium window above the detector, generating photoelectrons and depositing energy in the detector. The energy spectrum is fitted using a Gaussian distribution, and the full width at half maximum of resolution at 5.9 keV is obtained to be 26.54% (Extended Data Fig. 2b). Further tests on the detector demonstrate strong linearity in its energy

response, and the variation of energy resolution with energy can be described by a relationship proportional to $1/\sqrt{E}$ (ref. 64).

The deconvolution method is used to evaluate the position resolution of the detector[65]. When measuring the position resolution of the detector, the experimental distribution is treated as the convolution of the theoretical distribution with the detector resolution function. In practical terms, a flat and smooth-edged copper plate is positioned directly above the detector. X-rays propagate vertically from the source through both the detector and the copper plate. Consequently, the copper plate obstructs a portion of the X-rays, allowing only half of them to enter the detector. The resulting image obtained by imaging the X-rays with the detector shows only the half irradiated by X-rays. In this scenario, the theoretical distribution can be represented by a Dirac Delta function, whereas the experimental distribution can be expressed as the spatial resolution Gaussian function of the detector. The average $\sigma$ measured in the $X$ and $Y$ directions is 2.4 pixels (200 μm; Extended Data Fig. 2c).

## Simulation

The software framework Star-XP[66], specifically developed for this type of gas detectors, is used to simulate Migdal events. Star-XP is built on the well-established GEANT4 simulation platform, which incorporates high-precision neutron collision data from evaluated databases such as ENSDF and JEFF, with rigorous validation for neutron energies below 20 MeV (ref. 67). In particular, it uses the G4NDL4.6 dataset derived from ENDF, ensuring reliable low-energy neutron simulations. The framework considers various primary effects of charged particle motion and ionization in gas, incorporates electronic logic for digitized outputs consistent with experiments and, based on theoretically calculated Migdal electron spectra, integrates a dedicated Migdal effect generator to investigate event selection algorithms and efficiencies.

A comprehensive detector simulation has been implemented in GEANT4, incorporating all main structural components of the experimental setup, including the gas-sensitive detector, ceramic supports, GMCP glass frame, beryllium window and surrounding lead shielding. The model also contains the liquid scintillator system for monitoring neutron flux and spectrum inside the shielding enclosure. This full-scale implementation enables a realistic treatment of particle transport and interactions, allowing us to rigorously evaluate neutron activation backgrounds and ensuring the reliability of the subsequent physics analyses.

## D–D neutron flux and spectrum

The neutron flux and spectrum of the D–D neutron generator are measured and monitored using a 2″ × 2″ diameter EJ309 liquid scintillator detector. The energy linearity and resolution of the detector are calibrated by fitting the Compton edges and full energy deposition peaks of multiple radioactive gamma sources, and the parameterization is described[68,69]. The response matrix of the EJ309 liquid scintillator detector is simulated with GEANT4 (ref. 70), incorporating neutron interactions with the liquid scintillator, detector energy linearity and resolution and the neutron response function of the EJ309 liquid scintillator[68]. Neutron energy deposition is determined by applying pulse shape discrimination techniques[68] to distinguish neutron signals from gamma background, leveraging the fact that neutron signals typically exhibit a longer tail in the light emission waveform in liquid scintillator. An iterative unfolding process is used using the response matrix and measured energy spectrum. Various algorithms (GRAVEL[71] and MLEM[72]) and initial spectra (Gaussian and uniform) are tested, showing negligible differences in the unfolded neutron spectrum. Statistical uncertainties and the correlation matrix of the unfolded spectrum are obtained using the bootstrap method. The measured neutron spectrum has a peak energy at 2.5 MeV, aligning well with the simulated spectrum[73] (Extended Data Fig. 3j). The detailed information on the calibration of EJ309 liquid scintillator, neutron and gamma

discrimination measurements, and spectral deconvolution is provided in Extended Data Fig. 3 and Supplementary Information Note 2.

## Experimental details

Before the experiment, measurements of neutrons from the neutron generator and environmental gamma energy spectra are conducted using a liquid scintillator at 87° (Extended Data Fig. 4c,d). During the experiment, neutron beam monitoring is performed in the 0° and 180° directions using liquid scintillators for the D–D neutron generator beam, as shown in Extended Data Fig. 4a.

Data acquisition is conducted in two runs: run I in March 2024 and run II in July 2024. In run II, an additional detection unit is added (Extended Data Fig. 4b). Extended Data Fig. 4g,h shows the count rate variations of the Migdal detector and the liquid scintillator detector. The count rate of the liquid scintillator is normalized to the average count rate of the Migdal detector, demonstrating good consistency between the neutron beam and the detector counts. Extended Data Fig. 4e,f shows the variations in the $^{55}$Fe gain calibration during the experiment on each day. The pressure and temperature of the detector chamber are continuously monitored during the experiment, and the gas state fluctuations and good airtightness performance during the experiment are shown in Extended Data Fig. 4i–k.

## Identification of NR tracks and ER tracks

Inspired by the relevant work of the MIGDAL Collaboration[74], YOLOv8 (ref. 75) is applied for ER and NR recognition (training details and model performance are provided in Extended Data Fig. 8 and Supplementary Information Note 4). Considering accuracy and training speed, the YOLOv8m model architecture is the optimal choice for this study. By training, the model architecture can identify tracks in images and provide their classification and positional range information. Data annotation is performed using a platform called Label Studio, in which ERs and NRs are marked in the graphical interface. The training dataset consists of 6,000 experimental data track images and 2,400 simulated data images. For the experimental data images, half feature $^{55}$Fe characteristic X-ray photoelectron tracks, and the other half exhibit D–D source experimental NR tracks. For the simulated data images, half include electron tracks of 4–10 keV, and the other half represent NR tracks generated after the interaction of simulated 2.5 MeV neutrons with gas. The validation dataset consists of 3,000 experimental data images and 600 simulated data images, maintaining the same proportions as the training set. By training 200 epochs, the best-performing model achieves an ER identification accuracy of 99.0% and an NR identification accuracy of 99.7%. The model is then used to recognize and retain instances with both ER tracks and NR tracks, in which the distance between track clusters is less than 4 pixels, for further reconstruction and selection.

## Migdal event selection algorithm

To better distinguish between NR and ER tracks for events pre-selected by the YOLO program, the following procedure is implemented. To reconstruct NRs, the average ADC value of the non-zero pixels is calculated, and the pixels with values below the average are excluded. This requirement effectively removes non-NR track from the frame, as the ADC values of the pixels along the NR are significantly higher than those of other signals. The remaining pixels are fitted with a straight line using the least squares method, and the resulting line is designated as the central trajectory of the NR track. The standard deviation $\sigma$ of the resulting Gaussian distribution, which stands for the diffusion of NR, is then obtained. The two intersection points of the central track line with the image boundaries $(x_0, y_0)$ and $(x'_0, y'_0)$ are designated as initial pixel points. The endpoints of the NR track, $(x_n, y_n)$ and $(x'_n, y'_n)$, are iteratively refined using the centre-of-gravity method. Taking $(x_0, y_0)$ as an example, with $(x_0, y_0)$ as the centre and $5\sigma$ as the radius, the new centre of gravity $(x_1, y_1)$ is calculated with weighted formula $ADC \times \exp(d/d_0)$, where $d$ is the distance between the pixel and $(x_0, y_0)$

and $d_0$ is the pixel scale, and the centre-of-gravity method is applied repeatedly within this region until the calculated centre-of-gravity position converges between consecutive iterations.

To prevent the influence of NR tracks on the reconstruction of ER tracks, morphological erosion processing is applied to the NR track. All pixels with their centres at the midpoint between the two endpoints, at a distance of $4\sigma$ from the line, and within a radius of $4\sigma$, are considered as the NR and removed. To ensure the remaining tracks correspond to ER signals, the ER vertex $(x_e, y_e)$ is reconstructed for the residual tracks following the methodology described in ref. 76. The distance between the NR and ER is evaluated by

$$R = \frac{D - 4\sigma}{L_{\mathrm{ER}}} \tag{4}$$

where $D$ is the distance between the reconstructed ER start point and one of the NR endpoints, and $L_{\mathrm{ER}}$ stands for the remaining length of ER after NR subtraction. Events with $R > 0.5$ and ER track length exceeding $5 \times 83$ μm are discarded, whereas all events with ER track lengths below this threshold are retained. In other words, events are retained if the reconstructed ER vertex lies closer to the NR endpoints along the ER track. To reduce the impact of edge distortion, events are ignored if the edge ADC or edge hit count exceeds 20% of the related summation value. Moreover, events are disregarded if the NR vertex is near the electron track lying within the $3\sigma$ range from the edge, or if both ends of the track hit the edge. To minimize accidental coincidences and multi-track backgrounds, events are also disregarded if there are ADC values exceeding $4\sigma$ on both sides of the fitted line for the NR. Extended Data Fig. 5 shows the track information of six Migdal candidate events in the experiments.

## Quenching effect

The quenching effect in gases refers to the phenomenon in which the kinetic energy of ions is dissipated through inelastic collisions or excitation processes during their interaction with other molecules[77]. When ions collide with gas molecules, they transfer energy to the gas molecules, resulting in their excitation or ionization. This energy transfer process is typically inelastic, meaning the kinetic energy of the ions is converted into the internal energy of the gas molecules. In radiation detectors, the quenching effect influences the intensity and characteristics of the detection signals, causing ions with the same kinetic energy to deposit a lower detectable energy in the detector compared with electrons. The ratio between these two energies is defined as the quenching factor. We obtain the quenching factors for the working gas elements from the TRIM[78] database, as shown in Extended Data Fig. 6a, and incorporate the corresponding quenching effects into the simulation of NR tracks.

## Background

The background in the experiment mainly arises from three sources:
1. Secondary effects from neutron recoil processes: these include delta electrons and secondary NR generated during nucleon motion, de-excitation radiation from excited nuclear states and bremsstrahlung radiation from charged particles.
2. Beam-related background: this includes gamma rays produced by non-elastic collisions of neutrons, coincidental gamma rays released during the acceleration process of the neutron generator with recoil nuclei and β decay processes from neutron-activated gas atoms.
3. Environmental background: this includes radioactive elements present in the air and materials, as well as coincidences between delta electrons produced by cosmic muons and recoil nuclei.

Following is the detailed discussion of the analysis of specific background components (the results of all backgrounds are shown in Extended Data Table 1).

**Recoil-induced δ-ray.** The background of delta electrons is estimated using experimental data. First, delta electrons near the NR tracks are identified and selected. With the requirement that the electron tracks are at least 1 mm away from the vertex, the delta electrons in the middle of the track are obtained. Then, the energy spectrum of the selected delta electrons is plotted and fitted with an exponential function. The expected count of observed δ electrons in the 5–10 keV range is $0.59 \pm 0.40$ over the entire experimental period. Finally, Monte Carlo methods are used to estimate the relative efficiency ratio of selecting delta electrons at the top and middle of the NR track as 0.060, and the expected background count of δ electrons during the experiment is calculated to be $0.035 \pm 0.023$.

**Particle-induced X-ray emission.** Theoretically, the maximum energy of the Auger electrons and photoelectrons of the gas is only 0.5 keV (from oxygen)[79], which is barely detectable by the detector within the energy range of 5–10 keV.

**Bremsstrahlung processes.**
1. Quasi-free electron bremsstrahlung: in the Coulomb field of the recoiling ion, an electron from the nearby target atoms can be scattered, resulting in the emission of bremsstrahlung radiation[80].
2. Secondary electron bremsstrahlung: an electron ejected from the nearby target atoms by the impact of the recoiling ion can interact with other atoms in the target material, leading to the emission of bremsstrahlung radiation[81].
3. Atomic bremsstrahlung: the electron bound to the target atoms can be excited to highly bound states or to the continuum due to the impact of the recoiling ion. Subsequently, it can return to the initial state, emitting bremsstrahlung radiation[82]. If the electron excited to the continuum does not revert to its bound state, a phenomenon known as radiative ionization occurs, also resulting in the emission of bremsstrahlung radiation. In the analysis, the radiative ionization contribution is calculated and considered as part of the overall atomic bremsstrahlung.
4. Nuclear bremsstrahlung: this process arises from the Coulomb scattering interactions between the recoiling ion and the target atoms[83].

In general, the spectra of these four bremsstrahlung processes exhibit a layered structure within the continuum X-ray spectrum. Specifically, the quasi-free electron bremsstrahlung, secondary electron bremsstrahlung, atomic bremsstrahlung and nuclear bremsstrahlung processes predominate in distinct energy regions of the X-ray spectrum[84,85]. The theoretical differential cross-sections for the four bremsstrahlung radiation processes induced by protons or light ions with few MeV kinetic energies are described[80,82,85,86].

To estimate the expected number of electrons induced by the four bremsstrahlung processes, the expected number of X-ray emissions and the energy spectrum of X-ray emissions for each type of nucleus are estimated from the differential cross-section of bremsstrahlung. The spectrum is then input into GEANT4 to simulate the photoelectric process. A 200-μm vertex cut for NR and photoelectron is applied, and an energy cut of 5–10 keV is imposed. The resulting background expectation value is calculated to be $10^{-7}$, which can be neglected (see Extended Data Fig. 6c–f and Supplementary Information Note 3).

**Random track coincidences.** The random track coincidence is characterized through both data-driven and GEANT4 simulation approaches. In the experiment, the number of photoelectrons, Compton electrons and other possible processes that produce keV-level electrons occurring in the same frame as the NR tracks is estimated. The events with energy deposition in the range of 5–10 keV are selected and represented in a 2D distribution plot of $dE/dX$ compared with circularity. Out of $8.17 \times 10^5$ collected events, a total of 63 events are identified as photoelectrons and Compton electrons. The Monte Carlo simulation is then

applied to randomly distribute NRs and ERs in the same frame image. Through a selection algorithm, the accidental coincidences are identified as 0.29% of all cases. The expected yield of accidental coincidences in the dataset is then calculated to be 0.180.

Complementing this, full GEANT4 simulations evaluate three background components: (1) single-neutron-induced NR electron pairs; (2) independent neutron interactions producing separate NRs and electrons; and (3) neutron-generated NRs coinciding with gamma-induced electrons. The simulation accounted for all detector components and material interactions, with electrons (5–10 keV) and NRs (>35 keVee) selected using identical criteria to the experimental analysis. The simulated electron energy spectra show good agreement with experimental distributions (Extended Data Fig. 6b). The total simulated background of 0.156 events (0.128 single neutron + 0.019 multi-neutron + 0.009 neutron–photon) matches the data-driven estimate within uncertainties, validating our background modelling framework.

**Neutron activation.** GEANT4 simulations show that the production rates of unstable nuclides $^3$H and $^{14}$C are both less than 0.01 per million NR tracks with the energy greater than 35 keVee. Considering the half-life of $^3$H (12.32 years) and $^{14}$C (5,700 years), the background from neutron activation can be neglected.

**Trace contaminants.** The radioisotope content at natural abundances in He-DME-based gas mixtures is evaluated, including isotopes of $^1$H, $^4$He, $^{12}$C and $^{16}$O, as well as trace radionuclides such as $^{222}$Rn, $^{85}$Kr, $^{39}$Ar and $^{210}$Pb. Among the radioactive isotopes of $^1$H, $^4$He, $^{12}$C and $^{16}$O, $^3$H and $^{14}$C contribute almost all electrons by beta decay, with their activities in the detector being $(6.38 \pm 6.38) \times 10^{-8}$ Bq and $(4.58 \pm 4.58) \times 10^{-5}$ Bq, respectively. Other trace radioactive elements generate an average of $(7.25 \pm 0.94) \times 10^{-7}$ electrons per second with energies ranging from 5 to 10 keV. The probability that these electrons have a vertex distance less than 200 μm from ions is 0.00339. Therefore, the estimated value of the trace contaminants background, normalized to the number of experimental data, is $0.00106 \pm 0.00087$ (see Supplementary Information Note 3, Gas radioactivity).

**Muon-induced δ-rays.** The measurement results of the local cosmic muon flux in Lanzhou[87] is adopted to estimate the rate of δ-ray production in the detector sensitivity region. Considering the NR rate of D–D neutron source as 1 event per second, the ratio of δ-ray production and NR in the 5–10 keV energy region, accounting for the detector energy resolution, is $1.85 \times 10^{-5}$. As the locations of δ-ray production and NR are both evenly distributed, their positions are randomly sampled in the detector sensitivity region based on their production rate, and a vertex distance cut of shorter than 200 μm is applied. The 73% muon exclusive efficiency of the detector is also considered in the calculation. The simulation shows that the δ-rays would contribute 0.013 events, normalized to the number of experimental data.

**Secondary NR fork.** The production category, rate and energy spectrum of the primary recoil nucleus are estimated by simulating the interactions of the neutron from D–D source with the DME gas with GEANT4. The primary recoil nuclei are then put into TRIM[78] to simulate the following cascade process in the DME gas. The vertex distance between primary recoil and secondary recoil is required to be less than 200 μm, and the energy of secondary recoil nucleus should fall into the region of 5–10 keVee, considering the quenching factor[88] of NRs. By applying the cut on the 2D distribution of track deposition energy and track length, it is able to reduce the background yield to below $10^{-3}$, while maintaining a signal efficiency of approximately 98%.

### Uncertainty and significance
**Background error estimation.** The systematic uncertainty of recoil-induced δ-ray mainly comes from the YOLO model. The difference in the estimated yields between the two models (YOLOv8m and YOLOv8n) with different model sizes is considered as the systematic uncertainty of this background. For the systematic uncertainty of random track coincidences, both the error in the circularity and d$E$/d$X$ discrimination of electrons, as well as a 10% variation in the wobble discrimination parameter, and the difference in the background yields with different YOLO models, are considered to estimate the uncertainty. The main contributors to trace contaminants of radionuclides are $^3$H, $^{14}$C and $^{222}$Rn. As we have used gas sealed for over a year, the gas radioactivity should be significantly lower than the atmospheric average level. The systematic uncertainty of muon-induced δ-rays consists of the uncertainties of muon flux and detection efficiency. According to the measurement results, the impact of muon flux uncertainty can be neglected, whereas the detector energy resolution contributes a systematic error of 3%. The uncertainty of $\varepsilon_{ER}$ is taken as the difference between YOLO models. The energy resolution of the NR tracks is extrapolated based on the energy resolution $\propto 1/\sqrt{E_{deposit}}$ as described in ref. 64. The energy resolution affects the counting of events selected above 35 keVee in the NR energy spectrum, and we have taken this effect into account in the systematic error of $n_{tot}^{NR}$.

**Cross-section and significance estimation.** The profile likelihood method is used for significance calculation and Migdal effect probability error calculation. In this method, a probability model that depends on both the parameters of interest $\boldsymbol{\pi} = (\pi_0, \ldots, \pi_k)$ and additional nuisance parameters $\boldsymbol{\theta} = (\theta_0, \ldots, \theta_l)$ is used to describe the data. Denoting the density function as $f(\mathbf{X}|\boldsymbol{\pi}, \boldsymbol{\theta})$, where $\mathbf{X} = (X_0, \ldots, X_n)$ refers to independent observations, the full likelihood function can be expressed as

$$L(\boldsymbol{\pi}, \boldsymbol{\theta}|\mathbf{X}) = f(\mathbf{X}|\boldsymbol{\pi}, \boldsymbol{\theta}) \tag{5}$$

The general approach to constructing confidence intervals is to determine a corresponding hypothesis test. Here, the hypothesis test is $H_0$: $\boldsymbol{\pi} = \boldsymbol{\pi}_0$ versus $H_a$: $\boldsymbol{\pi} \neq \boldsymbol{\pi}_0$, and the test can be based on the likelihood ratio test statistic:

$$\lambda(\boldsymbol{\pi}_0|\mathbf{X}) = \frac{\sup\{L(\boldsymbol{\pi}_0, \boldsymbol{\theta}|\mathbf{X}); \boldsymbol{\theta}\}}{\sup\{L(\boldsymbol{\pi}, \boldsymbol{\theta}|\mathbf{X}); \boldsymbol{\pi}, \boldsymbol{\theta}\}} \tag{6}$$

A standard result in statistics is that $-2 \log \lambda$ converges in distribution to a $\chi^2$ distribution[89]. Therefore, we can determine the confidence level for the hypothesis $H_a$: $\pi \neq \pi_0$ by comparing the difference between $-2 \log \lambda$ across various intervals and its minimum value, and then referencing the corresponding $\chi^2$ confidence intervals.

The observed counts $X$ in this experiment follow a Poisson distribution: $X \sim \text{Pois}(\mu + b)$, where $\mu$ is the observed signal rate, and $b$ is the observed background rate. The observed background counts $Y$ are characterized by a Gaussian distribution: $Y \sim N(b, \sigma_b)$. Assuming $X$ and $Y$ to be independent, we have

$$f(x, y|\mu, b) = \frac{(\mu + b)^x}{x!} e^{-(\mu+b)} \cdot \frac{1}{\sqrt{2\pi}\sigma_b} \exp\left(-\frac{(y - b)^2}{2\sigma_b^2}\right) \tag{7}$$

Here, the hypothesis test is $H_0$: $\mu = 0$ versus Ha: $\mu \neq 0$. This model can be implemented using the model 5 in the TRolke library of CERN ROOT[90] to directly compute the profile likelihood function $-2 \log \lambda$, the results of which are shown in Extended Data Fig. 7a. The minimum of this likelihood function occurs at $\mu = 5.77$, and the corresponding $5\sigma$ value is greater than 0, providing significant evidence for the existence of the Migdal effect.

For the estimation of the upper and lower limits of the cross-section probability errors, the profile likelihood method is also used. It is important to note that the calculation of the cross-section probability must account for the signal efficiency. Therefore, the observed counts $X$ in the signal region are modified to $X \sim \text{Pois}(en + b)$, where $e$ is the signal

selection efficiency and $n$ is the number of Migdal events generated. The background counts $Y$ remain characterized by $Y \sim N(b, \sigma_b)$. Moreover, the signal efficiency is assumed to follow a Gaussian distribution: $Z \sim N(e, \sigma_e)$. The profile likelihood function for this model can also be directly computed using model3 in the TRolke library. The resulting likelihood function, along with the positions of the minimum $n_{\min}$ and its lower and upper limits $n_{\mathrm{ll}}$ and $n_{\mathrm{ul}}$, are shown in Extended Data Fig. 7b. The error propagation is applied as follows:

$$\Delta P_{\pm} = \sqrt{\left(\frac{n_{\min} - n_{\mathrm{ul/ll}}}{n_{\min}}\right)^2 + \left(\frac{n_{\mathrm{tot\ error}}^{\mathrm{NR}}}{n_{\mathrm{tot}}^{\mathrm{NR}}}\right)^2} \times \frac{n_{\min}}{n_{\mathrm{tot}}^{\mathrm{NR}}} \quad (8)$$

## Data availability

The Migdal beam experiment data are available at Science Data Bank (https://www.scidb.cn/s/JrqYjy). Other derived data, supporting the findings of this study, are available from the corresponding authors upon request.

## Code availability

The main codes supporting this study, including those related to simulation, event reconstruction, background calculation, YOLO and neutron energy spectrum monitoring, are available at https://gitee.com/marvelmigdal/migdal-related.git. The detector simulation software framework is available at GitHub (https://github.com/ElsevierSoftwareX/SOFTX-D-23-00679).

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

**Acknowledgements** This work is financially supported by the National Natural Science Foundation of China (grant nos. 12221005, 12575201, 12341502, 12235006, 12275134, 12335005, 12275232 and 2023YFA1606900), the Shandong Provincial Young Scientists Fund Project (class A) (grant no. ZR2025QA20), the Guangxi Key Research and Development Program (grant no. FN2504240030), the Guangxi Talent Program (Highland of Innovation Talents) and the Fundamental Research Funds for the Central Universities. L.W. acknowledges support from the State Key Laboratory of Dark Matter Physics for providing excellent research conditions and facilities.

**Author contributions** Q.L. and D.Y. conceived of the idea and designed the experiments. Y. Zheng supervised the project. R.M. developed the simulation software. S.C., Y.Y., J.Z. and W.H. performed the experiments with help from H.F., S.Z. and Z.Z.; L.K., C.S. and S.C. measured the neutron and gamma background. X.J., Y.Y. and J.Z. analysed the data. D.Y., H.F., H.L. and E.L. designed and constructed the detector. C.D., C.G., X.S., D.W., S.Z. and Z.Z. designed the Topmetal chips and readout electronics. Y. Zhang, J.W., Z.Y. and Z.W. designed and operated the D–D neutron source. P.L. provisioned the computing resources. L.S., J.S., B.Z. and L.W. performed the theoretical calculations. J.L. provided expert recommendations and conducted the experiment. Q.L. and D.Y. wrote and revised the paper. All authors provided input and comments on the paper.

**Competing interests** The authors declare no competing interests.

**Additional information**
**Correspondence and requests for materials** should be addressed to Qian Liu, Hongbang Liu, Xiangming Sun, Lei Wu or Yangheng Zheng.

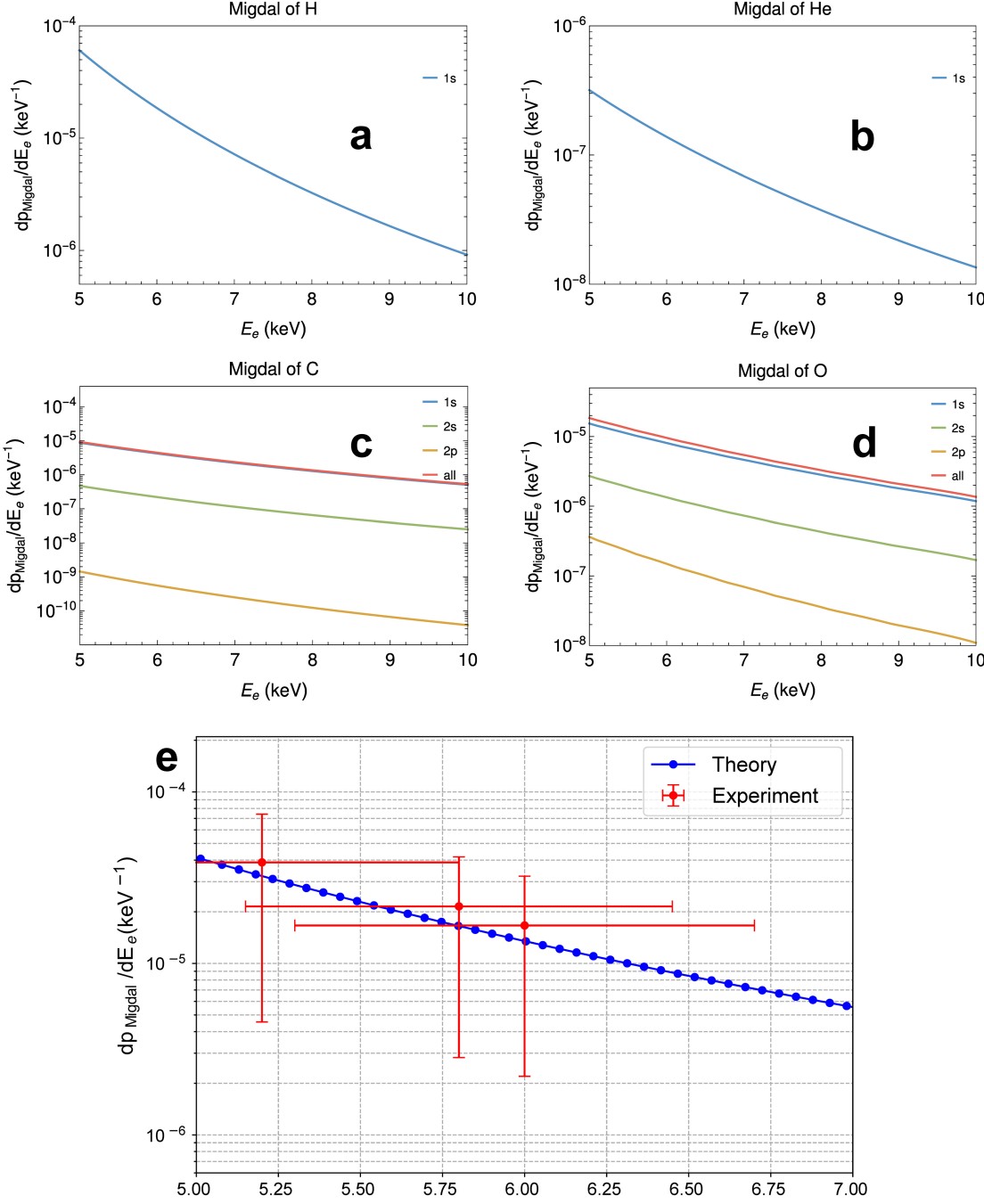

**Extended Data Fig. 1 | Migdal differential probabilities. a,b,c,d**, show theoretical calculation results of the differential probabilities of Migdal electron emission. **e**, compares the theoretical distribution of Migdal differential probabilities with experimental results in the gas mixture. The experimental data points are obtained by binning the electron energy from Migdal candidate events with a width of 0.2 keV, applying detection efficiency correction to each energy bin using simulation results, and normalizing based on the measured total cross-section ratio in the 5–10 keV.

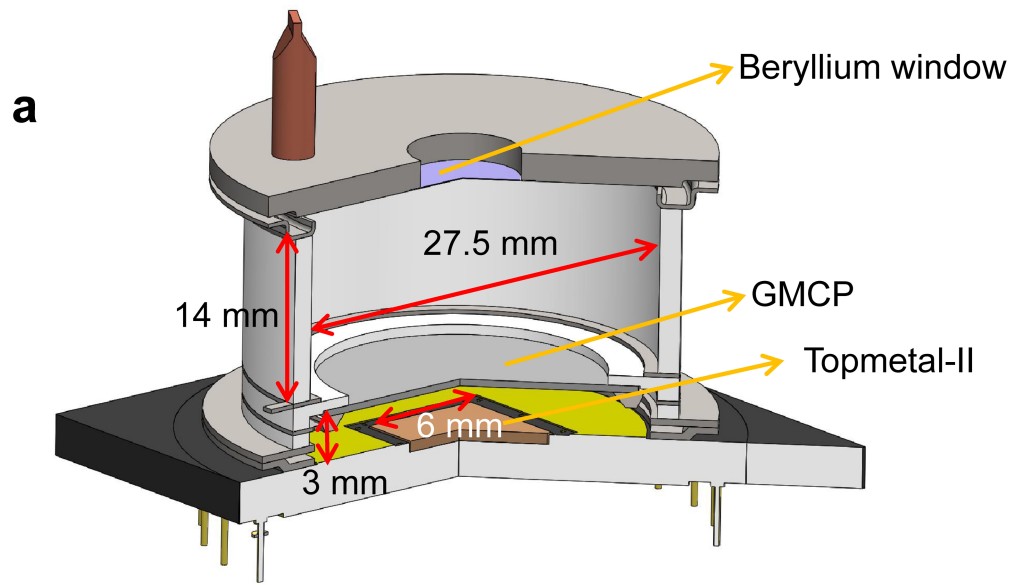

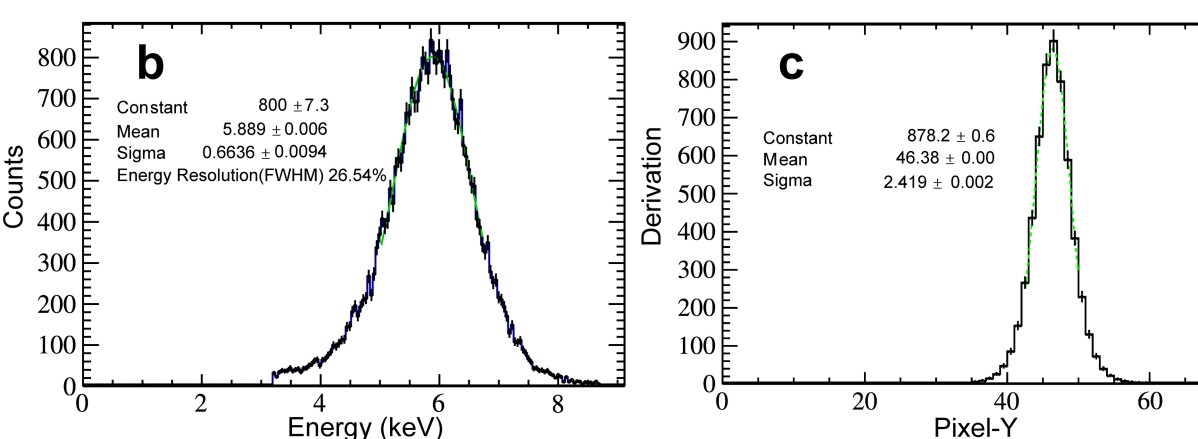

**Extended Data Fig. 2 | Schematic diagram of the detector structure and its calibration. a**, the height of the detector drift region is 14 mm with an electric field strength of 1.1 kV/cm. The thickness of the GMCP is 300 µm with 35 kV/cm electric field. The height of the induction region is 3 mm with a field of 2.4 kV/cm. The dimensions of the Topmetal-II are 6 × 6 mm. The mixed gas is filled at a pressure of 0.8 atm under room temperature (23 °C) conditions. **b**, calibration results of energy resolution @ 5.9 keV [55]Fe. **c**, calibration results of position resolution for 6.40 keV photoelectrons.

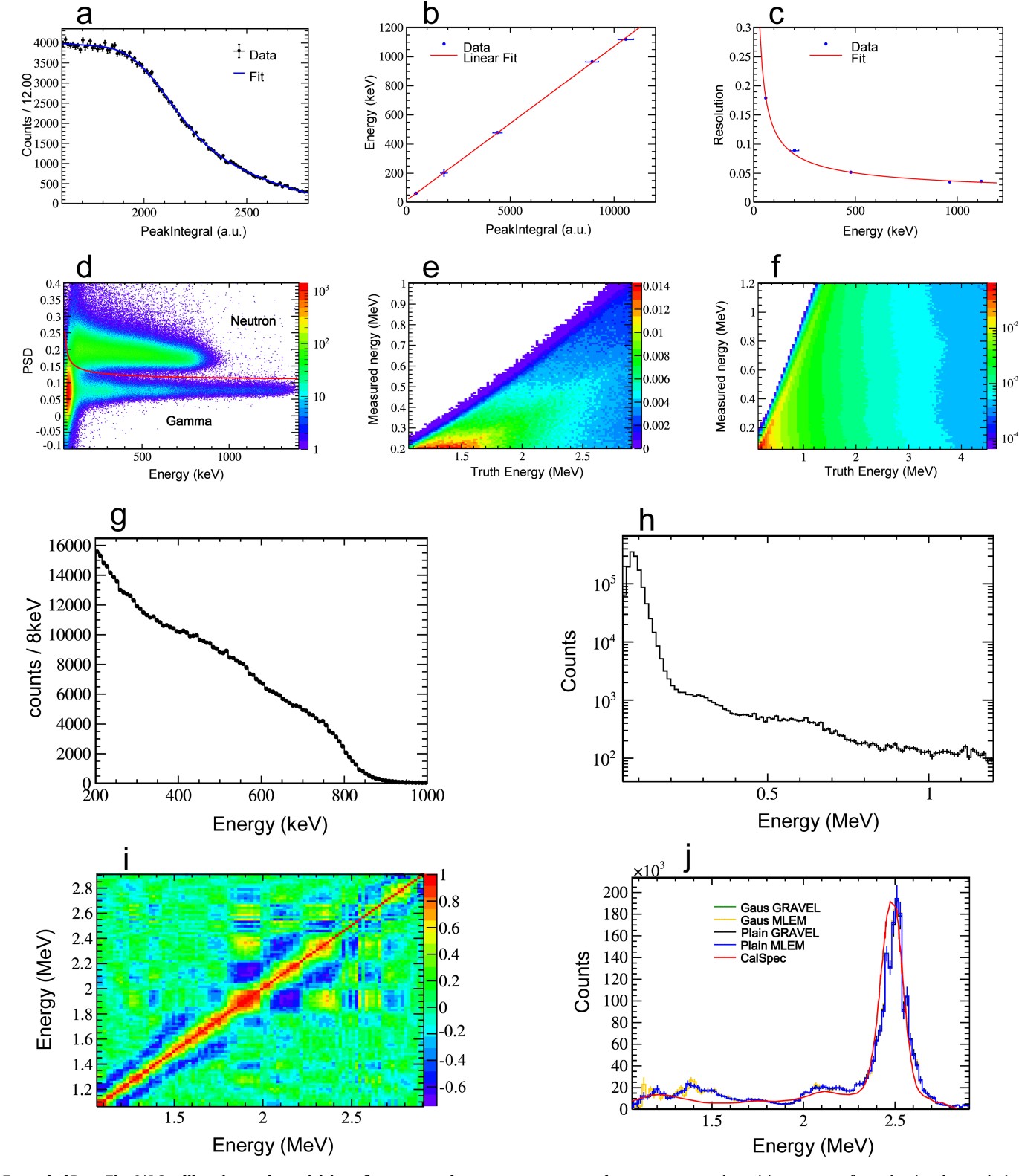

**Extended Data Fig. 3 | LS calibration and acquisition of neutron and environmental gamma energy spectra. a**, Compton edge fitting of [133]Ba. **b**, energy linearity fitting to EJ309 LS detector. **c**, Energy resolution fitting to EJ309 LS detector. **d**, neutron and gamma selection based on PSD capability of EJ309 detector. **e**, neutron, **f**, gamma response matrix of the EJ309 detector.

**g**, neutron, **h**, gamma energy deposition spectra after selection. **i**, correlation matrix of the unfolded spectrum with Plain guess and GRAVEL algorithm. **j**, the unfolded neutron spectra obtained with an EJ309 detector positioned at approximately 87°.

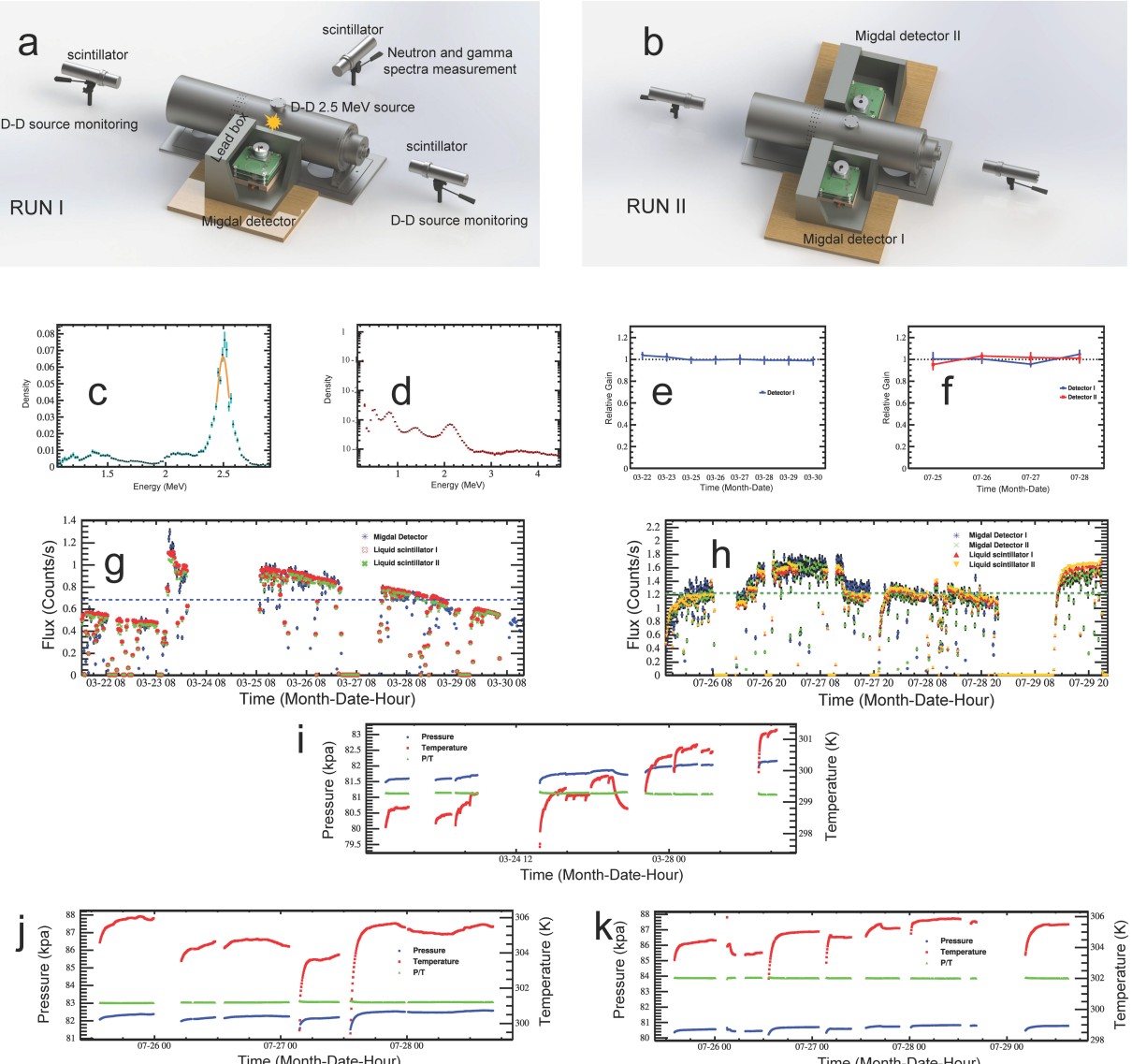

**Extended Data Fig. 4 | Experiment details. a**, **b**, rendered view of the experiment. **c**, neutron spectrum. **d**, gamma spectrum. Several prominent peak structures are observed in the gamma spectrum: the peak at 0.5 MeV originates from electron positron annihilation, the peak at 0.8 MeV is speculated to arise from the emission line of $^{56}Fe$ in the neutron generator casing, the peak at 1.4 MeV is hypothesized to result from activated $^{40}K$, and the peak at 2.2 MeV is attributed to the neutron capture process by hydrogen in polyethylene. **e**, **f**, the calibration result of the detector gain during each day of the experiments. **g**, **h**, the variations in the counting rate of the Migdal detector in two runs, as well as the neutron beam monitoring status by LSs. The neutron counts from LSs are normalized to the counting rate of the Migdal detector, with the dashed line representing the average counting rate. **i**, Run I detector I, **j**, Run II detector I, **k**, Run II detector II temperature (*T*) and pressure (*P*) within the chamber during the operation of the Migdal detector. *P/T* reflects the stability of the gaseous environment throughout the detector's operational period.

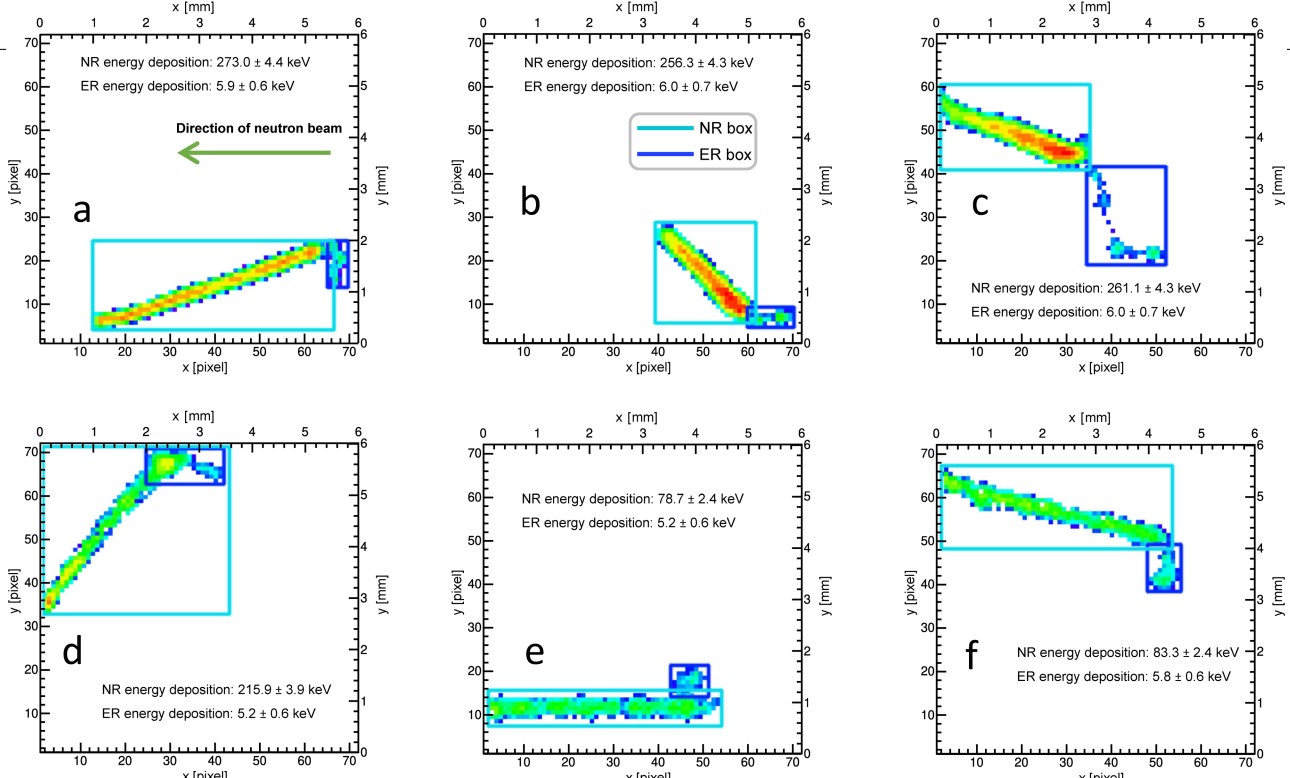

**Extended Data Fig. 5 | Selected candidate Migdal events. a**, **b**, and **c** are from Run I detector I; **d** and **e** are from Run II detector I; and **f** is from Run II detector II. The dark blue and light blue boxes are respectively the NR and ER regions detected by YOLO.

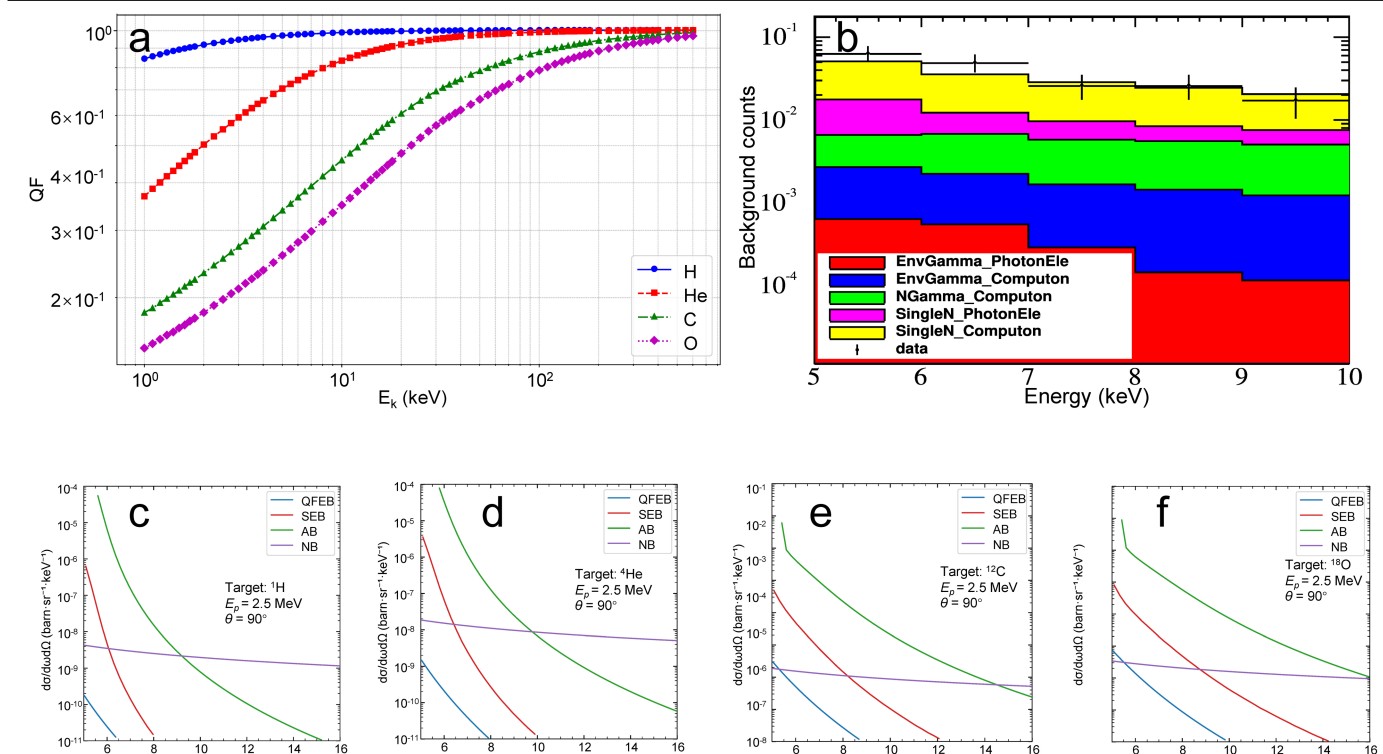

**Extended Data Fig. 6 | Quenching factor and background distribution.**
**a**, The relationship between the QF of different nuclei in the working gas and their kinetic energy. **b**, comparison of measured and simulated background electron spectra. **c**, **d**, **e**, **f**, the differential cross section of QFEB (blue lines),

SEB (red lines), AB (green lines), and NB (purple lines) induced by 2.5 MeV proton for different targets: $^1$H, $^4$He, $^{12}$C, and $^{18}$O. Meanwhile, we set $\theta = 90°$, where $\theta$ is the angle of the emitted photon with respect to the proton.

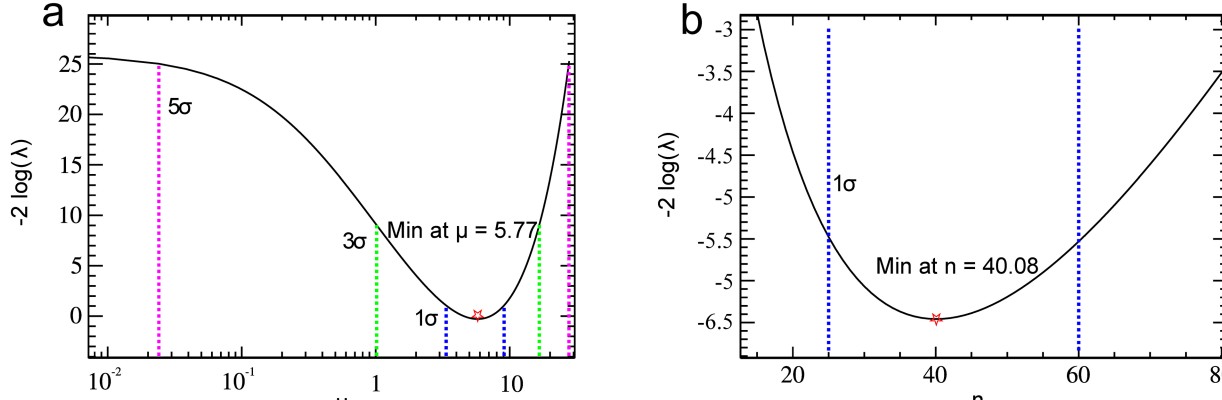

**Extended Data Fig. 7 | Significance and cross-sectional ratio profile likelihood. a**, Likelihood function for the significance test of the Migdal effect. **b**, likelihood function for the number of Migdal effect events. The upper and lower limits corresponding to a standard deviation are $n_{ll}$ = 24.88, $n_{ul}$ = 60.89, respectively.

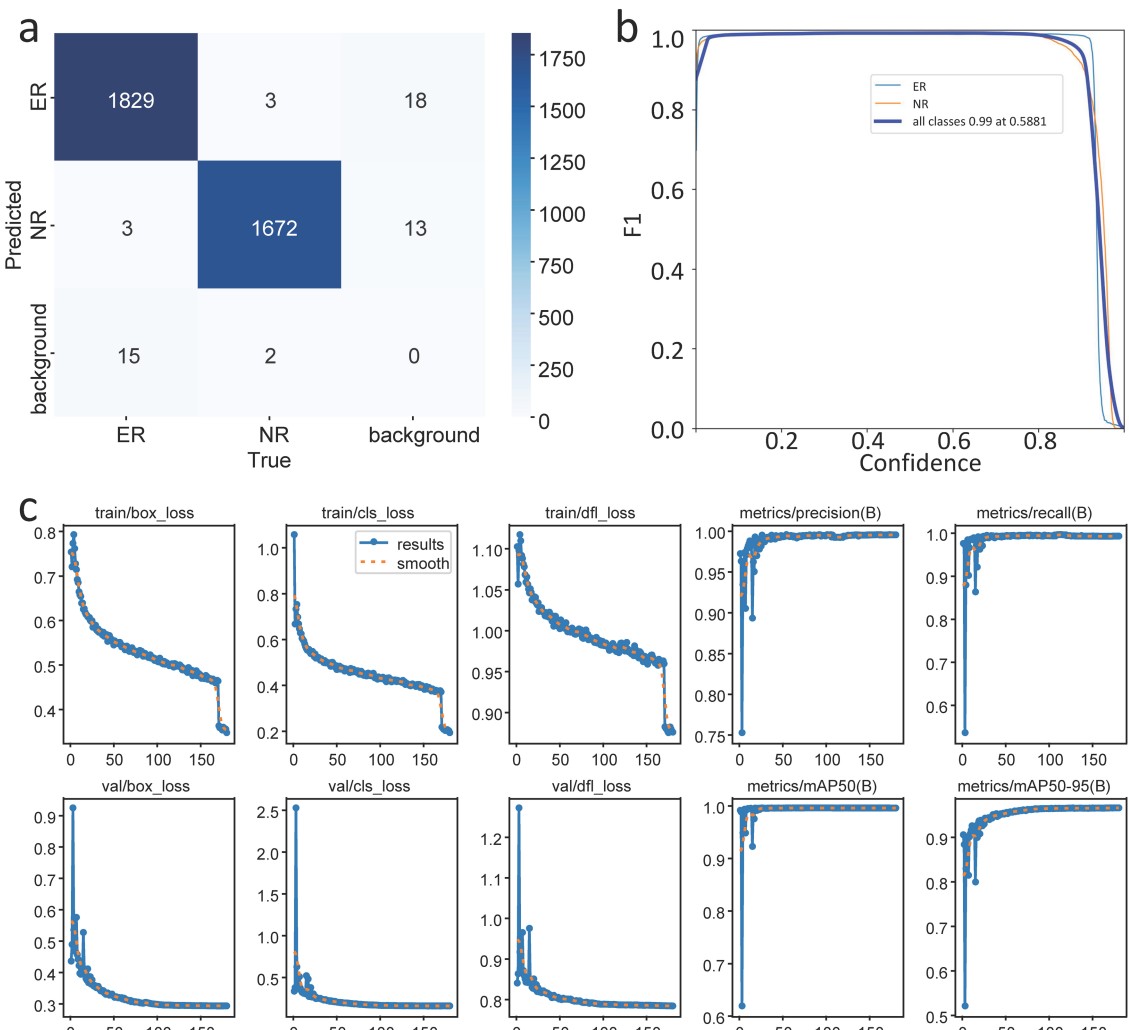

**Extended Data Fig. 8 | Training results of the YOLOv8 model. a**, The confusion matrix. **b**, F1-confidence curve. **c**, the loss functions converge with the increase of training epochs.

**Extended Data Table 1 | Expectation value is normalized to the number of experimental recoil nuclei**

| Background Component | Description | Expectation Value |
|---|---|---|
| Recoil induced δ ray | δ electron near NR track origin | $0.035 \pm 0.023(\text{stat.}) \pm 0.007(\text{sys.})$ |
| Particle Induced X-ray Emission | | |
| X-ray emission | Photoelectron near NR track origin | 0 |
| Auger electrons | Auger electron near NR track origin | 0 |
| Bremsstrahlung processes | | |
| Quasi-Free Electron (QFEB) | Photoelectron near NR track origin | $\approx 0$ |
| Secondary Electron (SEB) | Photoelectron near NR track origin | $\approx 0$ |
| Atomic (AB) | Photoelectron near NR track origin | $\approx 0$ |
| Nuclear (NB) | Photoelectron near NR track origin | $\approx 0$ |
| Random track coincidences | Photo-/Compton electron near NR track | $0.180 \pm 0.022(\text{stat.}) \pm 0.042(\text{sys.})$ |
| Muon induced δ ray | δ electron near NR track origin | 0.013 |
| Gas radioactivity | | |
| Trace contaminants | Electron from decay near NR track origin | $0.001 \pm 0.001(\text{sys.})$ |
| Neutron activation | Electron from decay near NR track origin | $\approx 0$ |
| Secondary nuclear recoil fork | NR track fork near track origin | $\approx 0$ |
| Total background | | $0.229 \pm 0.032(\text{stat.}) \pm 0.043(\text{sys.})$ |

An entry of "0" indicates that the process cannot occur, while "≈ 0" denotes that the event occurrence rate is <0.001.