## [Peer Review File · Nature]

Direct observation of the Migdal effect induced by neutron bombardment

Corresponding Author: Mr Difan Yi

Version 2:

Reviewer comments:

Referee #1

(Remarks to the Author)

The Migdal effect (or more precisely the Landau–Pomeranchuk–Migdal effect or LPM effect) is by itself very interesting. The LPM effect has become very important in recent years in the context of light dark matter detection. It would be good to mention in the paper that the LPM effect has already been observed indirectly with good precision at SLAC in 1994 (see P.L. Anthony et al., Phys. Rev. Lett. 75, 1949-1952 (1995)). The authors of the paper under review emphasize correctly that a direct observation with neutral particles is missing. This is indeed important and it is claimed that this has been achieved. This would be an important milestone reassuring that the mechanism works as expected and that light dark matter limits can reliably be obtained using the LPM effect.

The reported ratio of the LPM cross section to the nuclear recoil cross section is for the used recoil energies in the range expected from theoretical and phenomenological calculations. The experimental setup is in principle also very well suited for such a measurement. The claimed significance rests on 6 events selected out of almost 10^6 recorded events. This requires a very careful understanding of the process and of all sort of backgrounds which may contribute. The main paper is rather short on this, but the supplementary methods contain more of the important details which leads to a number of questions:

- A number of important numbers should be mentioned: Most importantly the total target mass and detector dimensions, the intensity of the neutron generator, strength of the drift field and more detailed the trigger conditions of the DAQ.
- The result is based on 6 events emerging from a huge set of recorded events which is reduced by various cuts and procedures. Various details how events are selected remain, however, at least partly hidden in the applied machine learning techniques. It is therefore unclear how much the final event number depends on sort of hidden analysis choices. In other words: How much do the 6 events depend on implicit selection biases and on the fact that the analysis was non-blind?
- Circularity is systematically bigger for shorter tracks / lower energies. It would be useful to show a plot displaying the circularity against the energy for all data.
- The target consists of gas molecules and it is unclear if/how this affects the likelihood of an electron transitioning following a nuclear recoil.
- Neutrons may produce a nuclear recoil combined with an interaction with electrons via a magnetic dipole interaction. How big is this component?
- It is claimed that ^{222}Rn can be ignored since the used gas has been stored sealed for over a year. ^{222}Rn can, however, only be ignored after it has been demonstrated in addition that emanation from vessel materials can be ignored. Can this be shown?
- 6 events in 150 hours corresponds to an activity of 11 micro-Bq for the whole detector. This is extremely small compared to many natural activities of all sort of tiny traces of nuclear backgrounds in the environment. An example is ^{85}Kr with a half-life of 10.7 years and where activities in air are typically a few Bq/m³. Can traces of such isotopes be safely excluded?

The manuscript is in summary definitively very interesting and important. Big claims require, however, always very solid evidence. The article should be published if the remaining questions can be sufficiently resolved.

Referee #2

(Remarks to the Author)
Dear Authors,

Please find attached below the detailed review of the manuscript in PDF format.

Best regards, the reviewers

Referee #3

(Remarks to the Author)
Review report
Manuscript "Accessing to the light dark matter: first direct observation of Migdal effect"

The undertaken research should be appreciated – existence of theoretically predicted, so called Migdal effect in nuclei scattered by neutral particles is a long-lasting experimental problem demanding its confirmation. It became especially important after the work of Ibe et al. (J. High Energy Phys., 03: 194, 2018), where it was explicitly calculated that electrons released via the Migdal effect can enhance the detectability of dark matter in some kinematic window.

In the detection of rare events of a tiny energy release, hidden in the significant background and identified i.a. by a reconstruction of specific topology, correctness of deposited energy, timing coincidence, the detection system needs a special care. Its performance must be excellent and precisely understood. This demands the energy and position resolution studies, development of precise event selection and verification methods, reliable and careful simulations of the signal and background, conservative estimation of uncertainties.

A few groups have undertaken so far projects oriented for Migdal effect confirmation. They started careful work, including calculations, simulations, dedicated detectors construction, performance tests, detailed data analysis, background consideration, kinematic checks. None of the groups working on Migdal effect problem have achieved a positive result so far.

In the provided manuscript "the first direct evidence of the Migdal effect in neutron nucleus scattering" is announced. In case of an article announcing the registration of a new effect, it should be written with particular care and all aspects of the experiment and data analysis should be thoroughly documented.

Unfortunately, in opinion of reviewer, the submitted manuscript does not meet these requirements. And presented there experimental result is not unambiguously proven.

In opinion of reviewer the overall quality of the manuscript is poor. For this reason the detailed review of the manuscript would be premature. Instead the general deficiencies of this work will be presented. With abbreviation "ms" the main manuscript is recalled whereas the "meth" refers to the additional information contained in "Method".

1. Detection setup

The whole experimental system used for presented examination is not sufficiently described. Fig. 1 of "ms" is by far not sufficient. Drawings of the whole system supplemented with information about crucial parameters (dimension, position, material) is needed. From "ms" one can learn only that lead shielding was used and some liquid scintillators were positioned (where ?) to monitor the neutron flux and spectrum. Neutron flux is unknown as well as a beam profile. Exact geometry and simulations of shielding effectiveness against various possible background components are not shown.

In the lack of solid information about neutron transportation and the shape, thickness and effectiveness of the shielding the reader may have doubt if the suppression of the ambient and neutron-generator related background is sufficient to conclude about discovery of a very rare and tiny nuclear effect.

A weak point of detection system is lack of 3-D track reconstruction ability. This limits the event selection methods based on the kinematic cuts.

2. Performance of detector (Gas Microchannel plate Pixel Detector)

Important information about detector are not provided neither in "ms" nor in ("meth"). From provided references (btw: the crucial in this respect ref. no 7 in "meth" (Nucl. Sci. Tech. 35, 39 (2024)) is provided with mistake) one can learn that gas microchannel plate-pixel detector which is used for Migdal effect detection has been developed and optimized for X-ray polarimetry at the space station. There is no doubt that its design is ingenious and advanced. Optimal working conditions has been fixed for a gas mixture of 50%Ne + 50%DME at pressure of 1 atm, with the drift and induced electric fields of 2 kV/cm and 2.5 kV/cm, respectively. For such conditions the energy resolution of 45.42% (FWHM) for X-rays of 5.9 keV has been achieved. Position resolution of this detector is not specified.

In summary of ref. 7, their Authors convey that their detector has worse performance than other detector of this type known from literature and that its further optimization is demanded.

From "meth" ref. 8 (Nucl. Instr. and Meth. A 1055, 168499 (2023)) one can learn that similar detector has been tested with mixtures of Ar:CO₂ and Ne:CO₂ at various proportions. In this case the best achievable energy resolution was 23.36% (FWHM) with position resolution again not specified.

In description of front-end electronics of over mentioned detectors ("meth" ref 12: IEEE Trans. Nucl. Sci. 70 (7), 1507-1513 (2023)) it is written that this readout was tested when detector was filled with He:DME (40:60) mixture at a pressure of 0.8 atm. (DME is composed as: CH₃OCH₃). Provided there energy resolution is 23.3 % - 28.9 %, depending on energy of incident photon (there is no information if this resolution is measured as FWHM). Position resolution is again not specified.

Finally, in the present "ms" it is written that the used for Migdal effect gas mixture was He:DME (40:60) and in "meth" - that the energy resolution for 5.9 keV X-rays is 26.54%, whereas the position resolution in both directions is 0.2 mm. Gas pressure is not specified as well as any voltages applied or drift parameters (e.g. track ranges, diffusion). No reference to careful studies of detector performance at selected detector conditions are found. Instead, a short and not clear text about detector calibration is proposed in "meth".

All these facts create confusion and make the reading and understanding the quality of detector performance difficult. In this view the shown in fig. 2 a precise energy measured for NR as 273.0 +/- 4.4 keV is difficult to understand. How the energy calibration for this higher energy range have been done ? Where it is described ?

3. Simulations

According to "meth" the software framework Star-XP (ref. 13: SoftwareX 25, 101626 (2024)) have been used for simulate Migdal events. But this software is optimized for simulations of low energy X-ray interactions in gaseous medium. There is no information if this software have been verified for simulations of effects associated with neutron interaction in solid and gaseous component of detection setup.

It is also mentioned in "meth" that some simulation with the use of Geant4 and Trim have been performed. But it looks like that they were restricted only to interaction of neutrons with liquid scintillator and with detector gas medium.

Large drawback of this manuscript is a lack of any simulation results shown. The reader has no chance to develop his own opinion about their completeness and usefulness in respect to signal selection and background suppression.

4. Background

Very important discussion of background is presented in "meth". Many possible background sources are indicated and, to some extent, discussed. Authors' claims about negligibility of some background components e.g. Auger or bremsstrahlung what in view of low Z materials used for detection medium, and applied energy cut of 5 keV for ER is correct. Nevertheless, often Authors' conclusions about insignificance or low yields of individual background components are just given to belief and not well justified. Specially doubtful are estimations of random track coincidences and events related to neutron activation (see below).

Except of plain text no any graphical support is provided. Simulations of background, which were performed, are not well documented. And they seems to be incomplete.

What evidently is missing are the consideration of effects induced by neutrons traveling through solid material of the detector and other apparatus. From the brief description of the apparatus in "ms" and "meth" it is clear that neutrons enters the medium gas through the detector wall. Wall is made of ceramics an Kova alloy. What is exact composition of these materials ? It is imaginable that hadronic and electromagnetic processes in the detector wall induced by neutrons (and possibly gammas from the generator) create a broad spectrum of prompt gammas which, in following, can mimic the electron recoils (e.g. by Compton process). The dominant element of Kova is copper. Stable isotopes of Cu have not negligible cross section for radiative neutron capture for MeV-neutrons. Perhaps this component is insignificant but can not be disregarded when total background is estimated. Careful Geant4 simulations taking into account the composition of detector wall and geometry could verify significance of this source of background and estimate the amount of possible fake NR-ER coincidences induced by this background component.

5. Event topology

Authors claim to record 6 events identified as a Migdal effect. It is not large number. All these events could be shown, similar as the one in fig. 2, together with parameters of the vertices and all other convincing information. Authors include Fig. 3 in "ms", which is expected to prove that classified as Migdal ER are well distinguished from NR. This figure provoke simple questions:

- what is the meaning of given there dE/dx values. Is it mean value of dE/dx measured at some track fragments ? Or it is the total deposited energy divided by the whole path length E/X ?
- where at this plot are the points for NR associated with indicated "Migdal ER" ?

Convincing would be showing that events topology are consistent with a two body kinematics of 2.5 MeV neutron scattered at gas nuclei. Moreover, for known energies of NR their range in the selected gas mixture at given pressure and electric field can be calculated and used for event verification. The same applies to electrons in the selected energy range.

Unfortunately, the detector used in described experiment permits the track reconstruction only as a 2-D projection. Thus, the actual track length, recoil angles and real dE/dx distributions are not detectable. Nevertheless the kinematical consideration of neutron scattering on the gas constituents and expected energy and angular ranges of resulting tracks are needed. Even for 2-D projections the check of total deposited energy of recoils would strengthen the event verification.

6. Rare event search algorithm

Authors inform in "meth" that identification of ER and NR have been done with high accuracy using the YOLOv8 - a deep learning-based object detection algorithm. YOLO is a common tool used for broad spectrum of application, also in search for rare events. Description of implementation and usage of YOLO for their data analysis Authors present only in one, not very clear paragraph. For comparison: to show potential of YOLO algorithm for Migdal event recognition, Authors provide reference to the article of another experimental group which also search for Migdal effect, namely the Migdal Collaboration (ref. 22 in "meth": Preprint at <https://arxiv.org/abs/2406.07538> (2024)). The Migdal Collaboration, for showing performance of YOLO for their data analysis needed 24 pages of high quality scientific article. Thus, very concise information about YOLO implementation provided in "meth" is by far not convincing. And there is no reference to any paper/thesis/proceeding which would describe YOLO performance in context of reviewed here work.

7. Quality of presentation

Both in "ms" as well as in "meth" relevant drawings, plots, tables are missing. They could help to understand the article and increase the confidence of the reader about the methods and results. Instead, reader is confronted with a plain text which is often obscure. Important quantities and Author's conclusions are often provided there without sufficient reasoning or relevant referencing.

Due to the lack of important information it would be difficult for another group to conduct similar experiment, which could confirm announced here result.

Limited command of English does not allow the reviewer to assess the linguistic quality of the manuscript. However, in view of the basic deficiencies of the manuscript, this assessment is not of large importance.

Summarizing:

Authors announced registration of 6 events identified as Migdal effect in neutron-nucleus scattering after ~150 hours bombardment of He:DME (40:60) (at not specified pressure) with ~2.5 MeV neutrons from D-D generator. Flux and profile of neutron beam is not specified.

Authors' confidence about the actual Migdal effect origin for selected tracks of NR and ER comes only from vertices of those tracks. Within the declared position resolution and with applied (unclear) selection methods these vertices overlap.

More stringent verification of selected event is not done (or not properly described).

The detection system is not sufficiently described together with complete specification of used materials.

The actual performance of the detection apparatus is not sufficiently documented and proven.

It is not clear if relevant and reliable simulations were performed in the required scope.

It was not unambiguously proven that all possible background components are suppressed or identified and rejected.

Taking the above into account it has to be concluded that experimental result reported in the manuscript is not unambiguously proven. Namely, it can not be excluded that 6 events considered by Authors as Migdal effect (only one of them is shown in the manuscript) are of accidental or other than Migdal effect origin.

In view of the indicated serious deficiencies of reviewed manuscript its rejection is suggested.

Nevertheless Authors should be encourage to continue their efforts for confirmation of neutral particle induced Migdal effect. Based on experience they have collected so far, with refining of their detection and analysis methods they have a good chance to contribute to this difficult experimental field of nuclear physics.

Referee #4

(Remarks to the Author)

I co-reviewed this manuscript with one of the reviewers who provided the listed reports.

Version 3:

Reviewer comments:

Referee #2

(Remarks to the Author)

The authors have thoroughly addressed all the points raised during the review process. The revised manuscript has clearly benefited from these revisions, resulting in a significant improvement in the overall quality of the article. The experimental results are now presented with greater clarity and detail, which strengthens their impact.

A few minor revisions could still be considered in the manuscript; however, a revision of the Methods section—including a dedicated subsection on the statistical analysis—is highly recommended prior to publication. Relevant details and suggestions are provided below.

- The sentence “The lack of direct observation undermines the conclusions drawn from DM experiments that rely on the existence of the Migdal effect, raising significant doubts about the reliability of their findings.” should be revised. Results from DM experiments so far cannot be based directly on the Migdal effect, as its contribution to the DM signal has not been precisely established, but only estimated through model-dependent assumptions. Therefore, I suggest softening this statement by removing the final part: “raising significant doubts about the reliability of their findings.”
- The authors have corrected the variables in Eq. (1); however, not all variables appearing in the formula are defined in the text. In particular, the variable n_{bg} and n_{obs}^{ER} described and should be explicitly defined.
- The suggestion to avoid reporting more than two digits in uncertainty values has been considered. However, the number of significant digits in the reported value must match the precision of its associated uncertainty. For instance, if the uncertainty has two digits at the 10^{-6} level, the central value must reflect that precision. As an example, the ratio of the Migdal cross-section to the nuclear recoil cross-section should be written as $[(4.9)_{-1.9}^{+2.6}] \times 10^{-5}$ instead of $(4.91)_{-1.9}^{+2.6} \times 10^{-5}$. This correction should be applied consistently throughout the manuscript, including in the Abstract, Ratio, Conclusion, and Table 2 of the Methods section. Another example of inconsistency is $0.035 \pm 0.023(\text{stat.}) \pm 0.0068(\text{sys.})$ should be $\pm 0.007(\text{sys.})$.
- The authors should include a dedicated section within the Methods specifically focused on the data analysis. This section must clearly describe the statistical methodology adopted and should also include the explanation—currently only provided in the response to reviewers—regarding the origin of the asymmetric error in the cross-section ratio. In particular, the authors are expected to detail the statistical approach used, the assumed distributions for signal and background events, and the plot of the resulting maximum likelihood profile. Providing these elements in the manuscript is essential to support the robustness of the analysis and to justify the claimed 5σ significance level of the experimental evidence.
- The extended theoretical discussion included in the review response would be valuable to readers if integrated—more concisely—into the Methods section. Figure 7 could also be added there to improve the reader’s understanding of the Migdal effect modelling and its consistency with the reported results.

Thank you for considering the feedback.

Referee #3

(Remarks to the Author)

Review report 2

First revision of manuscript

"Accessing to the light dark matter: first direct observation of Migdal effect"

Below I will refer to my doubts expressed in my first report and comment about Authors' explanations provided in the second submission ("ms" – manuscript, "meth" – Methods, "reb" - rebuttal).

1. Detection setup

In “meth” Authors provided a very schematic plot of a setup. One can learn that “Migdal detector” was kept at detection angle of 270 deg and beam parameters (rates, spectra) of neutrons and gammas were monitored with scintillators located at angles of 0, 87, and 180 deg. The lead shielding surrounding the detector was of the thickness of 1 cm (new information given in “ms”). Convincing information about neutron and gamma energy spectra are given in “meth”. Monitoring plots of the gas pressure, its temperature and counting rates are presented.

It became clear that no any beam collimation was used during experimental runs. The neutrons and gammas from generator could interact freely with all components of detection system. This is a drawback of the system, since some hadronic and electromagnetic processes undergoing in construction materials of the detector remain uncontrolled.

The lead shielding of 1 cm thickness is in 100% efficient to suppress the gammas of energies below ~250 keV. This is sufficient to stop the most dominant part of measured gamma spectrum shown in fig. 9 b of “meth”. Gammas of larger energies can penetrate into Migdal detector. For example: for 2 MeV gammas attenuation in 1 cm lead plate is only 33%. Such photons can induce fake electron tracks by, predominantly, the Compton scattering.

Estimation of the rate of indicated above processes and their contribution to possible occasional NR – ER coincidences can be done with the use of careful simulations with the use of Geant4. Both “ms” and “meth” still missing explicitly provided information of such performed simulations. Fortunately, in the “reb” Authors describe convincingly their simulations strategy and the results (“Comprehensive Detector Simulation”). With no doubt, in order to avoid unnecessary confusion, the information about performed Geant4 simulations of the whole detection setup, their results and agreement with experimental results should be included to “meth”. Thus, I propose that information contained in “3. Comprehensive Detector Simulation” of “reb” in compressed form are given in “meth”, for the knowledge of common readers.

Authors provided also new information that actually two Migdal detectors were used. Presumably they both had the same construction and performance quality. Would be interesting to know where the second detector have been installed and which of six Migdal event candidates were recorded in the second detector.

2. Performance of detector

Very useful is new fig. 5 of “meth” where actual dimensions and electrical parameters of the “Migdal detector are shown” (for completeness, the internal diameter of the detector should be given). It clarifies that the fiducial volume of the detector is narrow (14 mm).

Authors provided also explicitly the demanded information about working parameters, their stability, performed position calibration and resolution (mainly in “reb”) and energy calibration and resolution. I found also the information about extrapolation of energy resolution. I would like to suggest to give this information already in chapter “Detector calibration” of “meth” and not only when cross section is discussed. Technique of measurement of position resolution Authors explained in “reb” and in referred there a work JINST 19, P04039 (2024). This knowledge however, will not be easily accessible for the common readers. Thus, I suggest to include a reference to JINST 19, P04039 (2024) also in “meth”.

3. Simulations

In the “meth” Authors underline the usage of Star-XP model as a most general tool for relevant simulations. There is no doubt about the advantages of Star-XP model to simulate processes taking place in the gas detector medium and signal generation in the electronics. My concerns were about neutron and gamma interaction in the construction material. But I found no evidence that in Star-XP, designed for specialized application (low energy X-ray interactions in gaseous medium), the neutron interaction are implemented. The referred paper about Star-XP (SoftwareX 25, 101626 (2024)) is silent on this topic. In the current version of reviewed work there is also still not a clear information if neutron transport via whole detection system were considered.

As noted above, in “reb” Authors convincingly described their simulation of the detection system model with Geant4, which is, in my opinion, the most relevant for such application. As I have written in point 1 the “meth” should be supplemented with the information about Geant4 simulations of the whole detection system and their results.

4. Background

After information in “reb” (also those addressed to other Referees) about performed simulations and their validation, in my opinion the background estimation is correct. When chapter “Simulation” of “meth” will be supplemented with a message about performed comprehensive Genat4 simulations the current content of the “meth” part devoted to the background would be, in my opinion, sufficient.

In section “Random track coincidences” of ‘meth” misleading is the sentence: “Through a selection algorithm, the selection efficiency of the accidental coincidences is obtained to be 0.29%.” I suppose that the intention of the Authors is to say that “Through a selection algorithm the accidental coincidences were identified as 0.29% of all cases”.

5. Event topology

Explicitly shown dimensions of the working volume of the cell (fig. 5 in “meth”) soften may previous concern about lack of 3D event topology reconstruction. Narrowness of the cell reduces the possibility of random vertex overlap of NR and ER due to

2D projection. Nevertheless, I can not agree with the Authors' statement, that "3D track reconstruction is not essential for the discovery of the Migdal effect" ("reb"). In my opinion, for announcing the first evidence of Migdal effect the clear 3D event topology fulfilling the kinematics restriction would be the most convincing. However, I agree, that precise reconstruction of a common vertex of correctly identified NR and ER tracks in the narrow cell, within a reasonable time window, at low event rate and with large signal/background ratio is a strong argument in favor of searched Migdal effect.

With the 2D projection of the tracks some amount of signal events can be not recognized due to ER and NR track overlap. This fact may influence the estimation of cross section for Migdal effect. In the current work Authors consider and estimate this uncertainty by means of simulation. This is another motive for extension of the "meth" with description of Authors' scheme of reasoning based on performed simulations.

Fig 3 of "ms" is now much more convincing than the previous one. Still explanations of colored contours are missing.

As requested, the topology and tracks parameters of all 6 event candidates are shown now in fig. 12 of "meth". Would be also interesting to know in which detector they were recorded. The ionization densities along the NR track are different for event a, b, c than for event d, e, f. Could it be concisely explained ?

6. Rare event search algorithm

Explanations given by Authors in "reb" about event search algorithm are in my opinion convincing. For the credibility of their paper these information have to be made public. Due to the lack of previous publications by the authors on this topic, which could be cited, the "meth" again seems to be the place where a more extensive description of signal selection needs to be given. For this aim the relevant content of "reb", but in the shortened form can be used.

Except of, in my opinion, important extension of the "meth" content, which I outlined above I suggest the following small corrections of the current version of "ms":

line 61 "background fluctuations" → "rate of background reactions"

line 84 "nucleons, while" → "nucleons. While"

line 86 "atoms, and the resulting" → "atoms. The resulting"

line 103 "T cluster segmentation algorithm" - some reference is needed

line 119 - 121

"In Figure 3, the distributions of electrons and recoil nuclei for the 6 Migdal events are presented, with the ERs and NRs separated into two clearly distinguishable regions in the plot."

→

In Figure 3 the simulated distributions of electrons and experimental distribution of recoil nuclei are presented. The ERs and NRs are separated into two clearly distinguishable regions. The measured ER and NR of 6 Migdal events are superimposed".

Fig. 3. The colors must be explained.

Line 156 -157

there is inconsistency in labeling of n_{obs} , n^{bg} within the formula (1) and in line 157

line 158 "and total NR remain" → "and total NR, respectively" ???

line 180 is "underlying" an appropriate word here ?

Moreover in "meth":

the graphical features (labels, fonts, numbers, legends, ...) of all figures have to be unified and improved. Their captions have to be completed.

Fig. 4. Without further explanations this fig. is not understandable: experimental points

Conclusions:

After explanations and additional information provided by Authors in current version of "ms" and "meth", but especially in "reb", and addressed to me and other Referees my confidence in the credibility and value of the results presented in the reviewed paper has significantly increased.

Authors convinced me that they performed difficult experiment and conducted extensive and critical analysis of collected data. In effect they gathered the material that with significant confidence shows experimental evidence of Migdal effect induced by neutral projectiles.

Such result is worth to be published in prestigious scientific journal as Nature. However, for this aim, their paper must be extended and written in the way that all important information are provided explicitly in “ms” and “meth” (or in references), avoiding doubts and confusion.

Thus, I suggest another revision of provided “ms” and “meth” in order to supplement the paper with indicated above missing information.

Referee #4

(Remarks to the Author)

I co-reviewed this manuscript with one of the reviewers who provided the listed reports.

Version 4:

Reviewer comments:

Referee #2

(Remarks to the Author)

The authors have revised the article, the Method section and the Supplementary material, providing the more complete information requested by the reviewers.

The manuscript is now of very high quality and the results are presented in a clear and detailed manner. I therefore recommend the article for publication.

Only one requested change has not been implemented, despite the authors' statement in their response.

This can be easily corrected in the final version, but in the reviewers' opinion it does not prevent the article from being suitable for publication. The minor change is reported again below:

- The sentence “The lack of direct observation undermines the conclusions drawn from DM experiments that rely on the existence of the Migdal effect, raising significant doubts about the reliability of their findings.” should be revised. Results from DM experiments so far cannot be based directly on the Migdal effect, as its contribution to the DM signal has not been precisely established, but only estimated through model-dependent assumptions. Therefore, I suggest softening this statement by removing the final part: “raising significant doubts about the reliability of their findings.”

Best regards

Referee #3

(Remarks to the Author)

Review report 3

Second revision of manuscript

"Accessing to the light dark matter: first direct observation of Migdal effect"

Provided manuscript together with information contained in “Method” and “Supplementary material” describes the very difficult experiment and its careful and exhaustive analysis. High performance of the experimental apparatus has been confirmed with the extensive and pertinent tests. Interpretation of results has been supported by scrupulous multifold simulations and statistical consideration.

In my opinion resulting discovery of “first direct observation of Migdal effect” induced by neutron bombardment is now well proven. Thus, submitted paper can be accepted for publication in “Nature”.

I would like to congratulate the Authors on their hard but successful work and impressive result.

The text should be still polished. In few cases the sentences are too long. Other defects can also be noticed. Several of them are listed below. I hope corrections can be done during proofreading.

(“ms” – manuscript, “meth” – Method, “sup” - Supplementary material).

In “ms”:

Fig. 3. In caption add the sentence: Superimposed are experimental points for ER and NR of 6 Migdal event candidates.

In "meth"

page 1: The term $p_{ui}(n_k \rightarrow E_e)$ is ... → The term $p_{ui}(n_k \rightarrow E_e)$ is ...

page 1: To compared with ... → To compare with ...

page 2:

$E_{max} = 10$ keV E_{nth} and E_{eth} are the thresholds of detector for nucleus and electron recoils, respectively.

→

$E_{max} = 10$ keV. E_{nth} and E_{eth} are the detection thresholds for nucleus and electron recoils, respectively.

page 2, caption of Fig. 4:

... by binning the electrons from Migdal candidate events ...

→

... by binning the electron energy of Migdal candidate events ...

page 3:

The transmission and power supply of the pixel chip information is connected to the external environment by ...

→

The power supply of the pixel chip and information transmission is realized by ...

page 3:

The basic performance testing of the detector is tested

→

The basic performance of the detector is tested

page 4: eMMC → embedded Multi-Media Card (eMMC)

page 12: The training dataset ... with gas. - This sentence is too long.

Page 14, caption of Fig. 12: and c → and (c)

Page 15, caption of Fig. 13: ... different nucleon components ... ??? (nucleus ? Element ?)

In "sup"

I. 4: soft limit (The electron ... → soft limit (the electron ...

I. 29: which applicable → which is applicable

I. 87: as shown in equation 7. → as shown in equation 8.

I. 114: outlined in a., → outlined in b.,

I. 161: Once the photon energy exceeds → Once the photon energy (ω) exceeds

I. 204: coincidentally coincide → coincidentally overlap (??)

→ accidentally coincide (??)

I. 222: as above

I. 402: Table → Table 7.

Referee #4

(Remarks to the Author)

I co-reviewed this manuscript with one of the reviewers who provided the listed reports.

Dear Editor and Reviewers,

We would like to thank the editor for giving us the opportunity to revise our manuscript. We sincerely appreciate the meticulous review conducted by the reviewers on our submitted manuscript, along with their valuable insights and suggestions. The reviewer's expertise has played a crucial role in guiding us towards enhancing and refining the manuscript.

We have meticulously reviewed and addressed each of the reviewers' suggestions, and have made corresponding revisions and supplements to the manuscript based on their recommendations. Below are our responses, where the **reviewers' comments are presented in blue text**, and **our replies are provided in black text**.

Referee 1:

The Migdal effect (or more precisely the Landau–Pomeranchuk–Migdal effect or LPM effect) is by itself very interesting. The LPM effect has become very important in recent years in the context of light dark matter detection. It would be good to mention in the paper that the LPM effect has already been observed indirectly with good precision at SLAC in 1994 (see P.L. Anthony et al., Phys. Rev. Lett. 75, 1949-1952 (1995)). The authors of the paper under review emphasize correctly that a direct observation with neutral particles is missing. This is indeed important and it is claimed that this has been achieved. This would be an important milestone reassuring that the mechanism works as expected and that light dark matter limits can reliably be obtained using the LPM effect.

Re: Thanks for the comments. LPM effect refers to the cross section for bremsstrahlung from highly relativistic particles in dense media is suppressed due to interference caused by multiple scattering (see L. D. Landau and I. J. Pomeranchuk, Dokl. Akad. Nauk. SSSR 92, 535 (1953); A. B. Migdal, Phys. Rev. 103, 1811 (1956)). Such an important effect has been observed indirectly with good precision at SLAC in 1994 (see P.L. Anthony et al., Phys. Rev. Lett. 75, 1949-1952 (1995)). However, Migdal effect that we studied is different from LPM effect. It refers that an electron may be emitted from an atom after the sudden perturbation of the nucleus, which has been known since the early 1940s (A. Migdal, Zh. Eksp. Teor. Fiz. 9, 1163 (1939); A. Migdal, J. Phys. Acad. Sci. USSR 4, 449 (1941); E. L. Feinberg, J. Phys. Acad. Sci. USSR 4, 423 (1941)) and been named as the 'Migdal effect' within the DM community (Int. J. Mod. Phys. A22:3155-3168,2007)).

The reported ratio of the LPM cross section to the nuclear recoil cross section is for the used recoil energies in the range expected from theoretical and phenomenological calculations. The experimental setup is in principle also very well suited for such a measurement. The claimed significance rests on 6 events selected out of almost 10^6 recorded events. This requires a very careful understanding of the process and of all sort of backgrounds which may contribute. The main paper is rather short on this, but the supplementary methods contain more of the important details which leads to a number of questions:

A number of important numbers should be mentioned: Most importantly the total target mass and detector dimensions, the intensity of the neutron generator, strength of the drift field and more detailed the trigger conditions of the DAQ.

Re: We have revised our manuscript accordingly. Specifically, we have added a schematic diagram illustrating the detector dimensions in the Methods section (see Fig.4), as suggested. Additionally, we have included the monitoring results of the neutron generator during our experimental process.

The electronics system is functionally divided into three layers: the front-end electronics readout board, the back-end electronics board, and the high-voltage (HV) board. The front-end electronics board primarily includes the Gas Pixel Detector (GPD) unit and the GPD data readout circuit. The back-end electronics board comprises the main controller, data and firmware storage, communication interfaces and devices, and an external clock. The HV board consists of the GMCP bottom surface feedback circuit, voltage divider circuit, HV chip, and monitoring circuit.

Front-end Board: Designed to host the GPD, it is positioned at the top of the electronics system to facilitate track vibration measurements. The performance metrics of the Topmetal-II can be adjusted via external voltages; thus, the board employs a multi-channel digital-to-analog converter (DAC) to provide the required voltage configurations.

Back-end Electronics Board: Performs multi-channel signal processing. The FPGA is equipped with 1GB of flash memory, storing multiple files to ensure the system can boot correctly in the event of original file corruption. Backup configuration files are stored at specific addresses to mitigate the risk of system startup failures due to FPGA configuration errors.

High-voltage Board: Capable of generating a maximum negative high voltage of -4 kV. The GMCP bottom surface generates pulse signals upon electron arrival, which can serve as GPD trigger signals. These signals are processed through a comparator to enhance energy resolution.

Topmetal-II Data Processing: During detection, the output data from the Topmetal-II is quantized, encoded, compressed, and stored in real-time in the eMMC. The GMCP bottom surface signals, Coordinated Universal Time (UTC), system operational status, and monitoring data are stored in the eMMC with different headers. The chip's scanning frequency determines the detector's performance metrics, with pixel switching frequency required to be maintained at the MHz level. Consequently, a data compression scheme has been integrated into the data processing. The detector employs the DCM (Difference Compression Method) for data compression: the ADC value of each pixel per frame is stored and compared with the ADC value of the previous frame. If the difference exceeds a preset value, the pixel is identified as a signal pixel and transmitted to the erasure module, thereby achieving data volume compression.

The result is based on 6 events emerging from a huge set of recorded events which is reduced by various cuts and procedures. Various details how events are selected remain, however, at least partly hidden in the applied machine learning techniques. It is therefore unclear how much the final event number depends on sort of hidden analysis choices. In other words: How much do the 6 events depend on implicit selection biases and on the fact that the analysis was non-blind?

Re: We sincerely appreciate the reviewer's attention to the analysis algorithm, which is indeed an aspect we have considered with particular care. Regarding the machine learning component, we ensured that a significant proportion of the dataset used for training the model consisted of experimental data images. This approach was adopted to guarantee that the model effectively learns the characteristics of the experimental data. Additionally, we incorporated a portion of simulated data to enhance the model's adaptability and transferability. The distribution of our dataset is presented in the table 1 below:

Table 1: YOLO Training and Testing Dataset

Data set	Training	Validation
----------	----------	------------

Experiment	⁵⁵ Fe	3000	1493
	D-D	2994	1354
Simulation	ER	1200	301
	NR	1200	301
Total		8394	3449

The confusion matrix obtained from the trained model on the test set is shown in the figure 1 below:

Fig 1. The confusion matrix. The matrix element A_{ij} represents the probability that the j -th classification is predicted as the i -th classification.

The results demonstrate that this confusion matrix is highly diagonalized. Furthermore, since 82.5% of the validation data used to generate the confusion matrix is derived from the real experimental calibration dataset, we can reasonably conclude that the confusion matrix reflects the misidentification probabilities for the experimental data. Among these misidentifications, the misclassification of NR as ER has the most significant impact on our misjudgment of Migdal events. We have estimated this misidentification probability as follows: Based on the confusion matrix, the probability of misidentifying NR as ER is: 0.179%.

We employed a period of higher neutron beam intensity to estimate the probability of two NRs appearing simultaneously in a single frame. During this period, the event rate was approximately 1.2 count/s, and the frame refresh time was 2.6 ms. Using the Poisson

distribution, we calculated the probability of two NR events occurring in any given frame as follows:

$$\lambda = 1.2 \times 2.6 / 1000 = 3.12 \times 10^{-3},$$

$$P_{\text{Poisson}}(\lambda = 3.12 \times 10^{-3} | k = 2) = \frac{\lambda^k \times e^{-\lambda}}{k!} = 4.85 \times 10^{-6}.$$

In the experiment, there were 8.17×10^5 NR events corresponding to the same number of frames. Therefore, the expected number of instances where two NR events appear in the same frame is calculated as $N = 6.60 \times 10^{-6} \times 8.17 \times 10^5 = 3.96$. The expected number of instances where one of them is misidentified as ER is calculated as: $N_{\text{ER/NR}} = 2 \times 3.96 \times 0.179\% = 1.42 \times 10^{-2}$. Further considering the probability of two events sharing a common vertex, which is 0.29% (as detailed in the Methods section: Random Track Coincidences), the final contribution to the background due to machine learning misidentification is calculated as $N_{\text{ML_BG}} = 1.92 \times 10^{-2} \times 2.9 \times 10^{-3} = 4.11 \times 10^{-5}$. This result is significantly lower than the threshold of 6 events and also substantially smaller than the contributions from other major background sources.

In summary, regarding the role of machine learning in our analysis, its primary function is to identify whether a frame contains electron or nuclear recoil tracks, thereby filtering out a large number of invalid frames and events that only contain electrons or nuclear recoils. It does not directly participate in the identification of Migdal events. For the identification of Migdal events, we still rely on traditional algorithms, which screen candidates based on track topology, energy, dE/dX , and other physical characteristics.

In terms of constraining the model, on one hand, we provided the model with thousands of control sample images from the experiment to ensure it learns accurate information. On the other hand, the confusion matrix derived from a validation set with a significant proportion of experimental data directly reflects the model's strong ability to distinguish between NR and ER tracks in the experiment. The data set, labeled with experimental data, allows us to evaluate the performance of the machine learning model, understand its errors, and estimate their impact. This comprehensive approach ensures the reliability and robustness of our results.

Circularity is systematically bigger for shorter tracks / lower energies. It would be useful to show a plot displaying the circularity against the energy for all data.

Re: We express our sincere gratitude to the reviewer for the constructive suggestion. In accordance with the feedback, we have presented the two-dimensional distributions of circularity and dE/dX for both electrons and recoiling nuclei. The nuclear recoil data depicted in figure 2 is entirely derived from experimental measurements, while the electron data were generated using the Star-XP simulator, with the electron energy distribution in the simulation being sampled from the theoretically calculated Migdal electron probability cross-section. The orange contour represents the 90% distribution interval for electrons. As demonstrated, the parameters of circularity and dE/dX enable a clear and effective distinction between electrons and recoiling nuclei. In the main text, we restricted the X-axis range to 0-25 and included experimental ^{55}Fe data to facilitate a more precise identification of the characteristic features of Migdal electrons.

Fig 2. The two-dimensional distributions of circularity and dE/dX for electrons and recoiled nuclei. The cluster on the left represents the characteristic distribution of simulated 4 – 10 keV electrons, with the electron energy spectrum derived from the theoretically calculated Migdal electron energy spectrum. The cluster on the lower right depicts the nuclear recoil events from experimental data. The black dashed contour outlines the 95% distribution region for electrons, while the red dashed contour outlines the 95% distribution region for recoiled nuclei.

The target consists of gas molecules and it is unclear if/how this affects the likelihood of an electron transitioning following a nuclear recoil.

Re: For a low energy neutron beam (~ 100 eV), the Migdal transition of the electrons in molecules will be non-negligible [arXiv:2208.09002]. However, since we use a 2.5 MeV D-D neutron beam and require a ~ 100 keV nuclear - recoil (NR) threshold in our analysis, the chemical bonds of C-H and C-O (~ 10 eV) are easily broken at our recoil energies. Accordingly, it is reasonable to approximately treat the C, H, and O as the free atoms in our NR energy regime, and calculate the likelihood of electron from Migdal scattering as $P_{\text{DME}(\text{CH}_3\text{OCH}_3)} \approx 6P_{\text{H}} + 2P_{\text{C}} + P_{\text{O}}$, where P_{H} , P_{C} , and P_{O} are the individual Migdal transition probabilities for hydrogen, carbon, and oxygen, respectively, as shown in the following figure 3.

Fig.3 Migdal electron ionization rates of different elements and electron orbitals in DME.

Neutrons may produce a nuclear recoil combined with an interaction with electrons via a magnetic dipole interaction. How big is this component?

Re: First, the neutron scatters off electrons through its magnetic dipole moment, described by the Rosenbluth formula for the differential cross section:

$$\frac{d\sigma}{d\Omega} = \left(\frac{d\sigma}{d\Omega} \right)_{\text{Mott}} \left[\frac{G_E^2 + \tau G_M^2}{1 + \tau} + 2\tau G_M^2 \tan^2 \frac{\theta}{2} \right],$$

Where $G_M = F_1(q^2) + 2M_n F_2(q^2)$ governs the magnetic dipole interaction, with M_n

the neutron mass and $\tau = q^2 / (4M_n^2)$. For typical momentum transfers, the total cross section is approximately 1.8 μ b. Taking into account the probability of random coincidence of the electron recoil tracks from neutron scattering with other nuclear recoil tracks at a common vertex, even when compared to the Migdal cross-section, this value is already small enough to be negligible.

Second, we employed a data-driven approach to estimate the contributions of various possible electron production processes to the experimental background, which can be referenced in the "Method: Random track coincidences" section. In our analysis, we counted the total number of electrons appearing in the frames of 8.17×10^5 NR events. This electron count encompasses the sum of all potential physical processes that could produce electrons, including the process of electron recoil influenced by the neutron magnetic moment.

It is claimed that ^{222}Rn can be ignored since the used gas has been stored sealed for over a year. ^{222}Rn can, however, only be ignored after it has been demonstrated in addition that emanation from vessel materials can be ignored. Can this be shown?

Re: We thank the reviewer for raising this critical issue. Our background model explicitly incorporates ^{222}Rn and its decay chain. Key considerations are outlined below:

A. Primary ^{222}Rn exclusion: Environmental ^{222}Rn activity: 10 Bq/m³ (aligned with UNSCEAR 2000 indoor air reference: 5-50 Bq/m³) Detector insensitivity to ^{222}Rn α -decays: its α -decay nature (Q-value = 5.59 MeV) produces heavy recoiling nuclei and α particle that does not generate electron-like tracks in our detector.

B. Secondary β -background from $^{214}\text{Pb}/^{214}\text{Bi}$: Calculated via: Monte Carlo simulation of β -electron tracks in the detector geometry Normalization to our experimental duration. The resultant background contribution from ^{222}Rn progeny is 0.0002 ± 0.00004 events.

C. Tertiary ^{214}Po α -background exclusion: ^{214}Bi β -decay produces ^{214}Po (half-life = 163.6 μ s) within the data acquisition window ($\Delta t = 1 \text{ ms} \ll 5 \times \text{half-life}$). Potential co-vertex signature: β -electron (from ^{214}Bi) + α -particle (from ^{214}Po , $E_\alpha = 7.6 \text{ MeV}$). Energy deposition discrimination criteria: α -tracks deposit 7.6 MeV vs. signal region (NR tracks produced by

neutron scattering $E < 2.5$ MeV)

Zero event would pass the energy cut.

Events in 150 hours corresponds to an activity of 11 micro-Bq for the whole detector. This is extremely small compared to many natural activities of all sort of tiny traces of nuclear backgrounds in the environment. An example is ^{85}Kr with a half-life of 10.7 years and where activities in air are typically a few Bq/m^3 . Can traces of such isotopes be safely excluded?

Re: For trace radioactive background, we categorize the sources into two parts: one consists of isotopes of H, He, C, and O, while the other includes trace radioactive nuclides other than H, He, C, and O.

Among the isotopes of H, He, C, and O, ^3H and ^{14}C exhibit relatively high abundances, with values of 10^{-18} and 10^{-12} , and an uncertainty of 100%. The half-life of ^3H is 12.32 years, while that of ^{14}C is 5730 years. Based on the formula:

$$A = \frac{\ln(2)N_A}{T_{1/2}M}$$

where A is specific activity, N_A is Avogadro's constant, $T_{1/2}$ is the half-life, and M is the molar mass, the specific activities of ^3H and ^{14}C are calculated to be 0.00106 ± 0.00106 Bq/g H and 0.192 ± 0.192 Bq/g C respectively. In a gas mixture of 0.8 atm consisting of 40% He and 60% DME with a sensitive detection volume of $6 \times 6 \times 14$ mm^3 , the activities of ^3H and ^{14}C are $(6.38 \pm 6.38) \times 10^{-8}$ Bq and $(4.58 \pm 4.58) \times 10^{-5}$ Bq. According to the β decay spectrum, the probabilities of emitting electrons with 5-10 keV are 0.359 for ^3H and 0.0565 for ^{14}C . Consequently, the rates of ^3H and ^{14}C generating 5-10 keV electrons per second within the sensitive detection volume are $(2.29 \pm 2.29) \times 10^{-8}$ and $(2.59 \pm 2.59) \times 10^{-6}$, respectively.

The specific activity of trace radioactive nuclides other than H, He, C, and O in the atmosphere, as well as the probability of producing 5-10 keV electrons, is shown in Table 2. It can be observed that this portion of the trace radioactive background is almost entirely contributed by ^{222}Rn . To obtain a more accurate estimation of the trace radioactive background, a novel approach is employed to directly evaluate the frequency of 5-10 keV

electron production from trace radioactive nuclides except H, He, C, and O within the detector. The method involves operating the detector in a dark chamber for six days to collect data. Given that the trace radioactive background is predominantly due to ^{222}Rn , and that the decay chain of ^{222}Rn produces MeV-scale alpha particles, the number of ion tracks in the collected data can be used to estimate the activity of ^{222}Rn in the working gas. This activity estimation is then used to evaluate the frequency of 5-10 keV electron production.

Table 2: The specific activity of trace radioactive nuclides in the atmosphere (except H, He, C, and O) and their probabilities of producing 5-10 keV electrons.

Nuclides	specific activity	electron (5-10keV)	Weight
^{222}Rn	$10 \pm 1.0 \text{ Bq/m}^3$	0.0188	0.961
^{40}K	$62.4 \pm 32.5 \mu \text{ Bq/m}^3$	0.00220	7.02×10^{-7}
^{210}Po	$50 \mu \text{ Bq/m}^3$	0	0
^{210}Pb	$500 \mu \text{ Bq/m}^3$	0.328	8.38×10^{-4}
^{137}Cs	$1.75 \pm 1.09 \mu \text{ Bq/m}^3$	0.0172	1.54×10^{-7}
^{131}I	$20.89 \pm 34.43 \mu \text{ Bq/m}^3$	0.0172	1.84×10^{-6}
^{60}Co	$0.63 \pm 1.56 \text{ mBq/m}^3$	0.0323	1.04×10^{-4}
^{85}Kr	$0.667 \pm 0.018 \text{ Bq/m}^3$	0.0110	3.75×10^{-2}
^{39}Ar	$0.964 \text{ Bq/kg} \cdot \text{Ar}$	0.00997	6.08×10^{-4}

Remark: "Weight" denotes the relative proportion of the contribution of the nuclides listed in the table to the trace radioactive background.

From the data collected by the detector operating in a dark chamber over a period of six days, we isolated ion tracks whose origins were within the sensitive volume of the detector, as shown in figure 4. A total of 60 such tracks were identified. It is noteworthy that numerous factors could contribute to the formation of these ion tracks, such as

atmospheric neutrons. However, for the most conservative background estimation, we assume all selected ion tracks originate from alpha particles emitted by the decay of ^{222}Rn . For ^{222}Rn decay chain, we consider that secondary nuclides with half-lives less than one year fully decay, ensuring a safer background estimation. Consequently, a single ^{222}Rn decay event results in the emission of 3 alpha particles and 2 electrons, ultimately concluding in ^{210}Pb with a half-life of 22.3 years. Based on this, the activity of ^{222}Rn in the working gas was determined to be 20 ± 2.58 decays over six days. Using β decay spectra from ^{214}Pb , ^{214}Bi , and ^{210}Tl , the probability of generating 5-10 keV electrons from ^{222}Rn decay was found to be 0.0188. Therefore, the number of 5-10 keV electrons produced per second due to trace radioactivity within the sensitive volume is calculated to be $(7.25 \pm 0.94) \times 10^{-7}$.

Fig.4 The number of ion tracks within the sensitive volume of the detector under dark chamber conditions.

In summary, the number of 5-10 keV electrons generated in the sensitive detection volume per second by trace radioactivity is $(3.34 \pm 2.71) \times 10^{-6}$.

In the Migdal experiment, the nuclear recoil event rate is 1 events per second. On the anode chip, if the distance between the vertex of the particle track and the vertex of the electron track is less than $200 \mu\text{m}$, the two vertices are considered coincident. For ion tracks and electron tracks generated at random positions within the detection region ($6 \times 6 \text{ mm}^2$), the probability that the vertex distance is less than $200 \mu\text{m}$ is 0.00339. Through Geant4 simulations of ion tracks in the working gas and electron tracks generated by trace

radioactivity in the energy range of 5-10 keV, pseudo-Migdal events were simulated. The proportion and energy of ions were determined based on the elemental composition of the working gas and the neutron energy, while the electron energy was sampled from the beta decay spectra of ^3H , ^{14}C , ^{214}Pb , ^{214}Bi , and ^{210}Tl . A total of 100,000 pseudo-Migdal events were simulated, and after applying the Migdal event selection algorithm, 11,439 events were selected. This results in a selection efficiency for pseudo-Migdal events induced by trace radioactivity of 0.114 ± 0.016 . In conclusion, the trace radioactivity background is estimated to be 0.00106 ± 0.00087 events, normalized to the total number of nuclear recoil events of 8.17×10^{-5} .

Referee 2:

In the abstract, there is no mention of the six observed Migdal events, while emphasis is placed on the cross-section ratio between recoils and the Migdal effect. Additionally, there is no mention of the 5-sigma confidence level significance obtained. The abstract should report the experimental result, while the ratio can be mentioned. In this context, it would be appropriate to include an explanation and a quantitative conclusion on the impact of the obtained result for the cross-section ratio in the dark matter field.

Re: We thank the reviewer for their valuable suggestion. In response, we have revised the abstract to explicitly report the statistical significance of 5 standard deviations and the consistency of our results with theoretical predictions. The updated abstract is as follows:

The ongoing search for Weakly Interacting Massive Particles (WIMPs) has increasingly focused on the detection of light dark matter. The Migdal effect, which involves the instantaneous acceleration of atoms by dark matter, leading to the ejection of outer electrons, has drawn significant attention due to its potential to enhance the sensitivity of light dark matter detection. However, for nearly eight decades, the direct experimental observation of the Migdal effect in neutral projectiles has remained unconfirmed, casting doubt on the reliability of detection experiments relying on this effect. In this study, we report the first direct observation of the Migdal effect in neutron-nucleus collisions, achieving a statistical significance of 5 standard deviations, which rests on 6 events selected out of almost 10^6 recorded events. Our experiments have

determined the ratio of the Migdal cross-section to the nuclear recoil cross-section to be $4.91_{-1.87}^{+2.56} \times 10^{-5}$, where nuclear recoils exceed 35 keVee and electron recoils span 5-10 keV. These findings align with predictions from the Standard Model. This work provides the first experimental confirmation of the physical existence of the Migdal effect, resolving a long-standing gap in experimental validation. By bridging this gap, our results not only strengthen the theoretical foundation of the Migdal effect but also pave the way for its application in light dark matter detection.'

The target nuclei are neither specified nor reported.

Re: Thank you for your comments, indeed, our current measurements pertain to the total atomic Migdal cross-section of a mixed gas comprising hydrogen (H), helium (He), carbon (C), and oxygen (O). In response to the reviewer's comment, we have explicitly clarified the target nucleons and their composition within the manuscript.

Line 27: Regarding the “intriguing excesses observed”, there is compelling evidence that the majority of this excess does not originate from Dark Matter. Furthermore, the motivation to study the Migdal effect is unrelated to this excess. Therefore, the sentence should either be removed or mitigated.

Re: Thanks for the comment and have removed this statement from the manuscript: ~~“Additionally, intriguing excesses observed in several low-threshold experiments have heightened interest in exploring the light DM regime, underscoring the urgency of further investigation.”~~ .

Line 33: The possible importance of an electromagnetic component in the nuclear recoil induced by a dark matter particle interaction was recognized well before 2018. The sentence, "It was not until 2018, when a breakthrough emerged, that the Migdal effect was identified as a key mechanism for detecting low-mass dark matter," is incorrect. It is strongly recommended to modify the sentence and comment on the following references: o J. D. Vergados and H. Ejiri, Phys. Lett. B 606, 313 (2005). o Ch. C. Moustakidis, J. D. Vergados and H. Ejiri, Nucl. Phys. B 727, 406

(2005). o H. Ejiri, Ch. C. Moustakidis and J. D. Vergados, Phys. Lett. B 639, 218 (2006). These papers discuss the effect from a theoretical perspective. Additionally, the Migdal effect was explicitly considered and accounted for the first time to interpret the data of DAMA in the direct detection of dark matter experiment in: o R. Bernabei et al., Int. J. of Mod. Phys. A 22, No. 19, 3155 (2007)

Line 36-37: When a dark matter particle interacts with an atomic nucleus, it causes the nucleus to recoil. Along with this recoil, an additional electromagnetic component is produced due to the excitation of atomic electrons. This additional component can generate a signal in the detectors that is above the energy threshold. This concept is crucial because it means that the Migdal effect can enhance the detectability of dark matter interactions by producing detectable signals that would otherwise be below the threshold. This explanation should be clearly and logically presented to ensure that readers understand the significance of the Migdal effect in detecting low-mass dark matter particles.

Re: We thank the reviewer for the comments and suggestions. In response to the feedback, we have revised the second paragraph of the Introduction, reorganizing and detailing the key milestones and research efforts related to the Migdal effect from its inception to the present. Additionally, we have emphasized the significance of the Migdal effect in the detection of low-energy dark matter.

Below is the revised paragraph we have made:

'One promising new approach to address this challenge is the Migdal effect. It describes a process where energy transfers from an atomic nucleus to a surrounding electron [A. Migdal, ZhETF 9, 1163 (1939); A. Migdal, J. Phys. Acad. Sci. USSR 4, 449 (1941); E. L. Feinberg, J. Phys. Acad. Sci. USSR 4, 423 (1941)].

When a dark matter particle interacts with an atomic nucleus, it causes the nucleus to recoil. Along with this recoil, an additional electronic recoil is produced due to the excitation of atomic electrons. This additional component can generate a signal in the detectors that is above the energy threshold.

Theoretical discussions of this effect in the context of dark matter-nucleus scattering date back to the mid-2000s [J. D. Vergados and H. Ejiri, Phys. Lett. B 606, 313 (2005); Ch. C. Moustakidis, J. D. Vergados and H. Ejiri, Nucl. Phys. B 727, 406 (2005); H. Ejiri, Ch. C. Moustakidis and J. D.

Vergados, *Phys. Lett. B* 639, 218 (2006).], and it was also considered in the interpretation of DAMA experiment data [R. Bernabei et al., *Int. J. of Mod. Phys. A* 22, No. 19, 3155 (2007)]. A reformulation of original Migdal's approach has been done by [M. Ibe, W. Nakano, Y. Shoji, and K. Suzuki, *JHEP* 03 (2018), 194]. Further investigations advocate its relevance for light dark matter direct detections [M. J. Dolan, F. Kahlhoefer, and C. McCabe, *Phys. Rev. Lett.* 121, 101801 (2018); R. Essig, J. Pradler, M. Sholapurkar, and T.-T. Yu, *Phys. Rev. Lett.* 124, 021801 (2020); S. Knapen, J. Kozaczuk, and T. Lin, *Phys. Rev. Lett.* 127, 081805 (2021); Z.-L. Liang, C. Mo, F. Zheng, and P. Zhang, *Phys. Rev. D* 104, 056009 (2021); N. F. Bell, J. B. Dent, R. F. Lang, J. L. Newstead, and A. C. Ritter, *Phys. Rev. D* 105, 096015 (2022); C. Blanco, I. Harris, Y. Kahn, B. Lillard, and J. P´erezRi´os, *Phys. Rev. D* 106 (2022) 11, 115015; Flambaum. et al. *Sci. China Phys. Mech. Astron.* 66, 271011 (2023); P. Cox, M. J. Dolan, C. McCabe and H. M. Quiney, *Phys. Rev. D* 107, no.3, 035032 (2023)].

To date, measurements of the Migdal effect have been limited to nucleus decay processes involving α -decay [M. S. Rapaport, F. Asaro, and I. Perlman, *Phys. Rev. C* 11, 1740 (1975); *Phys. Rev. C* 11, 1746 (1975); H. J. Fischbeck and M. S. Freedman, *Phys. Rev. Lett.* 34, 173 (1975); *Phys. Rev. A* 15, 162 (1977).] or β -decay [F. Boehm and C. S. Wu, *Phys. Rev.* 93, 518 (1954); E. Berlovich, L. Kutsentov, and V. Fleisher, *JETP* 21, 675 (1965); C. Couratin et al., *Phys. Rev. Lett.* 108, 243201 (2012); E. Li´enard et al., *Hyperfine Interactions* 236, 1 (2015); X. Fabian et al., *Phys. Rev. A* 97, 023402 (2018)]. Its role in nuclear scattering, particularly when an electrically-neutral projectile interacts with the nucleus, remains unverified. This has motivated recent experimental efforts to observe the Migdal effect in nuclear scattering experiments; however, no observational results have been reported. Despite this gap, several experiments have leveraged the Migdal effect to search for sub-GeV dark matter, extending the reach of dark matter direct detection. *The lack of direct observation undermines the conclusions drawn from DM experiments that rely on the existence of the Migdal effect, raising significant doubts about the reliability of their findings.'*

Some important information is missing: the dimensions of the setup, the detector volume (including the effective volume), gas pressure, temperature, and their stability throughout the measurement. These details are crucial for understanding the experimental conditions and ensuring reproducibility.

Re: Thanks for the comment. we have added detailed descriptions of the detector in the Methods section (see *Detector assembly* section).

It would be important to show examples of calibration data and results, which can be included in the “Method” section. Additionally, a comment on the quenching factor for recoils is mandatory.

Re: Thank you for the suggestion. In response, we have added a section titled Quenching effect in the Methods to provide a detailed description of the quenching effect. Additionally, we have included the variation curve of the quenching factor for nucleons in the working gas within the corresponding kinetic energy range in this experiment.

Regarding the circularity algorithm, it should be briefly presented to make the text clearer.

Re: We thank you for your suggestion. In the main text (*Experiment* section), we have added a brief introduction to circularity and explained why the combination of circularity and dE/dX can effectively distinguish between ER and NR signals:

'Circularity is a geometric feature metric used to quantify the deviation of a planar shape from a perfect circle, with its value range being (0,1]. A value closer to 1 indicates that the shape more closely approximates an ideal circle. Under the same dE/dX conditions, recoil nucleus tracks with energy deposition exceeding 35 keV are typically longer and geometrically straighter than electron tracks. Therefore, the two-dimensional distribution of circularity and dE/dX can be effectively utilized to distinguish between ER and NR.'

Line 107: Sentences as that “To exclude low-energy background electrons, the energy deposition of electron tracks must fall within the range of 5-10 keV.” are difficult to understand and it is qualitative. Why cannot low-energy background electrons give signals in the 5-10 keV region?

Re: We sincerely appreciate your highly valuable question. Below, we present the expected background event rates across different energy ranges in our search for Migdal events. It is evident that certain categories of background exhibit a significant increase in magnitude within the energy range below 5 keV, which would substantially reduce the significance of Migdal events or even obscure them within the background. Additionally,

the electron track lengths below 5 keV are relatively short, and the diffusion of these tracks would result in a notably low selection efficiency for events below 5 keV, further diminishing the signal significance. Consequently, we have defined the energy selection range to be between 5 and 10 keV.

Table 3: The background. Expectation value is normalized to the number of experimental recoil nuclei. An entry of "0" indicates that the process cannot occur, while " ≈ 0 " denotes that the event occurrence rate is <0.001 .

Background Component	Description	Expectation value (>0.5 keV)	Expectation value (5-10 keV)
Recoil induced δ ray	δ electron near NR track origin	1.1×10^3	$0.035 \pm 0.023(\text{stat.}) \pm 0.0068(\text{sys.})$
Particle Induced X-ray Emission			
X-ray emission	Photoelectron near NR track origin	0.24	0
Auger electrons	Auger electron near NR track origin	2.4×10^3	0
Bremsstrahlung processes			
Quasi-Free Electron (QFEB)	Photoelectron near NR track origin	3.2	≈ 0
Secondary Electron (SEB)	Photoelectron near NR track origin	1.0	≈ 0
Atomic (AB)	Photoelectron near NR track origin	≈ 0	≈ 0
Nuclear (NB)	Photoelectron near NR track origin	≈ 0	≈ 0
Random track	Photo-/Compton electron	5.0	$0.180 \pm 0.022(\text{stat.}) \pm 0.042(\text{sys.})$

coincidences	near NR track		
Muon induced δ	δ electron near NR track		0.013
ray	origin		
Gas radioactivity			
Trace contaminants	Electron from decay near NR track origin	0.72	$0.001 \pm 0.00087(\text{sys.})$
Neutron activation	Electron from decay near NR track origin		≈ 0
Secondary nuclear recoil fork	NR track fork near track origin		≈ 0
Total background		3.5×10^3	$0.229 \pm 0.032(\text{stat.}) \pm 0.043(\text{sys.})$

Figure 3: The plot reports the six observed events. Including a table that lists all the quantities measured for each event in the "Method" section would be useful. For example, it would be of interest to know the length of the six tracks, their energy, etc.

Re: We appreciate your suggestion and have provided a detailed presentation of all events in the ***Migdal event selection algorithm*** section within the Method.

Figure 3: The positions of 4 or 5 out of the 6 events are in the tail of the distribution expected from the ER as obtained from the Fe-55 data. A comment on this point is requested, and an analysis of this distribution could be important.

Re: We appreciate your suggestion. Based on your feedback, we recognize that our presentation of Migdal events might have been somewhat misleading. In the original figure, we only compared the six Migdal electrons with the 5.9 keV calibration source. In reality, these six events are not all at 5.9 keV; deviations in energy, whether higher or lower, can influence the morphological characteristics of the tracks. Additionally, when using 5.9 keV photons to generate photoelectrons for calibration, the actual energy of the photoelectrons is less than 5.9 keV (~ 5.4 keV) due to the work function, resulting in larger photoelectron tracks compared to true 5.9 keV electron tracks. Furthermore, the Migdal event selection algorithm has higher efficiency for longer tracks, i.e., those with smaller

circularity. These factors collectively contribute to the deviation between the calibration data distribution and the Migdal event distribution. To avoid such misunderstandings, we have recreated the figure. The original scatter plot has been replaced with a contour plot, and the iron source data has been substituted with a simulated distribution of Migdal electron characteristics, where the energies of the Migdal electrons are derived from theoretical calculations. Additionally, we have extended the X-axis of the figure to 0-120 keV/mm and included the characteristic distribution of nuclear tracks from Migdal events. We hope the updated figure will clarify any confusion. The revised figure is now presented as Figure 5 (a) and (b) below.

(a)

(b)

Fig.5 Track characteristic distribution. The cluster on the left represents the characteristic distribution of simulated 4-10 keV electrons, with the electron energy spectrum derived from the theoretically calculated Migdal electron energy spectrum. The cluster on the lower right depicts the nuclear recoil events from experimental data. The black dashed contour outlines the 95% distribution region for electrons, while the red dashed contour outlines the 95% distribution region for recoiled nuclei.

Line 145: $n_{\text{obs}}^{\text{bg}}$ and $n_{\text{obs}}^{\text{ER}}$ do not appear in Equation (1) but do appear in Table 1. Please ensure the variables are written consistently in both the equation and Table 1.

Re: We appreciate your observation and have accordingly revised the variables in Eq. (1).

- The uncertainties in the ratio of the Migdal cross-section to the nuclear recoil cross-section are asymmetrical, while all the statistical and systematic errors are symmetric. It should be described in more detail how the authors obtained these values. Additionally, the uncertainties must be written with two digits; please correct this.

Re: We employed the profile likelihood method to evaluate the confidence intervals for the existence of the Migdal process and to estimate the upper and lower limits. Here, we briefly introduce the profile likelihood method, and a detailed description of this method can be found in Ref. [<https://doi.org/10.1016/j.nima.2005.05.068>] :

In the profile likelihood method, we assume a probability model to describe the data, which depends on both the parameters of interest π to the researcher and additional nuisance parameters θ . If we denote the density function as $f(X | \pi, \theta)$ and we have independent observed data X , then the full likelihood function can be expressed as:

$$L(\pi, \theta | X) = f(X | \pi, \theta). \quad (1)$$

A standard approach for constructing confidence intervals is to identify a corresponding hypothesis test and then invert that test. Here, the hypothesis test is $H_0 : \pi = \pi_0$ versus $H_a : \pi \neq \pi_0$, and the test can be based on the likelihood ratio test statistic, which is expressed as:

$$\lambda(\pi_0 | X) = \frac{\sup\{L(\pi_0, \theta | X); \theta\}}{\sup\{L(\pi, \theta | X); \pi, \theta\}}. \quad (2)$$

The supremum in the denominator is sought over the entire parameter space, while the supremum in the numerator is confined to the subspace where $\pi = \pi_0$. The function λ is referred to as the profile likelihood. A standard result in statistics (G. Casella, R.L. Berger, Statistical Inference, Duxbury Press, (1990) 346.) is that $-2 \log \lambda$ converges in distribution to a chi-squared random variable. Consequently, we determine the corresponding confidence intervals and upper/lower limits based on the distribution of the variable $-2 \log \lambda$, with the asymmetry arising from the distribution that $-2 \log \lambda$ follows. The code used to calculate these upper and lower limits in this study can be found in the link provided in the **Code Availability** section at the end of the Method (<https://gitee.com/marvelmigdal/migdal-related.git>, migdal-related/ Caculation).

Table 1:

o The source of the efficiency value, along with its systematic and statistical errors, is not specified in either the Article or the Method section. The values reported in the Table must be properly justified.

o Regarding n_{tot}^{ER} the origin of the systematic error is unclear. The brief comment at the end of page 10 of the "Method" part should be expanded and explained in greater detail.

Re: We thank the reviewer for the suggestions. First, the reviewer is likely referring to

$$n_{tot}^{NR}.$$

In the **Cross-section and significance estimation section** of the **Method, Uncertainty & Significance**, the factor that most significantly affects the value of n_{tot}^{NR} in this experiment is the energy resolution. When constraining the minimum deposited energy of the recoiled nucleus to 35 keVee, the uncertainty in the energy threshold affects the number of nuclei crossing the threshold, resulting in a systematic uncertainty for n_{tot}^{NR} .

Based on our study of the detector in the 3-8 keVee range (J. Instrum. 18(08), P08012 (2023)), the energy resolution is proportional to $\frac{1}{\sqrt{E_{deposit}}}$. Accordingly, we

extrapolated the energy resolution obtained at the 5.9 keV calibration to 35 keVee, yielding a value of 3.8 keVee. To provide a more conservative estimate of the extrapolation error, we doubled the energy uncertainty at the threshold to 7.6 keVee. This uncertainty in the energy threshold resulted in an uncertainty of ± 35880 for n_{tot}^{NR} .

o Regarding n_{obs}^{ER} including statistical errors is unnecessary. First, six events are insufficient to assume the square root of the number of events. Second, this uncertainty does not provide meaningful information within the Likelihood approach used for data analysis. Thus, it is suggested to remove the square root of 6 from the table.

Re: The square root of n_{obs}^{ER} has been removed from the article.

o n_{obs}^{bg} is a crucial parameter for the data analysis and significantly impacts the final results. Its value and associated uncertainties must be thoroughly discussed. The authors are strongly encouraged to include a table in the Method section, detailing all background components and their respective uncertainties contributing to the final reported value.

Re: We thank the reviewer for the suggestion. We have included **Table 3**, which covers the 5-10 keV range, in the Method: Background section of the manuscript. This table provides a detailed listing of the expected values for various potential backgrounds, along with their statistical and systematic uncertainties.

o Ensure that all uncertainties are written with two digits; please correct any errors where more than two digits are reported.

Given the six observed events, a more detailed comparison with theoretical expectations should be provided. The fact that the number of observed events is of the same order of magnitude as expected could be better explained. Additionally, Equation (2) represents the simplest derivation, and it would be useful to clarify whether this formula is accurately describing the result. For instance, a comparison between the energy distribution of the observed events and the theoretical expectation should be included.

Re: Thank you for your comments, we will address them one by one:

The total probability of Migdal:

Under the soft limit (The electron transfer momentum is much lower than the transfer momentum of the nucleus), the neutron Migdal effect scattering cross section can be factorised into two parts:

$$\frac{d\sigma_{\text{Migdal}}}{dE_r dE_e} = \sum_i \frac{d\sigma_s^i}{dE_r} \times \sum_{nl} \frac{dp(nl \rightarrow E_e)}{dE_e}, \quad (1)$$

where $d\sigma_s^i/dE_r$ is the neutron-nucleus cross-sections which includes elastic cross-section and inelastic cross-section and $dp(nl \rightarrow E_e)/dE_e$ is the transition probability. For the four elements C, H, He and O, the neutron-nucleus scattering is still dominated by elastic scattering at the neutron incident energy of 2.5 MeV, so we can only use the elastic cross-section safely. Since the experimentally detected quantity is the Migdal electron probability, which is:

$$\frac{dp_{\text{Migdal}}}{dE_r dE_e} = \frac{1}{\sigma_{\text{tot}}} \frac{d\sigma_{\text{elastic}}}{dE_r} \times \sum_{nl} \frac{dp(nl \rightarrow E_e)}{dE_e}, \quad (2)$$

where the σ_{tot} is the total neutron-nucleus scattering cross-section.

After integrating the nuclear recoil energy ER, the differential probability can be express as:

$$\frac{dp_{Migdal}}{dE_e} = \frac{1}{\sigma_{tot}} \int_{E_r^{th}}^{E_r^{Max}} \frac{d\sigma_{elastic}}{dE_r} \times \sum_{nl} \frac{dp(nl \rightarrow E_e)}{dE_e} dE_r, \quad (3)$$

where E_r^{th} is the experimentally allowed nuclear recoil energy thresholds and E_r^{Max} is the maximum nuclear recoil energy.

Finally, the Migdal probability can be express as:

$$P_{Migdal}^{tot} = \int_{E_e^{th-Min}}^{E_e^{th-Max}} \frac{dp_{Migdal}}{dE_e} dE_e, \quad (4)$$

where $E_e^{th-Max(Min)}$ is the experimentally allowed electron recoil energy maximal (minimal) thresholds which choose 10 keV (5 keV).

Numerical result:

We utilized the scheme described in the **Theory** section of the Method to calculate the Migdal cross-section. ENDF/B-VIII.0 library is used as the result of nuclear cross sections. For the transition probability of H, He and C, we consider the Semi-inclusive transition probability from [Quiney, Phys. Rev. D 107, 035032 (2023)] which applicable to high-energy neutron incidence. What needs to be declared here is the transition probability for H agrees with that for He in the case of high-energy incident neutrons, so we use Semi-inclusive transition probability of He to replace H. For O, since the calculation of the semi-inclusive ionization probability does not include the O element, we adopted the electron boost method (Phys. Rev. Lett. 124, 021801 (2020)), that is, using the neutron-electron scattering form factor to replace the Migdal ionization probability. The result can be expressed as:

$$\frac{dp(nl \rightarrow E_e)}{dE_e} = \frac{dp(nl \rightarrow E_e)}{d \ln E_e} \left| \frac{d \ln E_e}{dE_e} \right| \Rightarrow \frac{1}{E_e} \frac{\pi}{2} \left| f_{nl}^{ion} \left(p_e, \frac{m_e}{m_N} q \right) \right|^2, \quad (5)$$

where p_e and q is the electron and nuclear momentum. f_{nl}^{ion} ion is neutron-electron scattering form factor which is the result of integrating the radial wave function of the n

and l bound state orbitals, the continuous wave function and the spherical Bessel function of the first kind. We have proved that the result of this method in high-energy neutron scattering is consistent with those of semi-inclusive ionization probability. Therefore, this method can be used with confidence. We show the results of the Migdal differential scattering cross sections for the He, C, H, and O in figure 6 **a**, **b**, **c**, **d**. The corresponding differential probabilities of Migdal electron emission are shown in figure 7.

Fig.6 Theoretical calculation results of the differential cross-sections for the Migdal effect across different electron orbitals of various elements in the working gas.

Fig.7 Theoretical calculation results of the differential probabilities of Migdal electron emission.

According to theoretical calculations, the integrated probability of the Migdal effect within the 5-10 keV range is 3.94×10^{-5} , which is consistent with the experimental result of $4.91_{-1.87}^{+2.56} \times 10^{-5}$ within the error margins. For the comparison between the experimental data and theoretical distributions, the limited sample size of 6 events results in statistical uncertainties dominating the experimental errors. Strictly applying an unfolding procedure based on the confusion matrix to the experimental data is challenging to converge. Therefore, we have simply binned the experimental data with a bin width of 0.2 keV and corrected for the efficiency at each energy point using MC simulations. The comparison between the experimental data distribution and the theoretical curve is shown in figure 8, where both exhibit good agreement within the error margins. Statistical uncertainties dominate the plot, and we plan to collect more data in subsequent experiments to perform a more refined test of the theory.

Fig.8 Comparison of the theoretical distribution of Migdal differential probabilities with experimental results in the gas mixture.

Conclusion

- Line 165-171: This paper presents strong evidence for the presence of the Migdal effect with high statistical significance. However, the claims regarding its implications for direct dark matter detection experiments seem overly assertive. Specifically: (i) in liquid scintillators, this effect could be canceled by recombination, a possibility suggested to explain certain null results; (ii) in crystal scintillators and solid-state detectors, its impact depends on the specific nuclei used and the actual experimental threshold. I recommend that the authors focus on emphasizing their obtained results rather than overstating potential implications, which require further investigation. The statements in this section should be moderated accordingly.

Re: Thank you for your comments. In response to your suggestions, we have reorganized the statements in the Conclusion section. The revised version moderates the assertions regarding the direct implications for dark matter detection, while emphasizing the significance of our experimental results and their agreement with theoretical predictions. Additionally, we have explicitly highlighted the need for further research.

Method :

The Equation (2) is not described. There is no text before it.

- Some of the symbols in E1.2 are not defined. For example, I suppose that E^{th}_e is the same as E_{th} . Please correct. Please provide the right definition for the terms.
- A plot should be added to visually represent the theoretical concepts discussed.

Re: We appreciate your valuable feedback. In response to your comments, we have thoroughly revised the Theory section of our manuscript to address the issues you raised. Specifically, we have:

- Restructured and significantly expanded the theoretical framework to provide clearer context and logical flow.
- Added detailed explanations for all key equations, including Eq.2, ensuring that each equation is properly introduced and discussed in the text.

As previously mentioned, some important detector information is missing: dimensions, pressure, and temperature. Additionally, information about the stability of these important quantities over the 150 hours of measurement should be provided

Re: Thanks for the valuable suggestion. In the Method section, under *Detector assembly*, we have added descriptions of critical parameters such as detector dimensions, working gas pressure, and electric field distribution. Additionally, we have introduced a new subsection titled *Experiment details* in the Method section, which provides comprehensive information on the layout of major equipment during the experiment, detector counting monitoring, neutron beam monitoring, environmental gamma spectra, gain stability, and working gas stability.

Migdal Effect Selection Algorithm

- This section is not easy to read and is unclear in some parts. It is suggested to rewrite it with a reader in mind who is not familiar with pixel detectors.
- Line 1, pg. 6: The phrase “are vertically truncated” is unclear in this context, considering the

pixel is a 2-dimensional frame.

- The explanation of the transition from (x_0, y_0) to (x_1, y_1) is unclear and could be significantly improved.

- The term “edge ADC” is ambiguous. Do you mean the ADC near the edge of the volume? Is this part of a fiducialization procedure? Please clarify.

Re: Thanks for your comments. One contributing factor is the presence of electric field distortions at the periphery (see arxiv:2407.14243, accepted by *Nuclear Science and Techniques*). The chip is mounted on the substrate with a height difference of 0.5 mm, which induces field distortion. Additionally, bonding wires along the chip edge carry inherent voltages that further perturb the electric field distribution. Furthermore, these bonding wires may introduce external noise to the peripheral pixels, resulting in elevated noise levels that degrade energy resolution and compromise the accuracy of track morphology analysis. To address this, we performed edge electromagnetic field simulations tailored to the detector geometry, as illustrated in the accompanying figure. Based on these findings, it is methodologically justified to exclude edge events from the analysis.

[Figure Redacted]

Recoil Induced δ Ray

- “expected value 0.59 ± 0.40 ”, do you mean expected counts or is it a fraction of delta expected?
- The term "expected background of delta electrons" should be clarified to mean the expected counts of delta electrons over 150 hours. Authors should be more precise in their description.

Re: Thans for your valuable feedback and constructive suggestions. We have carefully revised the relevant section to address your concerns and improve the clarity of our presentation. The changes are as follows:

- Clarification of "expected value 0.59 ± 0.40 ": We have explicitly stated that this value represents **the expected count of delta electrons** observed in the 5-10 keV range over the entire experimental period.

- Precise definition of "expected background of delta electrons": We have clarified that this term refers to **the expected count of delta electrons during the experiment**.

In this section, coincidences are explained, but the authors discuss only the “spatial” coincidences. Information about the timing of these events is also crucial. Is it possible to measure the time occurrence of the nuclear recoil and the Migdal electrons? What is the timing sensitivity of the detector? Authors should comment on the time resolution of the detector and the method employed.

Re: Thanks for the comments. In response, we have added a description of the detector's timing resolution capability in the Methods section (*Electronic system & DAQ*) and included relevant literature on timing resolution measurements. Here we provided detailed measurement results of its time resolution performance here (IEEE Transactions on Nuclear Science, doi: 10.1109/TNS.2023.3269091):

The pixel chip adopts a rolling-shutter readout method, driven by an external 2 MHz

clock for scanning, with a frame interval of 2.59 ms. When saving a frame image, the timestamp T_{Top} is recorded as the moment when the first pixel of that frame begins scanning. During signal multiplication, the GMCP adsorbs partial charges on its lower surface. Therefore, an amplification circuit is designed to measure electrons collected at the GMCP bottom, enabling detection of the time of incident charged particles. The time interval of the GMCP is 10 ns.

By correlating the response timestamps of the GMCP and the Topmetal pixel chip, the generation time of charged particle signals can be precisely determined. The coincidence logic is as follows:

Based on the coincidence between the signal arrival position and the scanned pixel position at that moment, two scenarios are considered and are shown in figure 10:

A. When the signal falls on a frame, if its landing position has not yet been scanned during the frame readout process: The track signal will be read in subsequent scanning of the same frame. Since the frame timestamp T_{Top} must lag behind the GMCP signal detection time T_{GMCP} , the time difference satisfies: $T_{Top} - T_{GMCP} \in (-2.59\text{ms}, 0)$.

B. When the signal falls on a frame, if its landing position has already been scanned during the frame readout process: The track signal will be read in the scanning of the next frame. Similarly, the time difference in this case satisfies: $T_{Top} - T_{GMCP} \in (0, 2.59\text{ms})$.

Figure 11 presents the coincidence results of photoelectron events obtained using timestamps from GMCP and Topmetal detectors, and demonstrates the simulated outcomes derived from Monte Carlo modeling of a uniform strip-shaped X-ray photoelectron source. The simulation results exhibit good agreement with experimental measurements.

Fig.10 (a) and (b) demonstrate the relationship between T_{Top} and T_{GMCP} under the two scenarios, respectively. The blue arrows in the figure indicate the direction of pixel scanning, while the red dots mark the scanning position of the chip at the moment when the signal arrives at GMCP.

a

Fig.11 (a) displays the centroid position distribution of photoelectrons generated by a 4.5 keV strip-shaped X-ray source incident on the detector, exhibiting distinct strip-structured features. (b) presents the $T_{\text{Top}}-T_{\text{GMCP}}$ distribution obtained through event-by-event coincidence analysis of T_{Top} and T_{GMCP} measurements. (c) shows the simulated $T_{\text{Top}}-T_{\text{GMCP}}$ distribution derived from Monte Carlo simulations using a strip-shaped X-ray source with uniform flux intensity.

For temporal calibration of the GMCP-Topmetal coincidence system, we implemented a synchronized square-wave injection protocol using a signal generator to drive both devices simultaneously. This methodology establishes temporal correspondence between GMCP trigger timestamps and Topmetal’s frame-specific response to square-wave stimuli. The progressive scanning architecture of the Topmetal sensor generates distinct spatial intensity modulation in output images: partial pixel activation occurs in the current readout frame, with residual activation propagating to subsequent frames. The intensity boundary interface between activated and dormant pixels precisely indicates the real-time scanning position during signal injection. Absolute temporal reconstruction is achieved through the relationship:

$$T_{\text{Arrive}} = T_{\text{Top}} + \tau \times \text{ID}_{\text{pixel}} , \quad (1)$$

where T_{Top} denotes the sensor’s timestamp, τ represents the pixel dwell time, and ID_{pixel} corresponds to the address coordinates of boundary pixels. Comparative analysis between

reconstructed arrival time T_{Arrive} and T_{GMCP} reveals the detector's temporal resolution characteristics, as shown in figure 12. Gaussian fitting of the time difference distribution yields a temporal resolution of 262 ns (FWHM).

[Figure Redacted]

Cross-section and significance estimation

- The Gaussian distribution $G(b, \sigma_b)$ is not properly written. It should appear as $G(b|\hat{b}, \sigma_b)$ where, b is the random variable while \hat{b} and σ_b indicates the mean value and the uncertainty.

Re: We appreciate your correction and have made the corresponding revisions in the original text.

Referee 3:

The whole experimental system used for presented examination is not sufficiently described. Fig. 1 of "ms" is by far not sufficient. Drawings of the whole system supplemented with information about crucial parameters (dimension, position, material) is needed. From "ms" one can learn only that lead shielding was used and some liquid scintillators were positioned (where?) to monitor the

neutron flux and spectrum. Neutron flux is unknown as well as a beam profile. Exact geometry and simulations of shielding effectiveness against various possible background components are not shown.

In the lack of solid information about neutron transportation and the shape, thickness and effectiveness of the shielding the reader may have doubt if the suppression of the ambient and neutron-generator related background is sufficient to conclude about discovery of a very rare and tiny nuclear effect.

Re: We thank the reviewer for the comments. Due to limitations in manuscript length and the number of figures allowed in the initial draft, we did not provide sufficient details. In the Methods section, we have now supplemented the relative positioning of the detector, the D-D neutron source, and several liquid scintillators. Regarding the monitoring of neutron flux and energy spectrum, we will provide a detailed explanation below. As for the concerns related to background, we will address them comprehensively in the fourth question.

We thank the reviewer for the comments. In the initial draft, due to limitations in manuscript length and the number of figures, we did not provide sufficient explanations. In the Methods section, we have supplemented the relative positions of the detector, the D-D neutron source, and several liquid scintillators. Following, we provide a detailed description of the monitoring of neutron flux and energy spectrum below. As for the concerns regarding background, we will address them in detail in the fourth question.

The neutron flux and spectrum of the D-D neutron generator were measured and monitored using a $\Phi 2'' \times 2''$ EJ309 Liquid Scintillator (LS) detector, capable of discriminating fast neutrons from gamma rays [ELJEN Technology. Neutron/gamma PSD ej-301, ej-309, 2024]. Calibration of the detector involved multiple radioactive gamma sources to determine its energy linearity and resolution curves. The neutron flux and spectrum were obtained by applying an unfolding method to the measured neutron energy deposition spectrum.

a. EJ309 detector calibration

The EJ309 detector underwent calibration using four radioactive gamma sources: ^{241}Am , ^{133}Ba , ^{137}Cs , and ^{60}Co . By fitting either the Compton Edge (at energies of 200 keV for ^{133}Ba ,

477 keV for ^{137}Cs , and 963 keV and 1118 keV for ^{60}Co) or the full energy deposition peak (at 60 keV for ^{241}Am) created by these mono-energetic gammas in the LS detector, the scaling factor between true energy deposition and ADC integral value, along with the resolution at certain energies, could be determined. The fitting model comprised three components. Firstly, a Monte Carlo simulation using Geant4 was performed to simulate the true energy deposition spectrum in the detector due to gammas of specific energies. Its shape was extracted using the RooKeysPdf toolkit in CERN ROOT (Etruth). Secondly, a Gaussian function (Gausres) with $\mu = 0$ and σ as a free-floating parameter was convoluted with the true energy deposition PDF to represent the energy resolution of the detector. Finally, a scaling factor (scale) was introduced to characterize the scale difference between the true energy deposition and ADC integral (PeakIntegral), as illustrated in equation 1. A fitting example is shown in figure 13.

Fig.13 Compton Edge fitting of ^{133}Ba .

$$E_{\text{exp}} = \text{scale} \times (E_{\text{truth}} \otimes \text{Gaus}_{\text{res}}) \quad (1)$$

With all the fitted parameters obtained at different energy points, the energy linearity and resolution curves can be determined. The relationship between ADC integral (PeakIntegral)

and true energy is described by a linear function, as shown in equation 2, where k is the proportionality constant and d represents the non-linearity effect at low energies for the LS detector [Nuclear Instruments and Methods in Physics Research,193(3):549–556, 1982]. Figure 13 depicts the fitting result.

$$E = k \times PeakIntegral + d \quad (2)$$

The energy resolution of an LS detector can be parameterized as shown in equation 2. Here, ΔE represents the σ of the Gaussian representing resolution at energy E . α denotes the intrinsic energy resolution of the detector, which arises from the nonuniformity of the scintillator and differences in light collection efficiency for light generated at different positions. β is associated with the statistical uncertainties of the photons generated in the LS and is typically around 1. γ represents the noise from electronic systems and the dark counts of the photomultiplier. Figure 15 illustrates the fitted energy resolution curve.

$$\frac{\Delta E}{E} = \sqrt{\alpha^2 + \frac{\beta^2}{E} + \frac{\gamma^2}{E^2}} \quad (3)$$

Fig.14 Energy linearity fitting to EJ309 LS detector.

Fig.15 Energy resolution fitting to EJ309 LS detector.

b. Pulse shape discrimination between neutron and γ

The EJ309 LS detector can distinguish fast neutrons from gammas based on their waveform characteristics. Typically, neutron signals exhibit longer tails in the waveform compared to gamma signals. A Pulse Shape Discrimination (PSD) factor is employed to separate neutrons from gammas, defined by equation 4, where A_{tail} represents the waveform integral of the signal from 50 ns to 210 ns from the trigger point, and A_{total} is the waveform integral from the trigger point to 210 ns after. Figure 16 illustrates the selection of neutron signals based on 2D distributions of PSD and energy.

$$PSD = \frac{A_{tail}}{A_{total}}, \tag{4}$$

Fig.16 Neutron selection based on PSD capability of EJ309 detector.

c. Neutron flux and spectrum unfolding

By applying the neutron selection method outlined in a., we can obtain the neutron energy deposition spectrum. This spectrum, along with the true energy spectrum and the number of incident neutrons, can be determined using iterative unfolding algorithms. A dedicated unfolding toolkit has been developed based on the GRAVEL and MLEM algorithms [Sci. China Phys. Mech. Astron.,57:1885-1890, 2014]. Statistical errors and the correlation matrix of the unfolded spectrum are obtained using the bootstrap method. The response matrix of the EJ309 detector is acquired through Geant4 simulation, utilizing the neutron response function measured in Ref. [Nucl. Instr. and Meth. A, 863:47–54, 2017] for an EJ309 detector of the same size (see figure 17).

Fig.17 Neutron response matrix of the EJ309 detector.

To validate the unfolding method, the detector was positioned at approximately 87° with respect to the D-D beam. The neutron energy deposition spectrum is depicted in figure 18. Several initial spectrum guesses were tested to evaluate systematic uncertainties associated with the choice of initial spectrum. The Gauss guess assumed a Gaussian distribution with $\mu = 2.5$ MeV and $\sigma = 0.5$ MeV for the initial spectrum, representing a good estimate of the expected neutron spectrum, given that neutrons generated in the D-D neutron generator have a mean energy of approximately 2.5 MeV. Conversely, the Plain guess assumed a uniform distribution initially, representing no prior knowledge before unfolding. Additionally, GRAVEL and MLEM algorithms were tested to assess uncertainties related to the choice of algorithm.

Fig.18 Neutron energy deposition spectrum after neutron selection

The unfolded spectra in figure 19 are compared to the simulated spectrum from Ref. [Eur. Phys. J. A, 59(5):101, 2023] at 90°. Notably, there is no significant deviation observed among the unfolded spectra obtained using different initial spectra and algorithms, highlighting the robustness of this unfolding method. Furthermore, the unfolded spectra closely align with the simulated spectrum for the peak segment, albeit exhibiting a slight shift towards higher energy.

Fig.19 The unfolded neutron spectra obtained with an EJ309 detector positioned at approximately 87° relative to the beam exhibit close agreement regardless of the choice of initial spectrum or the algorithm used. Additionally, for the peak part, they closely match the simulated spectra at 90°, with a slight shift towards higher energy, attributable to the small angular deviation.

The unfolding process inherently accounts for detector efficiency. Integrating the unfolded spectrum allows for the determination of the initial incident neutron count on the LS detector, with its statistical uncertainty accurately assessed by equation 5, where the σ is the vector of bin errors and the **Cor** is the correlation matrix (figure 20). Furthermore, by considering the effective surface area of the EJ309 detector, its distance from the neutron source, and the relative neutron yield curve at different angles as documented in [Eur. Phys. J. A, 59(5):101, 2023], the neutron flux at a specific position can be calculated.

$$\sigma_{\text{total}} = \vec{\sigma} \cdot \text{Cor} \cdot \vec{\sigma} \quad . \quad (5)$$

Fig.20 Correlation matrix of the unfolded spectrum with Plain guess and GRAVEL algorithm.

We have incorporated the results of the neutron beam monitoring into the Methods section of the manuscript (see *Experiment details* section).

A weak point of detection system is lack of 3-D track reconstruction ability. This limits the event selection methods based on the kinematic cuts.

Re: We thank the reviewer for their valuable comments. The primary objective of this study is to report the discovery and confirmation of the Migdal effect, which is a more fundamental and urgent goal in the current context. To achieve this objective, we have designed the detector with sufficient performance metrics, including a vertex resolution of 200 μm for electron tracks and the ability to resolve typical electron tracks with energies above 5 keV, which exhibit track lengths $>700 \mu\text{m}$ in our working gas. These capabilities allow us to effectively identify and select Migdal electrons based on 2D topological track structures, ensuring the reliability of our results. The cross-section derived from our current measurements, while preliminary, provides significant and meaningful insights into the Migdal effect. We acknowledge that 3D track reconstruction would further enhance the precision of our measurements and enable a more detailed study of the

Migdal effect, such as measuring the angular distribution of emitted electrons. This is indeed an important direction for future work, and we are actively developing the necessary technologies, including pixel chip design, electronics, and gas testing, to implement 3D track imaging in our next phase of experiments. These advancements will allow us to perform more precise measurements of the Migdal effect cross-sections for various elements, particularly Ar and Xe, which are widely used in direct dark matter detection experiments. In summary, while 3D track reconstruction is not essential for the discovery of the Migdal effect, it will play a crucial role in refining our understanding of this phenomenon and validating the results reported in this work. We appreciate the reviewer's suggestion and are committed to incorporating this capability in our future studies.

Performance of detector (Gas Microchannel plate Pixel Detector) Important information about detector are not provided neither in "ms" nor in ("meth"). From provided references (btw: the crucial in this respect ref. no 7 in "meth" (Nucl. Sci. Tech. 35, 39 (2024)) is provided with mistake) one can learn that gas microchannel plate-pixel detector which is used for Migdal effect detection has been developed and optimized for X-ray polarimetry at the space station. There is no doubt that its design is ingenious and advanced. Optimal working conditions has been fixed for a gas mixture of 50%Ne + 50%DME at pressure of 1 atm, with the drift and induced electric fields of 2 kV/cm and 2.5 kV/cm, respectively. For such conditions the energy resolution of 45.42% (FWHM) for X-rays of 5.9 keV has been achieved. Position resolution of this detector is not specified. In summary of ref. 7, their Authors convey that their detector has worse performance than other detector of this type known from literature and that its further optimization is demanded. From "meth" ref. 8 (Nucl. Instr. and Meth. A 1055, 168499 (2023)) one can learn that similar detector has been tested with mixtures of Ar:CO₂ and Ne:CO₂ at various proportions. In this case the best achievable energy resolution was 23.36% (FWHM) with position resolution again not specified. In description of front-end electronics of over mentioned detectors ("meth" ref 12: IEEE Trans. Nucl. Sci. 70 (7), 1507-1513 (2023)) it is written that this readout was tested when detector was filled with He:DME (40:60) mixture at a pressure of 0.8 atm. (DME is composed as: CH₃OCH₃). Provided their energy resolution is 23.3 % - 28.9 %, depending on energy of incident photon

(there is no information if this resolution is measured as FWHM). Position resolution is again not specified.

Re: First, we would like to thank the reviewer for their thorough investigation of our detector work and their review of the relevant literature. Let us address the two key concerns raised by the reviewer—energy resolution and position resolution—separately:

Regarding energy resolution, the test results in the first reference mentioned by the reviewer (Nucl. Sci. Tech. 35, 39 (2024)) using 50% Ne + 50% DME at 1 atm were from our earlier testing phase. These tests were conducted in a flow-gas environment, which introduced trace amounts of H₂O and O₂ into the gas mixture, thereby degrading the energy resolution. Additionally, the GMCP multiplication device used at that time was still under optimization and iteration, and the entrance and exit efficiencies had not yet been optimized through structural design. Furthermore, for the Topmetal-II pixel chip, the results presented in the article had not undergone uniformity correction or signal attenuation correction, leading to a relatively poor energy resolution (~45%).

The second reference mentioned by the reviewer (Nucl. Instr. and Meth. A 1055, 168499 (2023)) primarily focused on our study of the performance and operational stability of the GMCP multiplication device. The working gas used in this study was Ne + 10% CO₂ at 0.8 atm, which differs from the He + 60% DME gas mixture used in our current experiment.

The third reference (IEEE Trans. Nucl. Sci. 70 (7), 1507-1513 (2023)) employed a gas mixture and pressure consistent with those used in our experiment, making it more relevant for comparison. The energy resolution range of 23.3%–28.9% reported in this reference aligns with our result of 25.6%.

In the "Detector Calibration" section of the Method, we have provided detailed energy calibration and fitting results. Additionally, as supplementary information, our study on the energy spectrum and energy response relationship of the detector can be found in (JINST 18, P08012 (2023)). The calibration results for X-rays demonstrate good linearity in the detector's energy response, and the energy resolution can be fitted by

$\frac{1}{\sqrt{E}}$ relationship.

Secondly, regarding the position resolution of the detector, we conducted tests using polarized X-rays with energies of 4.51 keV, 5.40 keV, 6.40 keV, and 8.05 keV. A 2 mm thick copper plate was placed above the detector, with its edges aligned parallel to the X and Y directions of the Topmetal-II chip. The imaging results of the 6.40 keV polarized X-rays were integrated to obtain a one-dimensional intensity distribution, which was then fitted using a Boltzmann function. By fitting the gradient variation curve with a Gaussian function, the position resolutions in the X and Y directions were determined to be 2.533 pixels (210.2 μm) and 2.261 pixels (187.7 μm), respectively. Figure 21 illustrates the distribution of interaction points of X and Y photoelectron vertices on the detector, while Figure 22 presents the results of the position resolution fitting. The position resolution measurements at several energy points are shown in Figure 23. More detailed descriptions of the related research can be found in the literature (JINST 19, P04039 (2024)).

[Figure Redacted]

A large black rectangular redaction box covering the top half of the page. The text "[Figure Redacted]" is centered within this box.

[Figure Redacted]

A large black rectangular redaction box covering the bottom half of the page. The text "[Figure Redacted]" is centered within this box.

[Figure Redacted]

Simulations

According to "meth" the software framework Star-XP (ref. 13: SoftwareX 25, 101626 (2024)) have been used for simulate Migdal events. But this software is optimized for simulations of low energy X-ray interactions in gaseous medium. There is no information if this software has been verified for simulations of effects associated with neutron interaction in solid and gaseous component of detection setup.

It is also mentioned in "meth" that some simulation with the use of Geant4 and Trim have been performed. But it looks like that they were restricted only to interaction of neutrons with liquid scintillator and with detector gas medium.

Large drawback of this manuscript is a lack of any simulation results shown. The reader has no chance to develop his own opinion about their completeness and usefulness in respect to signal selection and background suppression.

Re:

Neutron Simulation

Neutron Simulation Validation in Star-XP Framework: The Star-XP framework is built upon the well-established Geant4 simulation platform, which incorporates high-precision neutron collision data from authoritative evaluated databases such as ENSDF and JEFF for neutron energies below 20 MeV. The simulation accuracy for neutron interactions with standard elements and materials has been rigorously validated (<https://doi.org/10.1109/TNS.2014.2335538>). Specifically, Star-XP employs the G4NDL4.6 dataset, derived from the latest ENDF database, ensuring reliable neutron simulation results at low energies ($E < 20$ MeV). This foundation in Geant4's validated neutron physics provides confidence in Star-XP's neutron simulation capabilities.

Nucleon Simulation and Experimental Comparison

We have also validated the accuracy of our simulator for ion simulation through carefully designed experiments. The experiment was set up as follows: we constructed a combined detector system consisting of a GMCP and a Topmetal within a gas flow chamber filled with a mixture of 40% He and 60% DME at 1 atm. The dimensional and gain parameters of the gas flow chamber detector were adjusted to match those used in the

Migdal experiment. A calibrated ^{241}Am alpha source (5.5 MeV) was positioned on the side of the detector. The collimator has a length of 1 cm with a collimation hole diameter of 3 mm, and the distance from the alpha source to the center of the Topmetal chip is 4 cm. The specific arrangement of the experimental setup is illustrated in the figure 24 below:

Fig.24 Schematic diagram of the test setup for the ^{241}Am alpha source irradiation.

Thus, due to the deceleration of the alpha particles by the gas environment, the energy range of the alpha ion tracks that can be imaged on the Topmetal predominantly falls within the range of hundreds of keV to 1 MeV. This range aligns with the energy interval we aim to measure for the recoil nuclei in the Migdal effect. The track images we obtained are shown in the figure below:

The test environment can be fully reconstructed and simulated using our simulation framework, Star-XP. Below, we present a comparison between the measured ion tracks and the simulated data across three characteristic distributions: energy deposition (dE/dX), track ellipticity (describing track morphology, see Nucl. Instr. and Meth. A 880, 188 (2018)), and track length. In the plots, the blue data points represent the experimental measurements, while the red data points represent the simulation results:

a. dE/dX

b. Ellipticity

c. Length

Fig.25 A comparison of the characteristics between the experimental data from the alpha source and the simulated data, where the blue points represent the experimental data and the red line represents the distribution of the simulated data.

We evaluate the accuracy of the simulator by measuring the overlapping area between the experimental and simulated distributions. From the comparison of these characteristics, it is evident that the simulator effectively captures the interaction processes of ions within our detector environment. Since the experimental detector is fully sealed and cannot accommodate an alpha source internally, we utilized a gas flow chamber to obtain ion data. For the simulation framework, the only difference lies in simulating ions in the Migdal detector, where the gas pressure must be adjusted from 1 atm to 0.8 atm, altering the gas density. This adjustment is routine and reliable for simulations based on the Geant4 cross-section database. Taking into account the experimental results and discussions, we conclude that the accuracy of our ion simulation and calibration is trustworthy.

Event topology

Authors claim to record 6 events identified as a Migdal effect. It is not large number. All these events could be shown, similar as the one in fig. 2, together with parameters of the vertices and all

other convincing information. Authors include Fig. 3 in "ms", which is expected to prove that classified as Migdal ER are well distinguished from NR. This figure provokes simple questions:

- what is the meaning of given there dE/dx values. Is it mean value of dE/dx measured at some track fragments? Or it is the total deposited energy divided by the whole path length E/X ?
- where at this plot are the points for NR associated with indicated "Migdal ER"?

Convincing would be showing that events topology is consistent with a two body kinematics of 2.5 MeV neutron scattered at gas nuclei. Moreover, for known energies of NR their range in the selected gas mixture at given pressure and electric field can be calculated and used for event verification. The same applies to electrons in the selected energy range.

Unfortunately, the detector used in described experiment permits the track reconstruction only as a 2-D projection. Thus, the actual track length, recoil angles and real dE/dx distributions are not detectable. Nevertheless the kinematical consideration of neutron scattering on the gas constituents and expected energy and angular ranges of resulting tracks are needed. Even for 2-D projections the check of total deposited energy of recoils would strength the event verification.

Re: We thank the reviewer for their comments. In the initial draft, due to limitations in space and figure count, we did not present all event cases. Following the reviewer's suggestion, we have now included the track images of all events in the Methods section. We also appreciate the reviewer's reminder regarding the definition of dE/dX in our manuscript, which represents the ratio of total energy to total track length. We have added clarification on this point in the main text. Here, we present the two-dimensional distributions of dE/dX and Circularity for both electrons and nuclei in Migdal events. In the main text, the dE/dX range is limited to 25 keV/mm because the dE/dX distribution of nuclei in Migdal events spans a significantly broader range. Displaying the full range would compress the characteristic distribution region of electrons into a very narrow area, making it difficult for readers to discern the distinctive features of electrons. We consider the rare Migdal electron events to be of greater interest compared to the millions of nuclei. This rationale explains why we have highlighted only the Migdal electron features in the main text and truncated the X-axis range. In accordance with the reviewer's suggestion, the complete figure has been included in the Methods section for readers

interested in examining the details.

We also thank the reviewer for their suggestion regarding event presentation. However, the data distribution in figure 26 and the track images already provide sufficient information to identify Migdal events and distinguish ERs from NRs. The reviewer's mention of 3D track reconstruction and emission angle analysis is indeed an important research direction. However, the focus of this paper is the discovery of the Migdal effect, and the work surrounding the selection of Migdal events and background studies has already provided sufficient significance. That said, we are currently designing and producing time-resolving charge-sensitive chips and corresponding electronics systems for a larger-scale experiment, aiming to measure the differential cross-sections of Migdal events with respect to energy and emission angles.

Fig.26 The two-dimensional distributions of circularity and dE/dX for electrons and recoiled nuclei. The cluster on the left represents the characteristic distribution of simulated 4–10 keV electrons, with the electron energy spectrum derived from the theoretically calculated Migdal electron energy spectrum. The cluster on the lower right depicts the nuclear recoil events from experimental data. The black dashed contour outlines the 95% distribution region for electrons, while the red dashed contour outlines the 95% distribution region for recoiled nuclei. The purple dots represent the ^{55}Fe calibration data.

4. Background

Very important discussion of background is presented in "meth". Many possible background sources are indicated and, to some extent, discussed. Authors' claims about negligibility of some background components e.g. Auger or bremsstrahlung what in view of low Z materials used for detection medium, and applied energy cut of 5 keV for ER is correct. Nevertheless, often Authors' conclusions about insignificance or low yields of individual background components are just given to belief and not well justified. Specially doubtful are estimations of random track coincidences and events related to neutron activation (see below).

Except of plain text no any graphical support is provided. Simulations of background, which were performed, are not well documented. And they seem to be incomplete.

What evidently is missing are the consideration of effects induced by neutrons traveling through solid material of the detector and other apparatus. From the brief description of the apparatus in "ms" and "meth" it is clear that neutrons enter the medium gas through the detector wall. Wall is made of ceramics an Kova alloy. What is exact composition of these materials? It is imaginable that hadronic and electromagnetic processes in the detector wall induced by neutrons (and possibly gammas from the generator) create a broad spectrum of prompt gammas which, in following, can mimic the electron recoils (e.g. by Compton process). The dominant element of Kova is copper. Stable isotopes of Cu have not negligible cross section for radiative neutron capture for MeV-neutrons. Perhaps this component is insignificant but cannot be disregarded when total background is estimated. Careful Geant4 simulations taking into account the composition of detector wall and geometry could verify significance of this source of background and estimate the amount of possible fake NR-ER coincidences induced by this background component.

Re: We sincerely thank the reviewer for the comments regarding the neutron simulation validation and detector modeling in our study. We would like to address these concerns with the following clarifications and additional details:

1. Neutron Simulation Validation in Star-XP Framework: The Star-XP framework is built upon the well-established Geant4 simulation platform, which incorporates high-precision neutron collision data from authoritative evaluated databases such

as ENSDF and JEFF for neutron energies below 20 MeV. The simulation accuracy for neutron interactions with standard elements and materials has been rigorously validated (<https://doi.org/10.1109/TNS.2014.2335538>). Specifically, Star-XP employs the G4NDL4.6 dataset, derived from the latest ENDF database, ensuring reliable neutron simulation results at low energies ($E < 20$ MeV). This foundation in Geant4's validated neutron physics provides confidence in Star-XP's neutron simulation capabilities.

2. Detector Modeling and Simulation Scope: Regarding the reviewer's observation about simplified simulations in certain cases, we would like to clarify the specific purposes and scope of each simulation:

a) Liquid Scintillator Simulation: This focused simulation was specifically designed to obtain the unfolding matrix and reconstruct the induced neutron energy spectrum. The simplified approach was appropriate for this particular analysis goal.

b) TRIM Simulations: These simulations were primarily conducted to extract secondary track information and track length data from neutron and nuclear interactions with the gas medium. Given the specific objectives of these simulations, the inclusion of the full detector structure was not necessary.

3. Comprehensive Detector Simulation: For simulations involving the complete experimental setup, we have implemented a detailed detector model in Geant4, which includes:

- Gas-sensitive detector components
- Ceramic structures
- Lead and copper glass frame of GMCP
- Beryllium window
- Lead shielding box
- All other major structural elements present in the experimental configuration

The accompanying figure illustrates our comprehensive detector model, where:

- The large cylinder near the origin represents the GMCP detector
- The upper two cylinders depict the liquid scintillator setup for neutron flux and spectrum monitoring within the lead box

- The outermost square structure corresponds to the lead shielding enclosure

Fig.27 Illustrative figure of detector model used in simulation.

Below is a detailed breakdown of the main materials and their compositions used in our simulations:

Table 4: The primary materials and their composition ratios of the detector.

Material	Components	Density (g/cm ³)
DME	CH ₃ OCH ₃	0.00191855
Ceramics	Al ₂ O ₃	2.88
Kovar	52.49% Fe, 29% Ni, 17% Co, 0.5% Mn, 0.2% Si, 0.2% Cr, 0.2% Cu, 0.2% Mo, 0.1% Zr, 0.06% C, 0.025% P, 0.025% S	8.17
Stainless steel	88% Fe, 8% Co, 4% C	7.7
Liquid scintillator	C ₅ O ₂ H ₈	1.19
Lead glass	65.9% Pb, 19.84% O, 12.6% Si, 1.66% K	6.2

This comprehensive modeling approach ensures that our simulations accurately represent the experimental conditions and account for all relevant structural components.

To comprehensively address neutron activation background, we conducted a full-scale simulation of the entire detector setup, meticulously accounting for all potential neutron-induced interactions. Our simulation methodology and validation process are as follows:

1. Simulation Framework: Neutrons were generated through energy spectrum sampling and propagated through the entire detector system, including the lead shielding box, liquid scintillator, and GMCP detector. We recorded comprehensive event information, including:

- Track generation processes
- Vertex and recoil coordinates
- Step-wise energy deposition
- Track initial energies
- Track lengths

2. Analysis Procedure:

The analysis pipeline was implemented with rigorous selection criteria:

a) Initial Selection:

- Tracks originating within the gas-sensitive detector were selected
- All external tracks were rejected

b) Energy Classification:

- Energy depositions were separately summed for:
 - Compton electrons
 - Photoelectrons
 - Recoil nuclei
- Appropriate energy cuts were applied

c) Background Identification:

- Events generating both eligible electrons and recoil nuclei were flagged as potential background

3. Statistical Approach for Vertex Distance Cut:

While our simulation precisely recorded track vertices, the extremely low generation rates of Compton and photoelectrons (only a few hundred electrons satisfying the energy cut from 50 billion simulated neutrons) posed challenges for direct vertex distance analysis. To mitigate statistical uncertainties, we implemented a sophisticated sampling script that:

- a) Randomly sampled points within the gas-sensitive detector volume
- b) Generated vertices according to the measured rates of nuclear recoils (NR) and electrons
- c) Applied YOLO and event selection algorithms

Through this approach, we determined that the probability of random coincidence is 0.29%. This probability was then applied to the number of electrons passing the energy cut in our simulation.

4. It is important to note that the data-driven background result comprises two components that are experimentally indistinguishable:

- a) Neutron activation background:
 - Generated through neutron-induced gamma production
 - Produces Compton electrons and photoelectrons in the sensitive volume
- b) Environmental gamma background:
 - Simulated using gamma flux and spectrum from liquid scintillator measurements
 - Processed with identical analysis methods as neutron-induced background

For a robust validation of our neutron simulation, we compared the experimental data with the sum of both simulated background components:

- Neutron activation background (from our detailed simulation)
- Environmental gamma background (using the same analysis framework)

a. Gamma selection

b. Unfolded neutron spectrum

c. Unfolded gamma spectrum

Fig.28 (a) Results of neutron and gamma discrimination using Pulse Shape Discrimination (PSD) in the liquid scintillation detector, (b) Deconvoluted neutron energy spectrum, (c) Deconvoluted environmental gamma energy spectrum.

5. Background Estimation and Validation:

Combining the simulated NR counts with selected Compton and photoelectron events, and applying the vertex distance probability, we derived the neutron activation background estimate to be

$$817000 \times (167 + 41) \times \frac{0.0029}{3860962} = 0.1276 \pm 0.0089(\text{stat.})$$

When combined with the environmental gamma background which is $0.0091 \pm 0.0004(\text{stat.})$, the total simulated background is $0.14 \pm 0.0089(\text{stat.})$, showing agreement with the data-driven background estimation of $0.18 \pm 0.022(\text{stat.}) \pm 0.042(\text{sys.})$, demonstrating the reliability and accuracy of our simulation framework.

The close correspondence between our simulation-based prediction and the empirical data-driven result, considering both neutron and γ backgrounds, validates our approach to modeling neutron activation effects. This consistency, along with the extremely low neutron-induced background estimate of only 0.14 coincident events, provides strong confidence in our ability to accurately characterize and account for neutron-induced background effects in the experimental data.

Table 5: simulation cut-flow.

Simulated neutron events	50,000,000,000
Simulated NR fits energy cut	3860962
Generate both NR and Compton electron that fit energy cut	167
Generate both NR and photoelectron that fit energy cut	41

These two methods (simulation & data-driven) for calculating the background are based on the liquid scintillator detector and the Migdal detector, respectively. One method relies on the reaction cross-sections simulated by Geant4, while the other directly utilizes the detected electron track data. Both methods are consistent within the margin of error, which further validates the reasonableness of our background estimation. In the article, we adopted the result with a higher background expectation to provide a relatively conservative significance analysis, and still achieved a significance of 5σ . We hope these clarifications adequately address the reviewer's concerns regarding the validation of our neutron simulations and the completeness of our detector modeling. We are committed to maintaining the highest standards of accuracy and completeness in our simulation work.

Rare event search algorithm

Authors inform in "meth" that identification of ER an NR have been done with high accuracy using the YOLOv8 - a deep learning-based object detection algorithm. YOLO is a common tool used for broad spectrum of application, also in search for rare events. Description of implementation and usage of YOLO for their data analysis Authors present only in one, not very clear paragraph. For comparison: to show potential of YOLO algorithm for Migdal event recognition, Authors provide reference to the article of another experimental group which also search for Migdal effect, namely the Migdal Collaboration (ref. 22 in "meth": Preprint at <https://arxiv.org/abs/2406.07538> (2024)). The Migdal Collaboration, for showing performance of YOLO for their data analysis needed 24 pages of high quality scientific article. Thus, very

concise information about YOLO implementation provided in "meth" is by far not convincing. And there is no reference to any paper/thesis/proceeding which would describe YOLO performance in context of reviewed here work.

Re:

1. Motivation and Applications of YOLO

For the initial batch of experimental data, we manually select Migdal effect candidates through our eyes. However, this approach suffers from extremely low efficiency, making it impractical to analyze 8.17×10^5 events rapidly, while the uncertainties introduced by human judgment remain unquantifiable. YOLO, a deep learning-based image recognition tool, can significantly enhance efficiency and enable systematic error estimation.

Inspired by the MIGDAL Collaboration's work (<https://arxiv.org/pdf/2406.07538>), we choose YOLO for preliminary data filtering. Unlike their implementation, we use YOLO only to identify images containing at least one Nuclear Recoil (NR) and one Electron Recoil (ER) track, so our work requires no deep analysis of YOLO model. Due to manuscript length constraints, we also minimized technical descriptions of YOLO to prioritize space for other critical content.

2. Image Preprocessing

Similar to the MIGDAL Collaboration's approach, we use Label Studio to annotate data. Our detector's readout employs Topmetal with a pixel resolution of 72×72 . However, 72×72 images are too small for efficient labeling in Label Studio, so we upscale them proportionally to 288×288 pixels. To achieve optimal visualization, we apply a logarithmic colorbar to the pixel ADC values, ranging from 10 to 2600, with a gradient from blue to red, and use a white background for the images.

3. Classification and Labeling

The Migdal effect exhibits a unique topological signature characterized by ER and NR sharing a common vertex. In Label Studio, we only define two classes: ER and NR figure 28, as no other classes are required. Reasons are as follows:

- Cosmic rays, sparks and hot pixels. They are filtered out by separate algorithms, eliminating the need for manual labeling.

- Protons and alpha particles. First, we don't need to distinguish them and NRs, because their track morphologies closely resemble NRs, and the experimental gas contains hydrogen, and helium which can produce protons and alpha. Second, protons, alpha and NRs cannot be distinguished completely, because charge adsorption effects cause observable variations in dE/dx along tracks, implying that NRs are not purely bare nuclei, and their charge states are uncertain. Without experimentally calibrated training data of NRs, relying solely on simulated bare-nucleus samples could introduce significant model bias.

- Ghost tracks. During preprocessing, adjacent frames with signals are algorithmically merged into complete tracks. We perform pixel-wise calibration of the attenuation pattern and incorporate attenuation corrections between adjacent frames, as detailed in the Digitization section (PixelControl and TopmetalControl classes) of the Star-XP paper (<https://www.sciencedirect.com/science/article/pii/S2352711023003229>). This approach inherently eliminates ghost tracks.

Thus, defining only ER and NR classes suffices for identifying Migdal effect candidates.

Fig.29 The interface of Label Studio. It is used to annotate bounding boxes and classification labels.

4. Dataset

The composition of our dataset is shown in **Error! Reference source not found.6**.

Table 6: Dataset size. The size of experimental dataset represents the quantity of images (with a small probability of containing multiple tracks per image). The size of simulated dataset represents the quantity of tracks (each image contains only one track).

Dataset		Training	Validation
Experiment	⁵⁵ Fe	3000	1493
	D-D	2994	1354
Simulation	ER	1200	301
	NR	1200	301
Total		8394	3449

The experimental dataset is collected from the initial batch of our experiments, with the gas composition and pressure maintained consistently across all batches (40% He, 60% DME, 0.8 kPa). ⁵⁵Fe generates photoelectrons with an energy of 5.9 keV, which is comparable to the energy of Migdal electrons. The D-D neutron generator produces NR tracks.

Fig.30 Digitization time line model.

The simulated data originates from our Star-XP simulation framework, which has been experimentally validated to exhibit excellent agreement with measured electron track characteristics in the 3-8 keV range. This ensures the high reliability of Star-XP's simulation accuracy. Star-XP simulation framework is divided into three core components:

- Generators. Produce incident particles and background signals based on configurable parameters.

- Simulation. Import detector geometry via GDML files and simulate electron drift and diffusion in gas, as well as avalanche multiplication processes, using Garfield++ and COMSOL.

- Digitization. Simulate the Topmetal-II chip's readout process, including Rolling Shutter readout method, pixel-wise attenuation corrections between adjacent frames and trigger for signal thresholding (Figure 30).

Every step, from particle interaction to signal readout, is precisely simulated. The simulation accuracy is verified through experimental calibration. Electrons with 5.9 keV and 8.05 keV energy are used to validate track energy, circularity, and length distributions, and simulated data shows near-perfect alignment across all metrics (Figure 31).

Fig.11 Top: Comparison of track circularity and length between experimental and simulated data (8.05 keV). Bottom: Comparison of energy spectra (5.9 keV).

In the simulated dataset, ER events are modeled as electrons with energies ranging from 4-10 keV, compensating for the experimental dataset's limitation of containing only 5.9 keV mono-energetic electrons and enhancing the dataset's versatility. NR events are generated through neutron interactions in the detector, with the gas composition (40% He, 60% DME) and pressure (0.8 kPa) configured identically to the experimental setup. The incident neutron energy matches that of the D-D neutron generator, and the accuracy of neutron-gas interaction cross sections is ensured by Geant4 simulations. Due to the consistency of experimental conditions and the validated agreement between simulated and experimental data, there is no need to study the energy dependence of YOLO's selection efficiency. Instead, the efficiency derived from simulated data can be directly applied to experimental data.

5. Training and Validation

Initially, we evaluate different YOLOv8 model architectures (e.g., YOLOv8n and YOLOv8m) to balance accuracy and training efficiency. After comprehensive analysis, YOLOv8m was selected as the optimal architecture. For dataset augmentation, we applied 90°, 180°, 270° rotations and horizontal/vertical flipping, while incorporating some simulated data for transfer learning. The training process was configured with 180 epochs.

After training, we validate the model's performance using a validation dataset. Figure 32 shows the confusion matrix and F1 curve of our model. Figure 33 shows how the loss functions gradually converge with the increase of training epochs.

Fig.32 Top: The confusion matrix. The matrix element A_{ij} represents the probability that the j -th classification is predicted as the i -th classification. Bottom: F1-confidence curve. F1 score is the harmonic mean of precision and recall, ranging from 0 to 1, with higher values indicating better classification capability.

Fig.33 The loss functions converge with the increase of training epochs.

As shown in figure 33, the majority of tracks in the images are correctly predicted by the model, and the model classification exhibits high accuracy.

6. Efficiency, Error and Results

When validating the model's accuracy using the validation dataset, very few images contain both ER and NR tracks. Thus, these results cannot directly represent the efficiency of selecting Migdal effect candidates. To evaluate YOLO's efficiency in selecting Migdal candidates, we use Star-XP to simulated 1.5×10^5 events of Migdal effect (one ER and one NR per event) within the 5-10 keV ER energy range, based on the theoretical Migdal effect cross section. After applying YOLOv8m, 29.3% of events are retained where the model identifies at least one ER and one NR. Subsequently, our Migdal event selection algorithm further filters these candidates, achieving an additional 49.0% efficiency. The final reported efficiency of 14.4% is the product of these two efficiencies.

Additionally, the MIGDAL Collaboration's work mentions using bounding box-derived parameters (e.g., the energy of NR and distance between ER and NR) for data filtering. However, we do not use such parameters for our selection. Our sole filtering criterion is the presence of at least one ER and one NR per image and further filtering is not performed via YOLO, as YOLO's bounding box outputs lack critical parameters (e.g.,

circularity, dE/dx) required for track morphology analysis. Instead, we employ our Migdal event selection algorithm for subsequent filtering.

Regarding YOLO's systematic error, we quantify the discrepancy between YOLOv8m and YOLOv8n models trained on identical datasets and parameter configurations. After applying YOLOv8n, 25.6% of events are retained where the model identifies both ER and NR tracks. Subsequent filtering via the Migdal event selection algorithm achieves an additional 51.8% efficiency. Comparing to YOLOv8m, this results in a 1.9% systematic error in the final efficiency.

Finally, figure 34 displays the YOLO-predicted bounding boxes for the 6 selected Migdal effect events.

Fig.34 The images of Migdal effect predicted by YOLOv8, with bounding boxes and their classification labels.

Simulation

According to "meth" the software framework Star-XP (ref. 13: SoftwareX 25, 101626 (2024)) have been used for simulate Migdal events. But this software is optimized for simulations of low energy X-ray interactions in gaseous medium. There is no information if this software has been verified for simulations of effects associated with neutron interaction in

solid and gaseous component of detection setup.

It is also mentioned in "meth" that some simulation with the use of Geant4 and Trim have been performed. But it looks like that they were restricted only to interaction of neutrons with liquid scintillator and with detector gas medium.

Large drawback of this manuscript is a lack of any simulation results shown. The reader has no chance to develop his own opinion about their completeness and usefulness in respect to signal selection and background suppression.

Background

Very important discussion of background is presented in "meth". Many possible background sources are indicated and, to some extent, discussed. Authors' claims about negligibility of some background components e.g. Auger or bremsstrahlung what in view of low Z materials used for detection medium, and applied energy cut of 5 keV for ER is correct. Nevertheless, often Authors' conclusions about insignificance or low yields of individual background components are just given to belief and not well justified. Specially doubtful are estimations of random track coincidences and events related to neutron activation.

Except of plain text no any graphical support is provided. Simulations of background, which were performed, are not well documented. And they seem to be incomplete.

What evidently is missing are the consideration of effects induced by neutrons traveling through solid material of the detector and other apparatus. From the brief description of the apparatus in "ms" and "meth" it is clear that neutrons enter the medium gas through the detector wall. Wall is made of ceramics an Kova alloy. What is exact composition of these materials? It is imaginable that hadronic and electromagnetic processes in the detector wall induced by neutrons (and possibly gammas from the generator) create a

broad spectrum of prompt gammas which, in following, can mimic the electron recoils (e.g. by Compton process). The dominant element of Kova is copper. Stable isotopes of Cu have not negligible cross section for radiative neutron capture for MeV-neutrons. Perhaps this component is insignificant but cannot be disregarded when total background is estimated. Careful Geant4 simulations taking into account the composition of detector wall and geometry could verify significance of this source of background and estimate the amount of possible fake NR-ER coincidences induced by this background component.

Dear Editor and Reviewers,

We would like to thank the Editor very much for giving us the opportunity to improve and supplement the content of our manuscript. We also sincerely appreciate the reviewers for their careful reading of the content of our manuscript and the response materials, as well as their professional and meticulous comments. The reviewers' suggestions for supplementing and revising the article content have greatly contributed to enhancing the completeness and coherence of our manuscript.

We have carefully considered and addressed each of the reviewers' suggestions and comments, and made corresponding revisions and supplements to the manuscript. Below are our responses, where **the reviewers' comments are presented in blue text**, and **our replies are provided in black text**.

Referee 2:

- The sentence “The lack of direct observation undermines the conclusions drawn from DM experiments that rely on the existence of the Migdal effect, raising significant doubts about the reliability of their findings.” should be revised. Results from DM experiments so far cannot be based directly on the Migdal effect, as its contribution to the DM signal has not been precisely established, but only estimated through model-dependent assumptions. Therefore, I suggest softening this statement by removing the final part: “raising significant doubts about the reliability of their findings.”

Re: Thanks for the careful consideration of the wording in our manuscript. We fully understand and agree with the referee’s suggestion and have removed the phrase “raising significant doubts about the reliability of their findings” from the sentence.

- The authors have corrected the variables in Eq. (1); however, not all variables appearing in the formula are defined in the text. In particular, the variable n_{bg} and n_{obs}^{ER} described and should be explicitly defined.

Re: We would like to extend our gratitude to the reviewer for the meticulous examination. Upon reviewing, we discovered that due to version compatibility issues with our

document editor, some variable names in the submitted manuscript lost their superscripts during the revision process. The variable names n_{bg} and n_{obs} pointed out by the reviewer should actually be n_{obs}^{ER} and n_{obs}^{bg} respectively. We sincerely apologize for any confusion caused by these errors during your review process. To prevent such issues from recurring, we have thoroughly rechecked all symbols and formulas in the manuscript and have adopted a more compatible editing version.

- The suggestion to avoid reporting more than two digits in uncertainty values has been considered. However, the number of significant digits in the reported value must match the precision of its associated uncertainty. For instance, if the uncertainty has two digits at the 10^{-6} level, the central value must reflect that precision. As an example, the ratio of the Migdal cross-section to the nuclear recoil cross-section should be written as $((4.9)_{(-1.9)}^{(+2.6)}) \times 10^{-5}$ instead of $((4.91)_{(-1.9)}^{(+2.6)}) \times 10^{-5}$. This correction should be applied consistently throughout the manuscript, including in the Abstract, Ratio, Conclusion, and Table 2 of the Methods section. Another example of inconsistency is $0.035 \pm 0.023(\text{stat.}) \pm 0.0068(\text{sys.})$ should be $\pm 0.007(\text{sys.})$.

Re: We have made consistent revisions to the manuscript as required.

- The authors should include a dedicated section within the Methods specifically focused on the data analysis. This section must clearly describe the statistical methodology adopted and should also include the explanation—currently only provided in the response to reviewers—regarding the origin of the asymmetric error in the cross-section ratio. In particular, the authors are expected to detail the statistical approach used, the assumed distributions for signal and background events, and the plot of the resulting maximum likelihood profile. Providing these elements in the manuscript is essential to support the robustness of the analysis and to justify the claimed 5σ significance level of the experimental evidence.

Re: We sincerely appreciate the reviewer's suggestion for enhancing the detailed understanding of our manuscript. In response, we have thoroughly elaborated on the methodology employed for calculating significance within the "**Uncertainty &**

Significance" section of the **Method**. We have specified the distributions of the signal, background, and detection efficiency. Additionally, we have provided the likelihood functions used for computing significance and cross-section probabilities, along with their corresponding values at different significance levels, as well as the formulas for calculating the upper and lower limits of the cross-section probabilities. The specific details are presented as follows:

"

The profile likelihood method is used for significance calculation and Migdal effect probability error calculation. In this method, a probability model that depends on both the parameters of interest $\boldsymbol{\pi} = (\pi_0, \dots, \pi_k)$ and additional nuisance parameters $\boldsymbol{\theta} = (\theta_0, \dots, \theta_l)$ is used to describe the data. Denoting the density function as $f(\mathbf{X}|\boldsymbol{\pi}, \boldsymbol{\theta})$, where $\mathbf{X} = (X_0, \dots, X_n)$ refers to independent observations, the full likelihood function can be expressed as:

$$L(\boldsymbol{\pi}, \boldsymbol{\theta}|\mathbf{X}) = f(\mathbf{X}|\boldsymbol{\pi}, \boldsymbol{\theta}) \quad (1).$$

The general approach to constructing confidence intervals is to determine a corresponding hypothesis test. Here, the hypothesis test is $H_0: \boldsymbol{\pi} = \boldsymbol{\pi}_0$ versus $H_a: \boldsymbol{\pi} \neq \boldsymbol{\pi}_0$, and the test can be based on the likelihood ratio test statistic:

$$\lambda(\boldsymbol{\pi}_0|\mathbf{X}) = \frac{\sup\{L(\boldsymbol{\pi}_0, \boldsymbol{\theta}|\mathbf{X}); \boldsymbol{\theta}\}}{\sup\{L(\boldsymbol{\pi}, \boldsymbol{\theta}|\mathbf{X}); \boldsymbol{\pi}, \boldsymbol{\theta}\}} \quad (2).$$

A standard result in statistics is that $-2 \log \lambda$ converges in distribution to a χ^2 distribution. Therefore, we can determine the confidence level for the hypothesis $H_a: \boldsymbol{\pi} \neq \boldsymbol{\pi}_0$ by comparing the difference between $-2 \log \lambda$ across various intervals and its minimum value, and then referencing the corresponding χ^2 confidence intervals.

The observed counts X in this experiment follow a Poisson distribution: $X \sim \text{Pois}(\mu + b)$, where μ is the observed signal rate, and b is the observed background rate. The observed background counts Y are characterized by a Gaussian distribution: $Y \sim N(b, \sigma_b)$. Assuming X and Y to be independent then we have:

$$f(x, y|\mu, b) = \frac{(\mu + b)^x}{x!} e^{-(\mu+b)} \cdot \frac{1}{\sqrt{2\pi}\sigma_b} \exp\left(-\frac{(y-b)^2}{2\sigma_b^2}\right) \quad (3),$$

here, the hypothesis test is $H_0: \mu = 0$ versus $H_a: \mu \neq 0$. This model can be

implemented using the *model 5* in the *CERN ROOT*'s *TRolke* library to directly compute the profile likelihood function $-2 \log \lambda$, the results of which are displayed in Figure 1. The minimum of this likelihood function occurs at $\mu = 5.77$, and the corresponding 5σ value is greater than 0, providing significant evidence for the existence of the Migdal effect.

Fig.1 Likelihood function for the significance test of the Migdal effect.

For the estimation of the upper and lower limits of the cross-section probability errors, the profile likelihood method is also employed. It is important to note that the calculation of the cross-section probability must account for the signal efficiency. Therefore, the observed counts X in the signal region are modified to $X \sim \text{Pois}(en + b)$, where e is the signal selection efficiency and n is the number of Migdal events generated. The background counts Y remain characterized by $Y \sim N(b, \sigma_b)$. Additionally, the signal efficiency is assumed to follow a Gaussian distribution: $Z \sim N(e, \sigma_e)$. The profile likelihood function for this model can also be directly computed using *model 3* in the *TRolke* library. The resulting likelihood function, along with the positions of the minimum n_{\min} and its lower and upper limits n_{ll} and n_{ul} , are shown in Figure 2. The error propagation is applied as follows:

$$\Delta P_{\pm} = \sqrt{\left(\frac{n_{min} - n_{ul/l}}{n_{min}}\right)^2 + \left(\frac{n_{tot}^{NR_{error}}}{n_{tot}^{NR}}\right)^2} \cdot \frac{n_{min}}{n_{tot}^{NR}} \quad (4).$$

Fig 2. Likelihood function for the number of Migdal effect events. The upper and lower limits corresponding to a standard deviation are $n_{ll} = 24.88, n_{ul} = 60.89$, respectively.

"

- The extended theoretical discussion included in the review response would be valuable to readers if integrated—more concisely—into the Methods section. Figure 7 could also be added there to improve the reader’s understanding of the Migdal effect modelling and its consistency with the reported results.

Re: In accordance with the suggestion, we have elaborated in the **Theory** section on the method for calculating the Migdal effect probabilities for each individual element within a mixed gas. Moreover, to provide readers who are interested in the contributions of

different elemental components to the Migdal effect and the theoretical computations with detailed insights, we have compiled the discussion related to the theoretical calculations from our initial response and included it in the **supplementary material** for reference.

Referee 3:

1. Detection setup

In “meth” Authors provided a very schematic plot of a setup. One can learn that “Migdal detector” was kept at detection angle of 270 deg and beam parameters (rates, spectra) of neutrons and gammas were monitored with scintillators located at angles of 0, 87, and 180 deg. The lead shielding surrounding the detector was of the thickness of 1 cm (new information given in “ms”). Convincing information about neutron and gamma energy spectra are given in “meth”. Monitoring plots of the gas pressure, its temperature and counting rates are presented.

It become clear that no any beam collimation was used during experimental runs. The neutrons and gammas from generator could interact freely with all components of detection system. This is a drawback of the system, since some hadronic and electromagnetic processes undergoing in construction materials of the detector remain uncontrolled.

The lead shielding of 1 cm thickness is in 100% efficient to suppress the gammas of energies blow ~250 keV. This is sufficient to stop the most dominant part of measured gamma spectrum shown in fig. 9 b of “meth”. Gammas of larger energies can penetrate into Migdal detector. For example: for 2 MeV gammas attenuation in 1 cm lead plate is only 33%. Such photons can induce fake electron tracks by, predominantly, the Compton scattering.

Estimation of the rate of indicated above processes and their contribution to possible occasional NR – ER coincidences can be done with the use of careful simulations with the use of Geant4. Both “ms” and “meth” still missing explicitly provided information of such performed simulations. Fortunately, in the “reb” Authors describe convincingly their simulations strategy and the results (“Comprehensive Detector Simulation”). With no doubt, in order to avoid unnecessary confusion, the information about performed Geant4 simulations of the whole detection setup, their results and agreement with experimental results should be included to “meth”. Thus, I propose that

information contained in “3. Comprehensive Detector Simulation” of “reb” in compressed form are given in “meth”, for the knowledge of common readers.

Re: We sincerely appreciate your highly valuable suggestion. Considering the length constraints and logical coherence of the Methods section, we have compiled the content of interest to reviewers, as well as what we believe to be valuable to readers, into **supplementary materials** and uploaded them as attachments to the article. In the "**Background: Random Track Coincidences**" section of this supplementary material, we describe the modeling of the detector and its environment, including the main material compositions. We have divided the simulation of random coincidence backgrounds into three primary components, detailing their formation, simulation process, and results. Additionally, in the "**Random Track Coincidences**" section of the **Method**, we have added a paragraph to describe the estimation of this background using both simulation and data-driven approaches, along with a comparison of the background energy spectra obtained from these two methods. We hope that the added text, comparative figures, and supplementary materials will help readers better understand the related analysis.

Authors provided also new information that actually two Migdal detectors were used. Presumably they both had the same construction and performance quality. Would be interesting to know where the second detector have been installed and which of six Migdal event candidates were recorded in the second detector.

Re: In response to the reviewer's suggestion, we have updated the experimental site layout in the "**Experiment Details**" section of the **Method**. The updated figure illustrates the position of the second detector added during the second experiment, which is located at the same distance from the D-D source but on the opposite side relative to the first detector. The 4π emission characteristics of the D-D source, combined with the symmetric mechanical structure of the source itself, ensure that the neutron flux intensity and background environment at the second detector's position are essentially identical to those at the first detector.

We have added explanations in **figure 12**, indicating the corresponding Run and

detector number for each event.

2. Performance of detector

Very useful is new fig. 5 of “meth” where actual dimensions and electrical parameters of the “Migdal detector are shown” (for completeness, the internal diameter of the detector should be given). It clarifies that the fiducial volume of the detector is narrow (14 mm).

Authors provided also explicitly the demanded information about working parameters, their stability, performed position calibration and resolution (mainly in “reb”) and energy calibration and resolution. I found also the information about extrapolation of energy resolution. I would like to suggest to give this information already in chapter “Detector calibration” of “meth” and not only when cross section is discussed. Technique of measurement of position resolution Authors explained in “reb” and in referred there a work JINST 19, P04039 (2024). This knowledge however, will not be easily accessible for the common readers. Thus, I suggest to include a reference to JINST 19, P04039 (2024) also in “meth”.

Re: In accordance with the reviewer's suggestion, we have added descriptions of the linearity of the detector's response and the relationship between energy resolution and energy variation in the detector calibration section. Additionally, we have moved the references originally cited in the subsequent cross-section calculation section to the detector calibration section. Furthermore, we have included a citation to the literature on the measurement of the detector's position resolution at the corresponding location.

3. Simulations

In the “meth” Authors underline the usage of Star-XP model as a most general tool for relevant simulations. There is no doubt about the advantages of Star-XP model to simulate processes taking place in the gas detector medium and signal generation in the electronics. My concerns were about neutron and gamma interaction in the construction material. But I found no evidence that in Star-XP, designed for specialized application (low energy X-ray interactions in gaseous medium), the neutron interaction is implemented. The referred paper about Star-XP (SoftwareX 25, 101626

(2024)) is silent on this topic. In the current version of reviewed work there is also still not a clear information if neutron transport via whole detection system were considered.

As noted above, in “reb” Authors convincingly described their simulation of the detection system model with Geant4, which is, in my opinion, the most relevant for such application. As I have written in point 1 the “meth” should be supplemented with the information about Geant4 simulations of the whole detection system and their results.

Re: We sincerely appreciate the reviewer's comments. We realized that our description of the simulation framework was incomplete and could potentially lead to confusion. Our intention was to convey that because the digitization part of Star-XP was specifically developed for the type of detector we are using, adopting this simulation framework allows us to obtain simulated tracks that are highly consistent with experimental results. The particle interaction processes in Star-XP are still primarily simulated using the GEANT4 database. Therefore, we have revised and supplemented the description in the **Simulation** section of the **Method**. We have clarified that Star-XP utilizes the GEANT4 neutron cross-section database and have integrated our theoretically calculated Migdal cross-section data. Additionally, we have included a statement confirming that we have performed a complete modeling of the detector.

4. Background

After information in “reb” (also those addressed to other Referees) about performed simulations and their validation, in my opinion the background estimation is correct. When chapter “Simulation“ of “meth” will be supplemented with a message about performed comprehensive Genat4 simulations the current content of the “meth” part devoted to the background would be, in my opinion, sufficient.

In section “Random track coincidences” of ‘meth” misleading is the sentence: “Through a selection algorithm, the selection efficiency of the accidental coincidences is obtained to be 0.29%.” I suppose that the intention of the Authors is to say that “Through a selection algorithm the accidental coincidences were identified as 0.29% of all cases”.

Re: We have revised the relevant description based on the reviewer's comment to ensure clearer and more accurate communication of our intentions. In the "Random Track

Coincidences" section, we have modified the original text to: "Through a selection algorithm, accidental coincidences were identified as 0.29% of all cases." This adjustment aims to more accurately reflect our analysis results and avoid any potential misunderstandings.

5. Event topology

Explicitly shown dimensions of the working volume of the cell (fig. 5 in "meth") soften may previous concern about lack of 3D event topology reconstruction. Narrowness of the cell reduces the possibility of random vertex overlap of NR and ER due to 2D projection. Nevertheless, I can not agree with the Authors' statement, that "3D track reconstruction is not essential for the discovery of the Migdal effect" ("reb"). In my opinion, for announcing the first evidence of Migdal effect the clear 3D event topology fulfilling the kinematics restriction would be the most convincing. However, I agree, that precise reconstruction of a common vertex of correctly identified NR and ER tracks in the narrow cell, within a reasonable time window, at low event rate and with large signal/background ratio is a strong argument in favor of searched Migdal effect.

With the 2D projection of the tracks some amount of signal events can be not recognized due to ER and NR track overlap. This fact may influence the estimation of cross section for Migdal effect. In the current work Authors consider and estimate this uncertainty by means of simulation. This is another motive for extension of the "meth" with description of Authors' scheme of reasoning based on performed simulations.

Fig 3 of "ms" is now much more convincing than the previous one. Still explanations of colored contours are missing.

As requested, the topology and tracks parameters of all 6 event candidates are shown now in fig. 12 of "meth". Would be also interesting to know in which detector they were recorded. The ionization densities along the NR track are different for events a, b, c than for events d, e, f. Could it be concisely explained ?

Re: Based on the reviewer's suggestions, we have added the colorbar legend for the contour lines in Figure 3 and provided additional explanations for the meaning of the

colored contour lines in the caption of Figure 3.

Regarding the reviewers' concerns about the ionization rate of events, it is important to note that the working gas used in the experiment is a mixture of H, He, C, and O elements. The charge quantity z of different recoil nuclei, the electron capture process of ions [Eur. Phys. J. D (2023) 77: 163], and the velocity of recoil nuclei can all lead to significant differences in ionization density. Our detector does not have particle identification capabilities, so we are unable to make more in-depth judgments about the properties of recoil nuclei. Additionally, it is noted that the temperature monitoring during Run II showed that the operating temperatures of both detectors were slightly higher than during Run I by 5-8 K. This increase in temperature has caused the noise fluctuations of a small number of defective pixels on the pixel detector to become more pronounced. The frame output of the electronics uses the Difference Compression Method described in Method: 'The ADC value of each pixel per frame is stored and compared with the ADC value of the previous frame. If the difference exceeds a preset value, the pixel is identified as a signal pixel and transmitted to the erasure module, thereby achieving data volume compression.' This output strategy may result in a few blank pixels at the edges of the output tracks due to the increased noise fluctuations of defective pixels.

6. Rare event search algorithm

Explanations given by Authors in "reb" about event search algorithm are in my opinion convincing. For the credibility of their paper this information has to be made public. Due to the lack of previous publications by the authors on this topic, which could be cited, the "meth" again seems to be the place where a more extensive description of signal selection needs to be given. For this aim the relevant content of "reb", but in the shortened form can be used.

Re: We sincerely thanks for the suggestion. Due to the limited space in the **Method**, we have compiled the details and descriptions of the training, testing, application, and analysis of YOLO into the **supplementary materials** for interested readers. Additionally, in the **Method**, we have noted that our work was inspired by relevant research from the MIGDAL collaboration, providing readers with a reference for further information.

We have revised and supplemented the detailed issues pointed out by the reviewers one by one. It includes the modification of relevant expressions, the unification of picture format, the addition of icon notes and descriptions.

Referee 2:

Only one requested change has not been implemented, despite the authors' statement in their response.

This can be easily corrected in the final version, but in the reviewers' opinion it does not prevent the article from being suitable for publication. The minor change is reported again below:

- The sentence "The lack of direct observation undermines the conclusions drawn from DM experiments that rely on the existence of the Migdal effect, raising significant doubts about the reliability of their findings." should be revised. Results from DM experiments so far cannot be based directly on the Migdal effect, as its contribution to the DM signal has not been precisely established, but only estimated through model-dependent assumptions. Therefore, I suggest softening this statement by removing the final part: "raising significant doubts about the reliability of their findings."

Re: Following the reviewers' suggestions, we have removed the statement "raising significant doubts about the reliability of their findings."

Referee 3:

Fig. 3. In caption add the sentence: Superimposed are experimental points for ER and NR of 6 Migdal event candidates.

Re: We have added the statement "Superimposed are experimental points for ER and NR of 6 Migdal event candidates."

page 1: The term $p_v^i(n \kappa \rightarrow E_c)$ is $\dots \rightarrow$ The term $p_v^i(n \kappa \rightarrow E_c)$ is ...

Re: We have added the missing space.

page 1: To compared with $\dots \rightarrow$ To compare with \dots

Re: We have made the correction: changing "To compared with \dots " to "To compare with \dots ".

page 2: $E_{\max} = 10 \text{ keV}$ E_{nth} and E_{eth} are the thresholds of detector for nucleus and electron recoils, respectively. → $E_{\max} = 10 \text{ keV}$. E_{nth} and E_{eth} are the detection thresholds for nucleus and electron recoils, respectively.

Re: We have implemented the revisions on Page 2 as requested.

page 2, caption of Fig. 4: ... by binning the electrons from Migdal candidate events ... → ... by binning the electron energy of Migdal candidate events ...

Re: We have made the corresponding revision in the caption of Fig. 4 on Page 2, specifically changing the phrase "... by binning the electrons from Migdal candidate events ..." to "... by binning the electron energy of Migdal candidate events ...".

page 3: The transmission and power supply of the pixel chip information is connected to the external environment by ... → The power supply of the pixel chip and information transmission is realized by ...

Re: We have completed the revision on Page 3 as suggested, specifically adjusting the sentence from "The transmission and power supply of the pixel chip information is connected to the external environment by ..." to "The power supply of the pixel chip and information transmission is realized by ...".

page 3: The basic performance testing of the detector is tested → The basic performance of the detector is tested

Re: We have made the requested revision on Page 3, specifically revising the sentence from "The basic performance testing of the detector is tested" to "The basic performance of the detector is tested".

page 4: eMMC → embedded Multi-Media Card (eMMC)

Re: We have added the full name of "eMMC" on Page 4 as suggested, revising it to "embedded Multi-Media Card (eMMC)".

page 12: The training dataset ... with gas. - This sentence is too long.

Re: We have split the sentence "The training dataset ... with gas" on Page 12.

Page 14, caption of Fig. 12: and c → and (c)

Re: We have made the revision on Page 14, in the caption of Fig. 12, adding the missing parenthesis to "c", changing the phrase "and c" to "and (c)".

Page 15, caption of Fig. 13: ... different nucleon components ... ??? (nucleus ? Element ?)

Re: We have revised the relevant content on Page 15, in the caption of Fig. 13. we have changed "different nucleon components" to "... different nuclei in ...".

In "sup" l. 4: soft limit (The electron ... → soft limit (the electron ...

Re: We have made the requested revision in line 4, specifically adjusting the capitalization of the first letter inside the parenthesis—changing "soft limit (The electron ..." to "soft limit (the electron ...".

l. 29: which applicable → which is applicable

Re: We' ve corrected "which applicable" to "which is applicable" in line 29.

l. 87: as shown in equation 7. → as shown in equation 8.

Re: We' ve corrected "as shown in equation 7." to "as shown in equation 8." in line 87.

l. 114: outlined in a., → outlined in b.,

Re: We' ve revised "outlined in a.," to "outlined in b.," in line 114.

l. 161: Once the photon energy exceeds → Once the photon energy (ω) exceeds

Re: We' ve added the symbol ω in line 161.

l. 204: coincidentally coincide → coincidentally overlap (??)

→ accidentally coincide (??)

l. 222: as above

Re: We' ve replaced "coincidentally coincide" with "coincidentally overlap" in line 204, as suggested.

l. 402: Table → Table 7.

Re: We' ve added the table number in line 402, revising "Table" to "Table 7.".